# Adaptive kernel predictors from feature-learning infinite limits of neural networks

Clarissa Lauditi [1]  Blake Bordelon [1 2 3]  Cengiz Pehlevan [1 2 3]

## Abstract

Previous influential work showed that infinite width limits of neural networks in the lazy training regime are described by kernel machines. Here, we show that neural networks trained in the rich, feature learning infinite-width regime in two different settings are also described by kernel machines, but with data-dependent kernels. For both cases, we provide explicit expressions for the kernel predictors and prescriptions to numerically calculate them. To derive the first predictor, we study the large-width limit of feature-learning Bayesian networks, showing how feature learning leads to task-relevant adaptation of layer kernels and pre-activation densities. The saddle point equations governing this limit result in a min-max optimization problem that defines the kernel predictor. To derive the second predictor, we study gradient flow training of randomly initialized networks trained with weight decay in the infinite-width limit using dynamical mean field theory (DMFT). The fixed point equations of the arising DMFT defines the task-adapted internal representations and the kernel predictor. We compare our kernel predictors to kernels derived from lazy regime and demonstrate that our adaptive kernels achieve lower test loss on benchmark datasets.

## 1. Introduction

As neural-network-based artificial intelligence is increasingly impacting many corners of human life, advancing the theory of learning in neural networks is becoming more and more important, both for intellectual and safety reasons.

[*]Equal contribution  [1]John A. Paulson School of Engineering and Applied Sciences, Harvard University [2]Center for Brain Sciences [3]Kempner Institute. Correspondence to: Clarissa Lauditi <clauditi@g.harvard.edu>, Blake Bordelon <blake_bordelon@g.harvard.edu>, Cengiz Pehlevan <cpehlevan@seas.harvard.edu>.

*Proceedings of the 42nd International Conference on Machine Learning*, Vancouver, Canada. PMLR 267, 2025. Copyright 2025 by the author(s).

Progress in this endeavor is limited but promising. An influential set of results that motivates this paper identifies infinitely-wide neural networks under certain initializations as nonparametric kernel machines (Neal, 1995; Cho & Saul, 2009a; Lee et al., 2018; de G. Matthews et al., 2018; Arora et al., 2019b; Jacot et al., 2020; Lee et al., 2020). This is important because kernels are theoretically well-understood (Rasmussen & Williams, 2006; Scholkopf & Smola, 2001), offering rich mathematical frameworks for analyzing the expressivity and generalization properties of neural networks. A drawback of this identification is that it works in the *lazy* regime of neural network training (Chizat et al., 2020), i.e. when data representations are fixed at initialization and do not evolve during learning. However, state-of-the-art deep networks operate in the *rich*, feature learning regime (Geiger et al., 2020; Vyas et al., 2022; Yang & Hu, 2022; Vyas et al., 2023), where they adapt their internal representations to the structure of the data.

Motivated by these observations, here, we ask whether infinitely wide neural networks still admit nonparametric kernel predictors in the rich domain. And if so, what characterizes these predictors? Since in the rich regime data representations evolve dynamically according to the training dynamics, identifying the nature of predictors in this regime is crucial for uncovering the principles behind feature learning. Further, as wider networks are believed to perform better on the same amount of data (Hestness et al., 2017; Novak et al., 2018; Kaplan et al., 2020; Hoffmann et al., 2022), it opens the possibility of directly training infinitely-wide feature learning networks through their corresponding kernel machines for obtaining the best performing instances of a given model architecture.

In this paper, we derive kernel predictors for infinitely wide multilayer perceptrons (MLPs) and convolutional neural networks (CNNs) in the *rich* regime. They are kernel predictors in the sense that

$$f(\boldsymbol{x}) = \sigma\left(\sum_{\mu=1}^{P} a_\mu K(\boldsymbol{x}, \boldsymbol{x}_\mu)\right) \qquad (1)$$

where $\sigma$ is a nonlinear function, $a^\mu$ are scalar coefficients, $\{\boldsymbol{x}_\mu\}_{\mu=1}^{P}$ are training data, and $\boldsymbol{x}$ is the test point. The kernel $K$ depends on the architecture of the network, and,

importantly, adapts to the training data, unlike the Neural Tangent Kernel (NTK) (Jacot et al., 2020) and Neural Network Gaussian Process Kernel (NNGPK) (Cho & Saul, 2009b; Lee et al., 2018; de G. Matthews et al., 2018) derived from lazy infinite limits.

To arrive at these predictors, we start from existing work characterizing the feature-learning infinite-width limits of deep neural networks under gradient flow dynamics. Previously, Bordelon & Pehlevan (2022) adopted the dynamical mean-field theory (DMFT) (Martin et al., 1973; De Dominicis, 1978; Sompolinsky & Zippelius, 1981; Arora et al., 2019a; Arous et al., 2004) approach to the training dynamics of deep MLPs and CNNs under the *maximal update parameterization* ($\mu$P) (Yang & Hu, 2022; Yang et al., 2022; Mei et al., 2018; Rotskoff & Vanden-Eijnden, 2022), which leads to a feature-learning infinite-width limit. This DMFT results in a set of stochastic integro-differential equations in terms of summary statistics, which define the deterministic training time ($t$) evolution of the deterministic network predictor $f(\boldsymbol{x}, t)$. Performing inference with this framework requires solving these equations over the training time using sophisticated Monte Carlo techniques. Further, the complexity and history dependence of the equations make the predictor interpretation challenging.

Our main observation is that we can analyze the network predictor at convergence $f(\boldsymbol{x}, t = \infty)$ in two ways to obtain deterministic adaptive kernel predictors: (1) by interpreting the dynamics of gradient flow with added white noise as sampling the weights from a Bayesian posterior, (2) by studying the fixed points of DMFT equations for gradient flow with weight decay. Specifically, our contributions in this work are the following:

1. We study the noisy gradient-flow dynamics with weight decay in the rich regime for MLPs and CNNs. We identify two novel infinite-width limits (Table 1, Figure 1) that lead to adaptive kernel machine interpretations of neural networks. We name the corresponding kernels *adaptive Neural Bayesian Kernel* (aNBK) and *adaptive Neural Tangent Kernel* (aNTK) for reasons that will be apparent below.

2. To analyze the first of these limits, we introduce a novel Bayesian interpretation of feature-learning neural networks, where the posterior distribution at infinite width characterizes the network's state after training. Analyzing this posterior in the infinite width limit using statistical mechanics methods, we identify a min-max optimization problem, arising from a saddle point argument, that defines the aNBK predictor.

3. To analyze the second limit, we invoke the DMFT analysis of gradient-flow dynamics (Bordelon & Pehlevan, 2022). We show that when weight decay is added

to gradient flow dynamics the final learned network predictors behave as kernel predictors. We provide the DMFT fixed point equations that define the aNTK predictor.

4. We develop numerical methods to solve for our predictors.

5. For kernels arising from deep infinitely-wide linear Bayesian networks, we solve the saddle point equations exactly via recursion, bypassing sampling strategies needed for the nonlinear case. We analyze the behavior of kernel-task overlap parameters across depth for whitened data.

6. We provide comparisons of our adaptive kernel machines to trained networks and *lazy* NTK and NNGPK predictors for MLPs and CNNs. We demonstrate that our adaptive-kernels are descriptive of feature-learning neural network training on a variety of metrics, including test loss, intermediate feature kernels, and pre-activation densities. In addition, they outperform *lazy* kernel predictors on benchmark datasets.

## 1.1. Related works

**Neural networks as kernel machines.** In certain initialization and parameterization schemes, taking the width of a neural network to infinity leads the model to learn a kernel machine, with a kernel that depends only on the initial task-independent features which do not change during training (Jacot et al., 2020; Lee et al., 2018). This "lazy training" regime has been extensively studied, particularly in the context of infinitely wide networks (Chizat et al., 2020; Jacot et al., 2020; Lee et al., 2018; 2020). However, neural networks outside of the lazy limit (in the so-called "rich regime") often perform better than their corresponding initial NTKs (Vyas et al., 2022) and are not obviously related to kernel machines. Domingos (2020) argues that gradient descent training for any deep network corresponds to regression with a history-dependent "path kernel", though this definition does not satisfy the standard representer theorem where coefficients are only functions of the training data as in Equation (1). In general, the learned function of a network trained with gradient flow can always be written as an integral over the history of error coefficients on training data and a time evolving NTK. In the present work, however, we are focused on when the final solution a network converges to satisfies a *history independent* representer theorem as in Equation 1. Some experimental and theoretical works have indicated that this is often the case, where regression with the *final* NTK (computed with gradients at the final parameters) of a network provides an accurate approximation of the learned neural network function during rich training (Geiger et al., 2020; Atanasov et al., 2021; Wei et al., 2022). A complete theoretical understanding of when

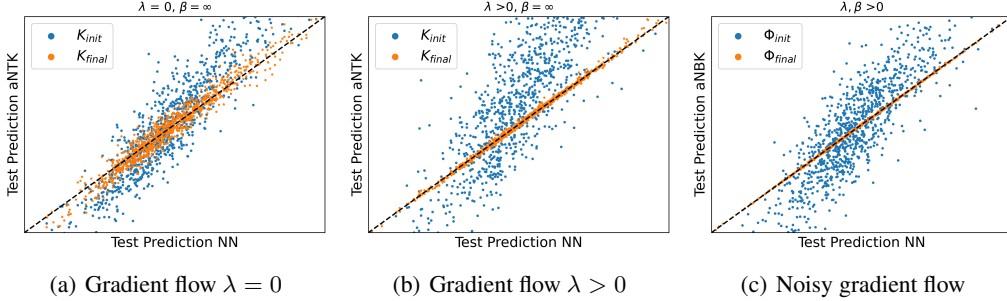

(a) Gradient flow $\lambda = 0$    (b) Gradient flow $\lambda > 0$    (c) Noisy gradient flow

*Figure 1.* Test network predictors of a two-layer MLP (width $N = 5000$) trained with $P = 300$ data of two-classes of CIFAR10 compared with the theoretical kernel regression predictors. Three panels are for different regimes of regularization $\lambda$ and temperature $1/\beta$. In all the three cases of $\lambda$, $K_{\text{init}}$ represent the network predictors at initialization, while $K_{\text{final}}/\Phi_{\text{final}}$ correspond either to aNTK at convergence for (a)/(b) or aNBK that at convergence for (c). (a) When $\lambda = 0$ the NN predictor of gradient flow without weight decay is not a kernel predictor; (b) instead, when $\lambda > 0$ the test prediction is well captured by $f_{\text{aNTK}}$ (Eq. (11)). (c) Bayesian empirical predictor is well-described by aNBK predictor at convergece. Here, the Bayesian predictor refers to the predictor obtained after thermalization when training with noisy gradient flow (i.e., Langevin dynamics). We expect these matches to be exact for adaptive kernels at infinite width.

and why the correspondence between final NTK and final network function holds is currently lacking.

**Adaptive kernels.** In Bayesian neural networks, several works have identified methods that describe learned network solutions beyond the lazy infinite-width description of NNGP regression. Some works pursue perturbative approximations to the posterior in powers of $1/\text{width}$ (Zavatone-Veth et al., 2022a; Roberts et al., 2022) or alternative cumulant expansions of the predictor statistics (Naveh & Ringel, 2021). Others analyze the Bayesian posterior for deep linear networks, which are more analytically tractable (Aitchison, 2020; Hanin & Zlokapa, 2023; Zavatone-Veth et al., 2022b; Bassetti et al., 2024) and actually also capture the behavior of deep Bayesian nonlinear student-teacher learning in a particular scaling limit (Cui et al., 2023). Several works on Bayesian deep learning have argued in favor of a proportional limit where the samples and width are comparable (Li & Sompolinsky, 2021; Pacelli et al., 2023; Aiudi et al., 2023; van Meegen & Sompolinsky, 2024; Baglioni et al., 2024; Fischer et al., 2024). In the context of fully connected (MLP) linear neural networks, (Li & Sompolinsky, 2021; Pacelli et al., 2023) argued that, the mean predictor under the posterior is the same as regression with the lazy NNGP kernel, though with a rescaled ridge and a predictor variance that depends on scale factor $Q(\alpha)$ that both change as a function of $\alpha = P/N$. Extending this result to deep networks Pacelli et al. (2023) found that each layer has a scale renormalization constant $Q_\ell(\alpha)$, while Aiudi et al. (2023) showed that convolutional architectures are characterized by a matrix of scale renormalization constants $Q_{s,s'}(\alpha)$ that capture learned space $\times$ space that adapt to the data.

In the same NTK parameterization, Fischer et al. (2024) and Seroussi et al. (2023) developed theories of feature learning where kernels adapt more flexibly (each entry in the kernel

can adapt due to feature learning). In this setting, feature learning acts as a $1/\text{width}$ effect in the posterior measure. Fischer et al. (2024) derive a general large deviation principle for the distribution of kernels under the posterior, which holds for any $\alpha = P/N$, however, they focus on linear response theory for the kernel updates since feature learning induces $\mathcal{O}(N^{-1})$ corrections to the the posterior distribution of kernels. Since the kernels do not concentrate in the proportional limit $P, N \to \infty$ with $P/N = \alpha$, Fischer et al. (2024) also compute how fluctuations in the kernels propagate across layers. They show that the linear response theory used to track fluctuations can also be used to compute feature learning corrections when linearizing the action around the NNGP kernels. Their results are thus non-perturbative in $\alpha = P/N$ but perturbative in what we call "richness" $\gamma_0$. Similarly, (Seroussi et al., 2023) explore variational approximations of the hidden neuron activation densities at finite width. More recent versions of these proportional theories have begun to explore other parameterizations (Rubin et al., 2024a) in linear networks, resorting to cumulant expansion of activation densities (up to Gaussian order) in one-hidden layer non-linear networks.

Closer to our approach, Yang et al. (2023) analyze rescalings of the Bayesian likelihood to induce representation learning at infinite width. This approach has also been used to explain sharp transitions in the behavior of the posterior as hyperparameters, such as prior weight variance, are varied in large width networks (Rubin et al., 2024b). Alternatively, some works have developed non-neural adaptive kernel algorithms, which filter information based on gradients of a kernel solution with respect to the input variables (Radhakrishnan et al., 2022), which exhibit improvements in performance over the initial kernel and can capture interesting feature learning phenomena such as grokking (Mallinar

et al., 2024).

In our work, we study the infinite width $N \to \infty$ limit of Bayesian networks at fixed $P$ in the *mean-field ($\mu P$)* parameterization setting. As a consequence, feature learning is not driven by finite width and cannot generally be extracted from linear response theory (or can be interpreted in terms of fluctuations since kernels concentrate). Contrary to previous results (Li & Sompolinsky, 2021; Pacelli et al., 2023; Aiudi et al., 2023; van Meegen & Sompolinsky, 2024; Baglioni et al., 2024), in our theories, the kernels deterministically adapt to data. These kernels can exhibit arbitrarily large changes in their structure (rather than just scale or spatial renormalization) at infinite width, and these cannot be obtained from linear response theory, at odds with previous computations (Fischer et al., 2024; Seroussi et al., 2023; Rubin et al., 2024a) (see Figures 1 & 3). Additionally, our adaptive kernels at infinite width are deterministic quantities, while in previous scaling theories, one has to compute kernel fluctuation statistics to get non-parametric predictors in the feature learning regime. To the best of our knowledge, we are also the first to solve for these adaptive kernels in the Bayesian setting without resorting to any perturbative (in richness $\gamma_0$) approach nor any Gaussian approximation of hidden layer pre-activation distributions, which are non-Gaussian at infinite width.

## 2. Preliminaries

We start by describing our setup. Here, for ease of presentation, we discuss the fully connected MLP setting, referring to Appendix D for the case of CNNs.

For an empirical training dataset $\mathcal{D} = \{\boldsymbol{x}_\mu, y_\mu\}_{\mu=1}^P$ of size $P$, input vectors $\boldsymbol{x}_\mu \in \mathbb{R}^D$ and labels $y_\mu$, we define the output of an MLP with $L$ layers as

$$
\begin{aligned}
f_\mu &= \sigma\left(\frac{1}{\gamma\sqrt{N_L}}\boldsymbol{w}^{(L)}\cdot\phi(\boldsymbol{h}_\mu^L)\right), \\
\boldsymbol{h}_\mu^{\ell+1} &= \frac{1}{\sqrt{N_\ell}}\boldsymbol{W}^\ell\phi(\boldsymbol{h}_\mu^\ell), \quad \boldsymbol{h}_\mu^1 = \frac{1}{\sqrt{D}}\boldsymbol{W}^{(0)}\boldsymbol{x}_\mu,
\end{aligned}
\tag{2}
$$

where $\phi(\cdot)$ represents a homogeneous transfer function in its parameters $\phi(a\boldsymbol{\theta}) = a^\kappa\phi(\boldsymbol{\theta})$, $\boldsymbol{h}_\mu^\ell \in \mathbb{R}^{N_\ell}$ at layer $\ell$ is the pre-activation vector and $\boldsymbol{W}^\ell \in \mathbb{R}^{N_{\ell+1}\times N_\ell}$ is the matrix of weights to be learned. We initialize each trainable parameter as a Gaussian random variable $W_{ij}^\ell \sim \mathcal{N}(0,1)$ with unit variance, in such a way that in the infinite width limit $N_\ell = N \to \infty$, $\forall\ell \in \{L\}$ the pre-activations at each layer will remain $\Theta_N(1)$. At the same time, we scale the network output by a factor $\gamma = \gamma_0\sqrt{N}$ in order to study feature learning. This allows to interpolate between a lazy limit description of NNs when $\gamma_0 \to 0$, and a rich regime description when $\gamma_0 = \Theta_N(1)$. This parameterization, known as *maximal update parameterization ($\mu P$)* (Yang & Hu, 2022), allows feature learning by enabling preactivations to evolve from their initialization during training even in the infinite-width limit.

## 3. Adaptive kernel limits of training dynamics

Next, we study the rich training dynamics of the NN defined as in Eq. (2) to arrive at adaptive kernel predictors. In particular, we study the infinite time limit $t \to \infty$ of the noisy gradient-flow dynamics

$$
d\boldsymbol{\theta}(t) = -\gamma^2\nabla_{\boldsymbol{\theta}}\mathcal{L}\,dt - \lambda\beta^{-1}\boldsymbol{\theta}(t)\,dt + \sqrt{2\beta^{-1}}d\boldsymbol{\epsilon}(t) \tag{3}
$$

for the collection of weights $\boldsymbol{\theta} = \text{Vec}\{\boldsymbol{W}^{(0)},\ldots,\boldsymbol{w}^{(L)}\}$, a loss function $\mathcal{L}(\boldsymbol{\theta})$ and for a ridge $\lambda$, $\forall\ell \in \{L\}$. Here, $\gamma^2 = \gamma_0^2 N$ ensures the feature updates to be $\Theta_N(1)$ in the infinite width limit (Bordelon & Pehlevan, 2022), and $d\boldsymbol{\epsilon}$ is a Brownian motion with covariance structure $\langle d\boldsymbol{\epsilon}(t)d\boldsymbol{\epsilon}(t')\rangle = \delta(t-t')\boldsymbol{I}$ (being $\delta$ the Dirac delta), whose contribution to the dynamics can be switched off by tuning the temperature $T = \frac{1}{\beta} \to 0$. When the Brownian motion is on, this dynamics can be interpreted as sampling from a Bayesian posterior, which will be detailed below. When it is turned off, this is gradient-flow dynamics with weight decay.

It is well know that infinite limits of this training dynamics in the lazy regime ($\gamma_0 \to 0$) lead to kernel machines defined by the NTK (Jacot et al., 2020) and NNGPK (Cho & Saul, 2009b; Lee et al., 2018; de G. Matthews et al., 2018). Here, we show that there are infinite limits in the rich regime that also lead to kernel predictors, but this time these kernels adapt to data. The order of limits for width, time, temperature, and feature learning strength parameters $\{N,t,\beta,\gamma_0\}$ to get either the already known *lazy* (NNGPK, NTK) or novel adaptive kernel predictors is shown in Table 1. The latter kernels correspond either to the infinite width limit of a NN at convergence (i.e. $t \to \infty$) that learns with $\beta > 0$ (aNBK) or to the infinite time limit of an infinitely wide NN learning with gradient flow and weight decay (i.e. $\beta \to \infty$) (aNTK).

| NNGPK | aNBK (ours) |
|---|---|
| $\lim_{\gamma_0\to 0}\lim_{N\to\infty}\lim_{t\to\infty}$ $\beta = \Theta_N(1)$ | $\lim_{N\to\infty}\lim_{t\to\infty}$ $\{\gamma_0,\beta\} = \Theta_N(1)$ |

| NTK | aNTK (ours) |
|---|---|
| $\lim_{\gamma_0\to 0}\lim_{N\to\infty}\lim_{\beta\to\infty}$ $t = \Theta_N(1)$ | $\lim_{t\to\infty}\lim_{N\to\infty}\lim_{\beta\to\infty}$ $\gamma_0 = \Theta_N(1)$ |

Table 1. Limiting orders for $\{N,t,\beta,\gamma_0\}$ in the dynamics (3) to get either (known) static kernels NNGPK & NTK (left column), or (new) adaptive kernel predictors (right column).[2]

Next, we give the functional forms of our novel aNTK and aNBK predictors and briefly discuss their derivations. Details are given in Appendix A.

## 3.1. aNBK

As described in Table 1, if we take first the $t \to \infty$ limit, and then the $N \to \infty$ limit with temperature $(1/\beta)$ and feature learning strength $(\gamma_0)$ fixed, we arrive at an adaptive kernel predictor (see Fig. 1)

$$f_{\text{aNBK}}(\boldsymbol{x}) = \sigma \left( \frac{\beta}{\lambda_L} \sum_{\mu=1}^{P} \Delta_\mu \Phi^L(\boldsymbol{x}_\mu, \boldsymbol{x}) \right), \quad (4)$$

with $\Delta_\mu = -\frac{\partial \mathcal{L}}{\partial s_\mu}$ being the error signal, $s_\mu = \frac{1}{\gamma_0 N} \boldsymbol{w}^{(L)} \cdot \phi(\boldsymbol{h}_\mu^L)$ the pre-readout as in Eq. 2, and $\Phi_\mu^L = \frac{1}{N} \phi(\boldsymbol{h}_\mu^L) \cdot \phi(\boldsymbol{h}^L)$ the train-test feature kernel at the last layer $L$. In this notation, the kernel matrix element is given by $\Phi_{\mu\nu}^L = \Phi^L(\boldsymbol{x}_\mu, \boldsymbol{x}_\nu)$. In the squared loss case with a linear readout $(\sigma(s) \equiv s)$, the predictor $f(\boldsymbol{x})$ becomes a kernel regression predictor of the form

$$f_{\text{aNBK}}(\boldsymbol{x}) = (\boldsymbol{\Phi}^L(\boldsymbol{x}))^\top \left[ \boldsymbol{\Phi}^L + \lambda_L \frac{\boldsymbol{I}}{\beta} \right]^{-1} \boldsymbol{y}, \quad (5)$$

where $\Phi^L(\boldsymbol{x})_\mu = \Phi^L(\boldsymbol{x}, \boldsymbol{x}_\mu)$ (for derivation, see Appendix A.2).

In Appendix A, we show that the kernel $\Phi^L$ is given by a solution to a min-max optimization problem that involves the data-adaptive kernel $\boldsymbol{\Phi}^L \in \mathbb{R}^{P \times P}$, intermediate layer adaptive-kernels $\boldsymbol{\Phi}^\ell \in \mathbb{R}^{P \times P}$ and dual adaptive-kernel variables $\hat{\boldsymbol{\Phi}}^\ell \in \mathbb{R}^{P \times P}$. Here, we present this min-max problem for the squared loss and linear readout for simplicity, see Appendix A for the full expressions. First, we define the action:

$$S(\{\boldsymbol{\Phi}^\ell, \hat{\boldsymbol{\Phi}}^\ell\}) = -\frac{1}{2} \sum_{\ell=1}^{L} \operatorname{Tr} \boldsymbol{\Phi}^\ell \hat{\boldsymbol{\Phi}}^\ell + \frac{\gamma_0^2}{2} \boldsymbol{y}^\top \left( \frac{\boldsymbol{I}}{\beta} + \frac{\boldsymbol{\Phi}^L}{\lambda_L} \right)^{-1} \boldsymbol{y}$$
$$- \sum_{\ell=1}^{L-1} \ln \mathcal{Z}_\ell[\boldsymbol{\Phi}^{\ell-1}, \hat{\boldsymbol{\Phi}}^\ell]. \quad (6)$$

where the functions $\mathcal{Z}_\ell[\boldsymbol{\Phi}^{\ell-1}, \hat{\boldsymbol{\Phi}}^\ell]$ are defined as

$$\mathcal{Z}_\ell[\boldsymbol{\Phi}^{\ell-1}, \hat{\boldsymbol{\Phi}}^\ell] = \int d\boldsymbol{h}^\ell \exp \left( -\frac{\lambda_{\ell-1}}{2} (\boldsymbol{h}^\ell)^\top (\boldsymbol{\Phi}^{\ell-1})^{-1} \boldsymbol{h}^\ell \right)$$
$$\exp \left( -\frac{1}{2} \phi(\boldsymbol{h}^\ell)^\top \hat{\boldsymbol{\Phi}}^\ell \phi(\boldsymbol{h}^\ell) \right), \quad (7)$$

with base case $\Phi_{\mu\nu}^0 \equiv \frac{1}{D} \boldsymbol{x}_\mu \cdot \boldsymbol{x}_\nu$. Then, the saddle point that dominates the distribution is

$$\{\boldsymbol{\Phi}_\star^\ell, \hat{\boldsymbol{\Phi}}_\star^\ell\}_{\ell=1}^{L} = \arg \min_{\{\boldsymbol{\Phi}^\ell\}} \max_{\{\hat{\boldsymbol{\Phi}}^\ell\}} S(\{\boldsymbol{\Phi}^\ell, \hat{\boldsymbol{\Phi}}^\ell\}). \quad (8)$$

*Derivation sketch.* Taking $t \to \infty$ at fixed temperature $\beta = \frac{1}{T}$ and finite width $N$ in Eq. (3) converges to a stationary distribution (Kardar, 2007; Welling & Teh, 2011;

Mingard et al., 2020; Naveh et al., 2021) over the trainable parameters $\boldsymbol{\theta}$ given the dataset $\mathcal{D}$, which can be interpreted as a Bayesian posterior with log-likelihood $-\beta\gamma^2 \mathcal{L}(\boldsymbol{\theta})$ and a Gaussian prior of scale $\lambda_\ell^{-1/2}$

$$p(\boldsymbol{\theta}|\mathcal{D}) = \frac{1}{Z} \exp \left[ -\beta\gamma^2 \mathcal{L}(\boldsymbol{\theta}) - \sum_{\ell=0}^{L} \frac{\lambda_\ell}{2} ||\boldsymbol{\theta}^\ell||^2 \right]. \quad (9)$$

The distribution of Eq. (9) can be studied in the overparameterized thermodynamic limit where the width at each layer $N \to \infty$ while $P = \Theta_N(1)$. When $\beta \to \infty$, the posterior is dominated by the set of global minimizers of the loss for the training data (i.e. solutions $\boldsymbol{\theta}$ that minimize $\mathcal{L}(\boldsymbol{\theta})$). We are interested in the partition function $Z$ of Eq. (9), which is function of the order parameters $\{\boldsymbol{\Phi}^\ell, \hat{\boldsymbol{\Phi}}\}_{\ell=1}^{L}$, since it has the form $Z \propto \int \prod_{\ell=1}^{L} d\boldsymbol{\Phi}^\ell d\hat{\boldsymbol{\Phi}}^\ell \exp \left( -NS(\boldsymbol{\Phi}^\ell, \hat{\boldsymbol{\Phi}}^\ell) \right)$ where $S(\boldsymbol{\Phi}^\ell, \hat{\boldsymbol{\Phi}}^\ell)$ is the Bayesian action given in Equation (6). When $N$ is large, $p(\boldsymbol{\theta}|\mathcal{D})$ is exponentially dominated by the saddle points of $S$, and looking for the values of $\{\boldsymbol{\Phi}^\ell, \hat{\boldsymbol{\Phi}}\}_{\ell=1}^{L}$ which makes $S$ locally stationary means solving the min-max optimization problem in Algorithm 1. In the lazy learning limit $\gamma_0 \to 0$, the dual kernels vanish $\hat{\boldsymbol{\Phi}}^\ell \to 0$ and we recover Gaussian preactivation densities, consistent with the NNGP theory (Lee et al., 2018). A full account of this derivation is given in Appendix A.1. □

We present a numerical algorithm to calculate the aNBK Regression Predictor from data in Alg. 1.

---

**Algorithm 1** aNBK Regression Predictor

---

**Input:** Dataset $\mathcal{D} = \{\boldsymbol{x}_\mu, y_\mu\}_{\mu=1}^{P}$ with covariance $\Phi_{\mu\nu}^0 = \frac{1}{D} \boldsymbol{x}_\mu \cdot \boldsymbol{x}_\nu$, hyperparameters $\{\gamma_0, \beta, \lambda_\ell\}$, step size $\delta$.
**Output:** Predictor $f(\boldsymbol{x})$ for any test point $\boldsymbol{x} \in \{P_{\text{test}}\}$.
**Generate:** Initial guesses for $\{\boldsymbol{\Phi}^\ell, \hat{\boldsymbol{\Phi}}^\ell\}_{\ell=1}^{L}$:
 $\boldsymbol{\Phi}^\ell = \langle \phi(\boldsymbol{h}^\ell)\phi(\boldsymbol{h}^\ell)^\top \rangle_{\boldsymbol{h}^\ell \sim \mathcal{N}(0, \boldsymbol{\Phi}^{\ell-1})}$, and $\hat{\boldsymbol{\Phi}}^\ell = 0 \quad \forall \ell$.
**while** *Kernels do not converge* **do**
  Define action the action $S$ of Equation (6) as differentiable function of $\{\boldsymbol{\Phi}^\ell, \hat{\boldsymbol{\Phi}}^\ell\}$, using importance sampling to estimate the functions $\mathcal{Z}_\ell$.
  Solve the inner optimization problem

$$\max_{\hat{\boldsymbol{\Phi}}^\ell} S(\{\boldsymbol{\Phi}^\ell, \hat{\boldsymbol{\Phi}}^\ell\})$$

  with gradient ascent
  Perform gradient updates on feature kernels

$$\boldsymbol{\Phi}^\ell \leftarrow \boldsymbol{\Phi}^\ell - \delta \frac{\partial}{\partial \boldsymbol{\Phi}^\ell} S(\{\boldsymbol{\Phi}^\ell, \hat{\boldsymbol{\Phi}}\}^\ell)$$

---

**Compute:** For a test point $\boldsymbol{x}$, $\boldsymbol{\Phi}^\ell(\boldsymbol{x}) = \langle \phi(\boldsymbol{h}^\ell(x))\phi(\boldsymbol{h}^\ell) \rangle_{p(h, \boldsymbol{h})}$ with importance sampling and $p(h, \boldsymbol{h})$ from Eq. (7) (see A.2).
**return** $f_{aNBK}(\boldsymbol{x})$ *as in Eq.* (5)

---

## 3.2. aNTK

The second order of limits we study is when $\beta \to \infty$ in Eq. (3). We are interested in the infinite time limit $t \to \infty$ of the dynamics when $N \to \infty$, which leads to a predictor

$$f_{\text{aNTK}}(\boldsymbol{x}) = \sigma \left( \frac{1}{\kappa \lambda_L} \sum_{\mu=1}^{P} \Delta_\mu K^{a\text{NTK}}(\boldsymbol{x}_\mu, \boldsymbol{x}) \right), \quad (10)$$

where again $\Delta_\mu = -\frac{\partial \mathcal{L}}{\partial s_\mu}$, $s_\mu$ the output pre-activation and $K_{\mu\nu}^{\text{aNTK}} = \lim_{t\to\infty} \frac{\partial f_\mu(t)}{\partial \boldsymbol{\theta}} \cdot \frac{\partial f_\nu(t)}{\partial \boldsymbol{\theta}} = \lim_{t\to\infty} \sum_\ell G_{\mu\nu}^{\ell+1}(t,t) \Phi_{\mu\nu}^\ell(t,t)$ is the *adaptive Neural Tangent Kernel*. The gradient kernel $G_{\mu\nu}^\ell(t,t) = \frac{1}{N} \boldsymbol{g}_\mu^\ell(t) \cdot \boldsymbol{g}_\nu^\ell(t)$ represents the inner products of gradient vectors $\boldsymbol{g}_\mu^\ell(t) \equiv N\gamma_0 \frac{\partial s_\mu(t)}{\partial \boldsymbol{h}_\mu^\ell(t)}$ which are usually computed with back-propagation.

For a squared loss and a linear readout, a homogenous activation function $\phi(\cdot)$, the final predictor has the form

$$f_{\text{aNTK}}(\boldsymbol{x}_\star) = \boldsymbol{k}^{\text{aNTK}}(\boldsymbol{x}_\star)^\top [\boldsymbol{K}^{\text{aNTK}} + \lambda_L \kappa \boldsymbol{I}]^{-1} \boldsymbol{y}. \quad (11)$$

The factor $\kappa$ appears in the expressions from the weight decay contribution to the dynamics (3), since we consider a $\kappa$ degree homogeneous network as specified in 2.

In the infinite width limit $N \to \infty$, the neurons in each hidden layer become independent and the $\Phi, G$ kernels can be computed as averages (Bordelon & Pehlevan, 2022)

$$\begin{aligned} \Phi_{\mu\nu}^\ell(t,t) &= \langle \phi(h_\mu^\ell(t)) \phi(h_\nu^\ell(t)) \rangle \\ G_{\mu\nu}^\ell(t,t) &= \langle g_\mu^\ell(t) g_\nu^\ell(t) \rangle \end{aligned} \quad (12)$$

where $\langle \cdot \rangle$ denotes the averages over a stochastic process for pairs $\{h_\mu^\ell, z_\mu^\ell\}_{\mu=1}^P$ which obey the dynamics

$$\begin{aligned} h_\mu^\ell(t) &= e^{-\lambda t} \xi_\mu^\ell(t) \\ &+ \gamma_0 \int_0^t dt' \, e^{-\lambda(t-t')} \sum_\nu \Delta_\nu(t') \, g_\nu^\ell(t') \, \Phi_{\mu\nu}^{\ell-1}(t,t') \\ z_\mu^\ell(t) &= e^{-\lambda t} \psi_\mu^\ell(t) \\ &+ \gamma_0 \int_0^t dt' \, e^{-\lambda(t-t')} \sum_\nu \Delta_\nu(t') \phi(h_\nu^\ell(t')) G_{\mu\nu}^{\ell+1}(t,t') \\ g_\mu^\ell(t) &= \dot{\phi}(h_\mu^\ell(t)) z_\mu^\ell(t) \end{aligned} \quad (13)$$

where $\xi_\mu^\ell(t), \psi_\mu^\ell(t)$ are stochastic processes inherited from the initial conditions, which become suppressed at large times. These equations are one way (but not the only way) to converge to a set of fixed point condition for the final features and final predictor (see Appendix F). Alg. 2 provides pseudocode for calculating the predictor in this setting.

*Derivation sketch.* A derivation to obtain the DMFT dynamics as in Eq. (13) can be found in (Bordelon & Pehlevan, 2022). Here, we want to stress the difference between

the $\lambda = 0$ (disccused in (Bordelon & Pehlevan, 2022)) and $\lambda > 0$ cases, the last of which leads to the kernel predictor as in Eq. (10). In the case of gradient flow with weight decay, since the predictor dynamics can be written by the chain rule $\frac{df_\mu}{dt} = \frac{df_\mu}{ds_\mu} \frac{ds_\mu}{dt}$, we can just track the dynamics of the output pre-activation

$$\frac{ds_\mu}{dt} = \sum_{\alpha=1}^P K_{\mu\alpha}^{\text{aNTK}}(t,t) \Delta_\alpha(t) - \lambda_L \kappa s_\mu \quad (14)$$

if we suppose $\sigma'(s_\mu) \neq 0$ [3]. How to arrive at this formula can be found in Appendix E. Here, at the fixed point the output pre-activation is $s_\mu = \frac{1}{\lambda_L \kappa} \sum_\mu \Delta_\mu K^{\text{aNTK}}(t,t)$, which recovers the kernel predictor of Eq. (10). $\square$

In principle, for any given value of $\lambda$, in order to have an estimate of $\boldsymbol{K}^{a\text{NTK}}$ at convergence, one has to simulate the stochastic non-Markovian field dynamics as shown in Algorithm 2. When $\lambda > 0$, the contribution from initial conditions $\xi_\mu(t), \psi_\mu^\ell(t)$ (see Eq. (13)), is exponentially suppressed at large time, while the second term of Eq. (13) contributes the most only when the system has reached convergence. This is not true if we switch-off the regularization $\lambda$, in which case the contribution from $\xi_\mu^\ell(t), \psi_\mu^\ell(t)$ persist late in training, since without the weight decay term, the initial conditions prevent the dynamics from converging to a fixed kernel predictor.

In Fig. 1 we clearly demonstrate this, by comparing the predictor of a two-layer MLP trained on a subset of CIFAR10 with the theoretical predictor of Eq. (11). For the first case when $\lambda = 0$, the network predictor at convergence is not the kernel predictor aNTK. Instead, when $\lambda > 0$, the network dynamics is well-described by Eq. (11). We refer to Appendix E.1 for the case of CNNs.

---

**Algorithm 2** aNTK Regression Predictor

**Input:** Data $\boldsymbol{\Phi}^0$, $\boldsymbol{y}$ and hyperparameters $\{\gamma_0, \lambda_\ell\}$.
**Output:** Predictor $f_{\text{aNTK}}(\boldsymbol{x})$ for any test point $\boldsymbol{x} \in \{P_{\text{test}}\}$.
**Generate:** Initial guesses for $\{\boldsymbol{\Phi}^\ell, \boldsymbol{G}^\ell\}_{\ell=1}^L$: $\boldsymbol{\Phi}^0$, $\boldsymbol{G}^{L+1} = \boldsymbol{1}\boldsymbol{1}^\top$.
**Draw** $\mathcal{S}$ samples for random fields at initialization $\xi_{\mu,s}^\ell(t) = \frac{1}{\sqrt{N}} W^\ell(0)\phi(h_{\mu,s}^{\ell-1})(t)$ and $\psi_{\mu,s}^\ell(t) = \frac{1}{\sqrt{N}} W^\ell(0) g_{\mu,s}^{\ell+1}(t)$
**while** *Kernels do not converge* $\forall \ell \in \{L\}, \forall s \in \{\mathcal{S}\}$ **do**
  Implement the non-Markovian dynamics of Eq. (13)
  Compute new $\{\boldsymbol{\Phi}^\ell, \boldsymbol{G}^\ell\}$

**Compute** $K_{\mu\nu}^{\text{aNTK}} = \lim_{t\to\infty} \sum_{\ell=0}^L G_{\mu\nu}^{\ell+1}(t,t) \Phi_{\mu\nu}^\ell(t,t)$
**return** $f_{aNTK}(\boldsymbol{x})$ *as in Eq. (11)*

---

[3]As specified in the Appendix E, we restrict the readout activations to those with $\sigma'(s_\mu) \neq 0$, otherwise, the gradient signal does not backpropagate through the network, preventing convergence to a kernel predictor.

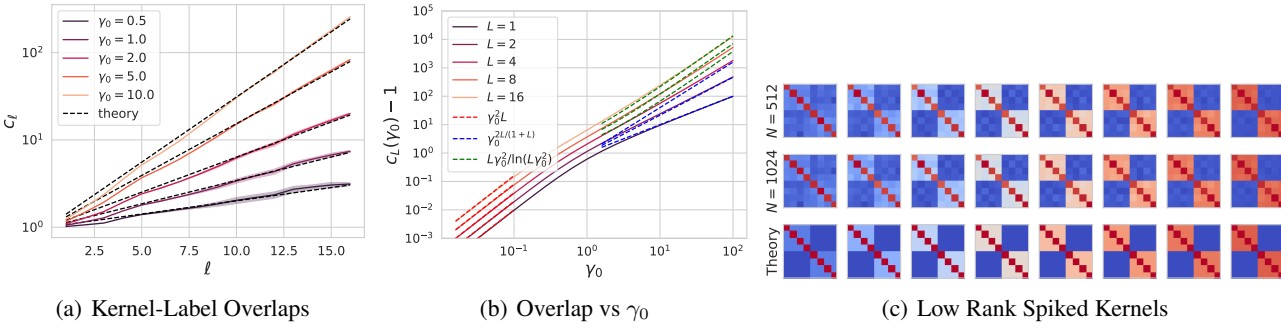

(a) Kernel-Label Overlaps        (b) Overlap vs $\gamma_0$        (c) Low Rank Spiked Kernels

*Figure 2.* Linear networks with whitened data are determined by a set of kernel-label overlap matrices. (a) The overlap variables $c_\ell$ increase exponentially with $\ell$ with rate that depends on $\gamma$. Solid lines taken from Langevin dynamics on $N = 1024$ network. (b) The alignment of the final layer $c_L$ as a function of $\gamma_0$ and $L$ exhibits three distinct scaling regimes. (c) Examples of learned kernels (at each layer $\ell$) in depth $\ell \in \{L = 8\}$, $\gamma_0 = 4.0$ and finite width $N$ networks compared to the $N \to \infty$ theory.

## 4. Infinitely-wide feature learning deep linear networks

Deep linear networks ($\phi(h) \equiv h$) provide a simpler framework for analysis than their nonlinear counterparts (Saxe et al., 2014; Advani & Saxe, 2017; Arora et al., 2019a; Aitchison, 2020; Li & Sompolinsky, 2021; Jacot et al., 2022; Zavatone-Veth et al., 2022a), yet they still converge to non-trivial feature aligned solutions. In deep linear networks, preactivations remain Gaussian at each layer when $P = \Theta_N(1)$ for the Central Limit Theorem (CLT), and this greatly simplifies the saddle point equations to algebraic formulas which close in terms of the kernels for both aNBK and aNTK theories. This means that we can solve for the adaptive kernels in both cases without any refined sampling strategy. The deep linear case of gradient flow dynamics can be found in (Bordelon & Pehlevan, 2022). Here, we report the solution to the saddle point equations that define the kernels in the feature-learning Bayesian setting, specializing to the regression problem (the generic loss case can be found in Appendix A)

$$\mathbf{\Phi}^\ell - \frac{\mathbf{\Phi}^{\ell-1}}{\lambda_{\ell-1}} \left( \mathbf{I} + \frac{\mathbf{\Phi}^{\ell-1}}{\lambda_{\ell-1}} \hat{\mathbf{\Phi}}^\ell \right)^{-1} = 0 \qquad \forall \ell = 1, \dots, L$$

$$\hat{\mathbf{\Phi}}^\ell - \frac{\hat{\mathbf{\Phi}}^{\ell+1}}{\lambda_\ell} \left( \mathbf{I} + \frac{\mathbf{\Phi}^\ell}{\lambda_\ell} \hat{\mathbf{\Phi}}^{\ell+1} \right)^{-1} = 0 \qquad \forall \ell = 1, \dots, L-1$$

$$\hat{\mathbf{\Phi}}^L + \frac{\gamma_0^2}{\lambda_L} \left( \frac{\mathbf{I}}{\beta} + \frac{\mathbf{\Phi}^L}{\lambda_L} \right)^{-1} \boldsymbol{y}\boldsymbol{y}^\top \left( \frac{\mathbf{I}}{\beta} + \frac{\mathbf{\Phi}^L}{\lambda_L} \right)^{-1} = 0.$$
(15)

Here, as a consistency check, it is easy to see that in the lazy limit $\gamma_0 \to 0$ the dual kernels $\hat{\mathbf{\Phi}}^\ell = 0 \quad \forall \ell \in \{L\}$, and as a consequence all the kernels $\mathbf{\Phi}^\ell$ will stay equal to the data covariance matrix $\mathbf{\Phi}^0$, consistent with lazy learning. However, for the rich regime where $\gamma_0 > 0$ the $\hat{\mathbf{\Phi}}^\ell$ kernels do not vanish and alter the fixed point kernels $\mathbf{\Phi}^\ell$ with target dependent information in the form of low rank spikes (by definition of $\hat{\mathbf{\Phi}}^L$).

To illustrate this spike effect in the learned kernels in the rich limit, we specialize to whitened input data $\boldsymbol{K}^x = \boldsymbol{I}$. We show in Appendix A.3 that the equations get simplified even further, since now the kernels $\mathbf{\Phi}^\ell$ only grows in the rank-one $\boldsymbol{y}\boldsymbol{y}^\top$ direction. By defining some set of scalar variables $\{c_\ell, \hat{c}_\ell\}_{\ell=1}^L$, which are the overlaps with the label direction $\boldsymbol{y}^\top \mathbf{\Phi}^\ell \boldsymbol{y} = c_\ell$ and $\boldsymbol{y}^\top \hat{\mathbf{\Phi}}^\ell \boldsymbol{y} = \hat{c}_\ell$ we find

$$c_\ell = \left( 1 + \frac{\gamma_0^2 c_L}{(\beta^{-1} + c_L)^2} \right)^\ell \qquad \forall \ell \in \{L\} \qquad (16)$$

which means that there is an exponential dependence of the overlap on the layer index $\ell$ (full derivation can be found in the Appendix) as Fig. 2(a) shows. From Eq. (16) we derive the scalings for lazy, large depth, and large feature strength limits. For the last layer overlap $c_L$ these are

$$\begin{aligned} c_L &\sim 1 + L\gamma_0^2 & \gamma_0^2 L \to 0 \\ c_L &\sim \gamma_0^{2L/(L+1)} & \gamma_0 \to \infty, L \text{ fixed} \\ c_L &\sim \frac{L\gamma_0^2}{\ln(L\gamma_0^2)} & L \to \infty, \gamma_0 \text{ fixed} \end{aligned} \qquad (17)$$

which closely match the theory in their respective regimes plotted in Fig. 2(b). In Fig. 2(c) we show examples of learned kernels for a $L = 8$ network and $\gamma_0 = 4.0$ matching the finite width $N = 1028$ network trained with Langevin dynamics.

## 5. Numerical Results

**Two-layer MLPs.** In Fig. 3(a) we compare test losses of lazy vs feature learning kernels for a two-layer MLP trained on a $P$ subset of two classes of CIFAR10 in a regression task. The *green* curve is the performance of NNGPK, *Orange* is the aNTK, *red* is the aNBK. There is a gap in performance between the lazy predictors and the adaptive feature learning predictors. However, when sample size $P$ is small, feature

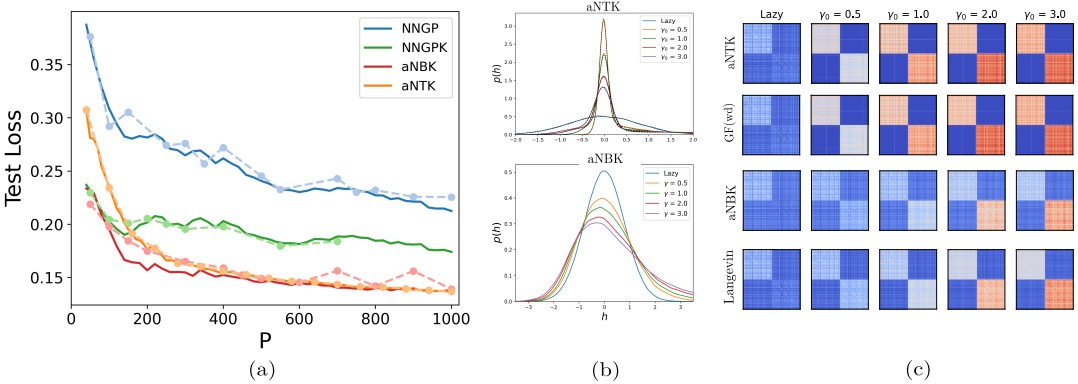

*Figure 3.* Feature learning theories outperform lazy predictors for a two-layer MLP trained with Squared Loss (SL) on two classes of CIFAR10 (airplane vs automobile). (a) Test losses as a function of sample size $P$. Solid lines refer to theories, dashed lines to numerical simulations on a $N = 5000$ network. *Blue* is the NNGP lazy predictor; *green* is the deterministic NNGPK kernel predictor; *orange* is aNTK with feature learning strength $\gamma_0 = 0.3$; *red* is aNBK predictor with the same $\gamma_0$. (b) Non-Gaussian pre-activation densities as a function of $\gamma_0$ for (*top*) aNTK and (*bottom*) aNBK. Black dashed lines are theoretical predictions. (c) Learned feature kernels of the adaptive theories closely match their relative finite width $N$ network trainings and evolve with $\gamma_0$.

learning in a data-limited scenario can let the model to overfit on test points and lazy learning can be beneficial in a small window of $P$.

In this plot, we also include the Neural Network Gaussian Process in *blue* (Neal, 1995; Lee et al., 2018; de G. Matthews et al., 2018), where for a number of patterns $P$ the solution space is sampled from the posterior of Eq. (9) by taking $\gamma = \Theta_N(1)$. Here the mean predictor is equivalent to the NNGPK predictor, however there is also a variance term, which comes from the fact that we are averaging over all possible random weights.

In the rich scenarios, we derive the preactivation distributions $p(\boldsymbol{h})$ as a function of $\gamma_0$ (Fig. 3(b) *top* and *bottom*) at convergence. At initialization, $p(\boldsymbol{h})$ follows $\mathcal{N}(0, \boldsymbol{\Phi}^0)$. However, as learning proceeds and features are learned, the densities accumulate non-Gaussian contributions which are identified by our theories (e.g. Eq. (7)). Fig. 3(c) shows that there is a clustering of $P = 100$ data points by category in the feature space defined by the adaptive kernels.

**Deep MLPs.** In Fig. 4 we show simulations of a Bayesian $L = 5$ Tanh MLP and compare to our infinite-width predictors. In order to study how feature learning propagates through depth, we plot the theoretical adaptive kernels of the aNBK theory versus the empirical kernels of Langevin dynamics at thermalization as a function of the layer index $\ell$. Coherently with the deep linear case in 4, the last layer feature kernel aligns first with the labels, since the clustering of features by class is more evident in this case. There is a small difference between theory and experiments because solving the min-max problem when $L > 1$ makes the convergence of 1 harder, given the dependency of the

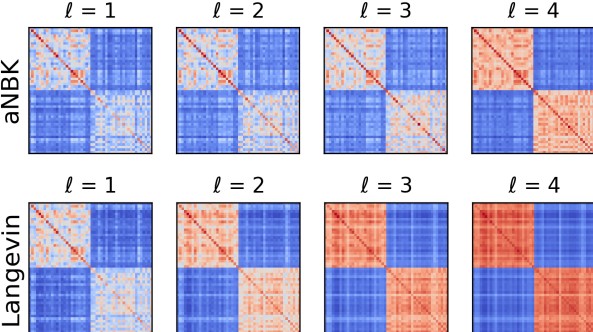

*Figure 4.* Theory vs empirical kernels at each layer for a 5HL MLP with $\phi(h) \equiv \tanh(h)$ learning $P = 50$ patterns of MNIST with $\boldsymbol{y} = \{\pm 1\}^P$ labels. Alignments $\mathcal{A}(\boldsymbol{\Phi}^\ell, \boldsymbol{\Phi}^\ell_{\exp}) = \frac{\mathrm{Tr}(\boldsymbol{\Phi}^\ell \boldsymbol{\Phi}^\ell_{\exp})}{\|\boldsymbol{\Phi}^\ell\|\|\boldsymbol{\Phi}^\ell_{\exp}\|}$ between theory and empirical kernels are $\mathcal{A}(\boldsymbol{\Phi}^1, \boldsymbol{\Phi}^1_{\exp}) = 97\%$, $\mathcal{A}(\boldsymbol{\Phi}^2, \boldsymbol{\Phi}^2_{\exp}) = 81\%$, $\mathcal{A}(\boldsymbol{\Phi}^3, \boldsymbol{\Phi}^3_{\exp}) = 73\%$ and $\mathcal{A}(\boldsymbol{\Phi}^4, \boldsymbol{\Phi}^4_{\exp}) = 89\%$.

non-Gaussian single-site density of Eq. (7) from the layer index $\ell$.

### 5.1. CNNs

DMFT for infinite-width CNNs under gradient-flow was previously derived in (Bordelon & Pehlevan, 2022), which we solve numerically here for the first time. In Fig. 5 we show comparisons of DMFT kernel predictors at convergence for a two-layer MLP and a two-layer CNN with kernel size $k = 8$ and stride 8. Black dashed curves, which are the theories, closely match the full colored lines, which are the network predictors on $N = 1028$ width networks. CNN outperform the MLP at large sample size $P$ for the same $\gamma_0$.

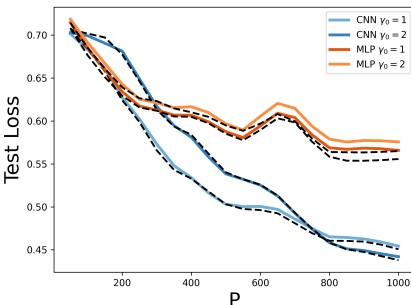

*Figure 5.* Test Loss as a function of sample size $P$ for DMFT theories at convergence: two-layer MLP vs two-layer CNN trained on $P$ animate/inanimate data on CIFAR10. Dashed lines are for theory, full-colored curves for empirical kernels.

See Appendix E.1 and Figure 5 for simulations of adaptive convolutional kernels derived from the feature-learning setting.

## 6. Discussion

In this paper, we develop a theory of non-parametric feature kernel predictors for MLP and CNN architectures in $\mu$P/mean-field parameterization. By analyzing gradient flow dynamics with weight decay and/or white noise, we identify two distinct infinite-width adaptive kernel predictors: aNBK, representing a Bayesian description of DNNs, and aNTK, corresponding to the fixed points of gradient flow with weight decay (Bordelon & Pehlevan, 2022). Unlike static NNGP (Neal, 1995; Lee et al., 2018) and NTK (Jacot et al., 2020) predictors, our kernels adapt to data. The feature learning strength, controlled by $\gamma_0$, recovers lazy training when $\gamma_0 \to 0$ and enables richer representations for $\gamma_0 > 0$ (Bordelon et al., 2024). We study their impact across architectures and benchmark tasks on real datasets.

We also analyze infinitely wide deep linear networks in the feature-learning regime, where the saddle point equations simplify. Assuming a white data covariance matrix, the order parameters reduce to scalar overlaps $c_\ell$ between kernels and labels at each layer $\ell$, exhibiting an exponential dependence on $\ell$. This implies that for fixed $\gamma_0$, deep linear networks align last-layer kernels first and propagate alignment backward. We derive scaling laws for these overlaps in the lazy, large-width, and large-depth regimes.

Our numerical results show that our adaptive kernels outperform NNGP and NTK at large sample size and match the performance of a trained NN in the feature-learning regime even in moderate widths (e.g. $N = 5000$). Our theory predicts non-Gaussian pre-activation densities at convergence and data-clustered feature kernels, whose alignment with label covariance increases with $\gamma_0$.

Future work could focus on reducing solver computational costs by developing more efficient optimization techniques.

## Acknowledgements

We thank Enrico M. Malatesta and Hugo Cui for useful and engaging discussions. B.B. is supported by a Google PhD Fellowship and NSF CAREER Award IIS-2239780. C.L. is supported by DARPA Award DIAL-FP-038, and The William F. Milton Fund from Harvard University. C.P. is supported by NSF grant DMS-2134157, NSF CAREER Award IIS-2239780, DARPA Award DIAL-FP-038, a Sloan Research Fellowship, and The William F. Milton Fund from Harvard University. This work has been made possible in part by a gift from the Chan Zuckerberg Initiative Foundation to establish the Kempner Institute for the Study of Natural and Artificial Intelligence.

## Impact Statement

This paper presents work whose goal is to advance the theoretical understanding of deep neural networks. There are many computational/algorithmic consequences of our work, none which we feel must be specifically highlighted here.

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

## A. Multi-layer deep Bayesian MLPs

As mentioned in the main text, we would like to study feature learning when the solution space is sampled from a posterior that is a Gibbs distribution with a likelihood $\mathcal{L}(\boldsymbol{\theta}; \mathcal{D})$ and a Gaussian prior $\frac{\lambda}{2}||\boldsymbol{\theta}||^2$ and by assuming width $N_\ell = N \, \forall \ell \in \{L\}$. We can do this computation for generic loss and non-linear activation functions as Eq. (2) shows. The representer theorem for Bayesian network is indeed independent on the activation choice. Here, the posterior takes the form

$$p(\boldsymbol{\theta}|\mathcal{D}) = \frac{1}{Z} \exp\left[-\beta\gamma^2 \mathcal{L}(\boldsymbol{\theta};\mathcal{D}) - \sum_{\ell=0}^{L} \frac{\lambda_\ell}{2}||\boldsymbol{\theta}^\ell||^2\right]. \tag{18}$$

being $\boldsymbol{\theta} = \text{Vec}\{\boldsymbol{W}^{(0)}, \ldots, \boldsymbol{w}^{(L)}\}$ the collection of weights, $\mathcal{D} = \{\boldsymbol{x}_\mu, y_\mu\}_{\mu=1}^{P}$ the dataset with $P$ patterns, and $\beta = \frac{1}{T}$ the temperature inverse. Again, we are interested in the infinitely overparameterized limit where $N \to \infty, P = \mathcal{O}_N(1)$. Here, when $\beta \to \infty$ the posterior becomes the uniform distribution over the set of global minimizers $\theta^\star \in \arg\min_{\boldsymbol{\theta}} \mathcal{L}(\boldsymbol{\theta})$. In this setting, one needs to rescale the loss function $\mathcal{L} \to \gamma^2\mathcal{L}$ with $\gamma = \gamma_0\sqrt{N}$ in order to avoid for the Gaussian prior to dominate over the likelihood when $N \to \infty$, suppressing any interaction with the learning task $\mathcal{D}$. From the normalization factor in Eq. (18), the partition function reads

$$Z = \int \prod_{\ell=0}^{L} d\boldsymbol{W}^\ell e^{-\frac{\beta}{2}\gamma_0^2 N \sum_\mu \mathcal{L}(y^\mu, f^\mu) - \sum_{\ell=0}^{L} \frac{\lambda_\ell}{2}||\boldsymbol{W}^\ell||^2} \tag{19}$$

and since we consider the dataset as fixed, we wish to integrate out the weights and move to a description in the space of representations. This can be done by simply enforcing the definitions of Eq. (20) through the integral representations of some Dirac-delta functions

$$\int \prod_{\mu,\ell} d\boldsymbol{h}_\mu^{\ell+1} ds_\mu \langle \prod_{\mu,\ell} \delta\left(\boldsymbol{h}_\mu^{\ell+1} - \frac{1}{\sqrt{N}}\boldsymbol{W}^\ell\phi(\boldsymbol{h}_\mu^\ell)\right) \prod_\mu \delta\left(s_\mu - \frac{1}{\gamma_0 N}\boldsymbol{w}^{(L)} \cdot \phi(\boldsymbol{h}_\mu^L)\right)\rangle_{\boldsymbol{\theta}\sim\mathcal{N}(0,\boldsymbol{\Lambda}^{-1})} \tag{20}$$

where $\boldsymbol{\Lambda} = \text{diag}(\lambda_0\boldsymbol{I}, \lambda_1\boldsymbol{I}, \ldots, \lambda_L\boldsymbol{I})$, getting

$$Z = \int \prod_\mu \prod_{\ell=0}^{L-1} \frac{d\boldsymbol{h}_\mu^{\ell+1} d\hat{\boldsymbol{h}}_\mu^{\ell+1}}{2\pi} \int \prod_\mu \frac{ds_\mu d\hat{s}_\mu}{2\pi(\gamma_0 N)^{-1}} e^{i\sum_{\mu,\ell} \boldsymbol{h}_\mu^{\ell+1}\cdot\hat{\boldsymbol{h}}_\mu^{\ell+1} - \frac{1}{2}\sum_{\mu,\nu}\sum_\ell(\hat{\boldsymbol{h}}_\mu^{\ell+1}\cdot\hat{\boldsymbol{h}}_\nu^{\ell+1})\left(\frac{\phi(\boldsymbol{h}_\mu^\ell)\cdot\phi(\boldsymbol{h}_\nu^\ell)}{N\lambda_\ell}\right)+i\gamma_0 N\sum_\mu s_\mu\hat{s}_\mu}$$
$$\times e^{-\frac{1}{2}\sum_{\mu,\nu}\hat{s}_\mu\hat{s}_\nu\left(\frac{\phi(\boldsymbol{h}_\mu^L)\cdot\phi(\boldsymbol{h}_\nu^L)}{\lambda_L}\right)-\frac{\beta}{2}N\gamma_0^2\sum_\mu\mathcal{L}(y^\mu,\sigma(s^\mu))}$$
$$= \int \prod_{\mu\nu}\prod_{\ell=1}^{L} \frac{d\Phi_{\mu\nu}^\ell d\hat{\Phi}_{\mu\nu}^\ell}{2\pi N^{-1}} \int \prod_\mu \frac{ds_\mu d\hat{s}_\mu}{2\pi(\gamma_0 N)^{-1}} e^{\frac{N}{2}\sum_{\mu\nu}\sum_\ell \Phi_{\mu\nu}^\ell\hat{\Phi}_{\mu\nu}^\ell - \gamma_0 N\sum_\mu s_\mu\hat{s}_\mu + \frac{N}{2}\hat{s}_\mu\frac{\Phi_{\mu\nu}^L}{\lambda_L}\hat{s}_\nu - \frac{\beta}{2}N\gamma_0^2\sum_\mu\mathcal{L}(y_\mu,\sigma(s_\mu))}$$
$$\times e^{N\sum_{\ell=0}^{L-1}\ln \mathcal{Z}[\Phi_{\mu\nu}^{\ell-1},\hat{\Phi}_{\mu\nu}^\ell]}. \tag{21}$$

In the last expression, we introduced the adaptive feature kernels as

$$\Phi_{\mu\nu}^\ell = \frac{1}{N}\phi(\boldsymbol{h}_\mu^\ell) \cdot \phi(\boldsymbol{h}_\nu^\ell) \tag{22}$$

and, again, we enforced their definitions in $Z$ with some conjugated variables $\hat{\Phi}_{\mu\nu}^\ell$. Both $\{\Phi_{\mu\nu}^\ell, \hat{\Phi}_{\mu\nu}^\ell\}$ will become deterministic quantities in the $N \to \infty$ limit.

The single-site density in Eq. (21) is given by

$$\mathcal{Z}_\ell = \int \prod_\mu \frac{dh_\mu d\hat{h}_\mu}{2\pi} e^{i\sum_\mu h_\mu\hat{h}_\mu - \frac{1}{2}\sum_{\mu\nu}\hat{h}_\mu\frac{\Phi_{\mu\nu}^{\ell-1}}{\lambda_{\ell-1}}\hat{h}_\nu - \frac{1}{2}\sum_{\mu\nu}\phi(h_\mu)\hat{\Phi}_{\mu\nu}^\ell\phi(h_\nu)}$$
$$= \int \frac{\prod_\mu dh_\mu}{\sqrt{2\pi \det\left(\frac{\Phi^{\ell-1}}{\lambda_{\ell-1}}\right)}} e^{-\frac{1}{2}\sum_{\mu\nu} h_\mu\left[\left(\frac{\Phi^{\ell-1}}{\lambda_{\ell-1}}\right)^{-1}\right]_{\mu\nu} h_\nu - \frac{1}{2}\sum_{\mu\nu}\phi(h_\mu)\hat{\Phi}_{\mu\nu}^\ell\phi(h_\nu)}. \tag{23}$$

At each layer, this decouples over the neuron index because we supposed the hidden layers having the same width dimension $N$ for $\ell = 1, \ldots, L$, and represents the normalization factor of a non-Gaussian pre-activation density distribution $p(h_\mu^\ell)$ where the non-Gaussian part is proportional to $\hat{\Phi}^\ell$, while the Gaussian contribution has a covariance that is the feature kernel at previous layer $\Phi^{\ell-1}$.

In the large $N \to \infty$ limit, we can collect the partition function of Eq. (21) in the compact form

$$Z = \int \prod_{\mu\nu} \prod_{\ell=1}^{L} \frac{d\Phi_{\mu\nu}^\ell d\hat{\Phi}_{\mu\nu}^\ell}{2\pi N^{-1}} \int \prod_\mu \frac{ds_\mu d\hat{s}_\mu}{2\pi(\gamma_0 N)^{-1}} e^{-NS(\{\Phi^\ell, \hat{\Phi}_{\ell=1}^L\}_{\ell=1}^L)} \tag{24}$$

being $S(\{\Phi^\ell, \hat{\Phi}_{\ell=1}^L\}_{\ell=1}^L)$ the intensive Bayesian action

$$S = -\frac{1}{2} \sum_{\mu\nu} \sum_\ell \Phi_{\mu\nu}^\ell \hat{\Phi}_{\mu\nu}^\ell + \gamma_0 \sum_\mu s_\mu \hat{s}_\mu - \frac{1}{2} \sum_{\mu\nu} \hat{s}_\mu \frac{\Phi_{\mu\nu}^L}{\lambda_L} \hat{s}_\nu + \gamma_0^2 \frac{\beta}{2} \sum_\mu \mathcal{L}(y_\mu, \sigma(s_\mu)) - \sum_\ell \ln \mathcal{Z}. \tag{25}$$

At infinite width $N$, this partition function is exponentially dominated by the saddle points of $S$. We thus identify the kernels that make $S$ locally stationary ($\delta S = 0$) by the equations

$$\frac{\partial S}{\partial \hat{\Phi}_{\mu\nu}^\ell} = 0 \quad \forall \ell = 1, \ldots, L \tag{26a}$$

$$\frac{\partial S}{\partial \Phi_{\mu\nu}^\ell} = 0 \quad \forall \ell = 1, \ldots, L-1 \tag{26b}$$

$$\frac{\partial S}{\partial \Phi_{\mu\nu}^L} = 0 \tag{26c}$$

$$\frac{\partial S}{\partial \hat{s}_\mu} = 0 \tag{26d}$$

$$\frac{\partial S}{\partial s_\mu} = 0 \tag{26e}$$

where the last two saddle points fix the output pre-activation given the loss function $\mathcal{L}$. If we explicitly write down the partial derivatives, we get

$$\Phi^\ell = \langle \phi(h^\ell)\phi(h^\ell)^\top \rangle \quad \forall \ell \in \{L\} \tag{27a}$$

$$\hat{\Phi}^\ell = (\Phi^\ell)^{-1} - \lambda_\ell (\Phi^\ell)^{-1} \langle h^{\ell+1} h^{\ell+1\top} \rangle (\Phi^\ell)^{-1} \quad \forall \ell \in \{L-1\} \tag{27b}$$

$$\hat{\Phi}^L = -\frac{1}{\lambda_L} \hat{s}\hat{s}^\top \tag{27c}$$

$$\hat{s} = -\beta\gamma_0 \frac{\partial \mathcal{L}}{\partial s} \tag{27d}$$

$$s = \frac{1}{\gamma_0 \lambda_L} \Phi^L \hat{s} \tag{27e}$$

from which we get a kernel predictor on a unseen test point $x$, since

$$f(x) = \sigma\left(\frac{\beta}{\lambda_L} \sum_{\mu=1}^P \Delta_\mu \Phi^L(x_\mu, x)\right) \tag{28}$$

being $\Delta_\nu = -\frac{\partial \mathcal{L}}{\partial s_\nu}$ the pattern error signal for that given loss.

### A.1. Regression problem

In this specific case, where the form of the loss function is known $\mathcal{L} = \sum_{\mu=1}^P (y_\mu - f_\mu)^2$ and the readout is linear, i.e. $f(s_\mu) = s_\mu \quad \forall \mu \in \{P\}$, we can integrate over the output pre-activations and its conjugated parameter $\{s_\mu, \hat{s}_\mu\}$. After

integrating, we obtain

$$
\begin{aligned}
Z &= \int \prod_{\mu\nu} \prod_{\ell=1}^{L} \frac{d\Phi_{\mu\nu}^{\ell} d\hat{\Phi}_{\mu\nu}^{\ell}}{2\pi N^{-1}} e^{-NS(\{\Phi_{\mu\nu}^{\ell}, \hat{\Phi}_{\mu\nu}^{\ell}\}_{\ell=1}^{L})} \\
&= \int \prod_{\mu\nu} \prod_{\ell=1}^{L} \frac{d\Phi_{\mu\nu}^{\ell} d\hat{\Phi}_{\mu\nu}^{\ell}}{2\pi N^{-1}} e^{-\frac{N}{2}\sum_{\ell} \operatorname{Tr}(\boldsymbol{\Phi}^{\ell}\hat{\boldsymbol{\Phi}}^{\ell}) + \frac{N}{2}\boldsymbol{y}^{\top}\left(\frac{\boldsymbol{I}}{\beta} + \frac{\boldsymbol{\Phi}^L}{\lambda_L}\right)^{-1}\boldsymbol{y} - N\sum_{\ell=0}^{L-1}\ln \mathcal{Z}[\boldsymbol{\Phi}^{\ell-1}, \hat{\boldsymbol{\Phi}}^{\ell}]}
\end{aligned}
\tag{29}
$$

where the important quantity to be extremized in the limit $N \to \infty$ is the intensive action

$$
S(\boldsymbol{\Phi}^{\ell}, \hat{\boldsymbol{\Phi}}^{\ell}) = -\frac{1}{2}\sum_{\ell} \operatorname{Tr}\left(\boldsymbol{\Phi}^{\ell}\hat{\boldsymbol{\Phi}}^{\ell}\right) + \frac{\gamma_0^2}{2}\boldsymbol{y}^{\top}\left(\frac{\boldsymbol{I}}{\beta} + \frac{\boldsymbol{\Phi}^L}{\lambda_L}\right)^{-1}\boldsymbol{y} - \sum_{\ell=1}^{L-1}\ln \mathcal{Z}[\boldsymbol{\Phi}^{\ell-1}, \hat{\boldsymbol{\Phi}}^{\ell}].
\tag{30}
$$

Here, the saddle points which render the action $S$ locally stationary $\delta S = 0$ with respect to these $2L$ matrix order parameters can be collected as

$$
\boldsymbol{\Phi}^{\ell} = \langle \phi(\boldsymbol{h}^{\ell})\phi(\boldsymbol{h}^{\ell})^{\top}\rangle \quad \forall \ell = 1, \dots, L
\tag{31a}
$$

$$
\hat{\boldsymbol{\Phi}}^{\ell} = (\boldsymbol{\Phi}^{\ell})^{-1} - \lambda_{\ell}(\boldsymbol{\Phi}^{\ell})^{-1}\langle \boldsymbol{h}^{\ell+1}\boldsymbol{h}^{\ell+1\top}\rangle(\boldsymbol{\Phi}^{\ell})^{-1} \quad \forall \ell = 1, \dots, L-1
\tag{31b}
$$

$$
\hat{\boldsymbol{\Phi}}^{L} = -\frac{\gamma_0^2}{\lambda_L}\left(\frac{\boldsymbol{I}}{\beta} + \frac{\boldsymbol{\Phi}^L}{\lambda_L}\right)^{-1}\boldsymbol{y}\boldsymbol{y}^{\top}\left(\frac{\boldsymbol{I}}{\beta} + \frac{\boldsymbol{\Phi}^L}{\lambda_L}\right)^{-1}
\tag{31c}
$$

being $\Phi_{\mu\nu}^{0} = \frac{\boldsymbol{x}_{\mu}\cdot\boldsymbol{x}_{\nu}}{\lambda_0 D}$ the data matrix covariance in this notation.

Notice that the last layer dual's kernel $\hat{\Phi}_{\mu\nu}^{L}$ vanishes in the lazy limit $\gamma_0 \to 0$ and so do all the dual kernels at previous layers $\hat{\Phi}_{\mu\nu}^{\ell} = 0$, while for non-negligible $\gamma_0$ we see that the each hidden layer features are non-Gaussian from Eq. (23). Details on the numerical implementation of the numerical solver for Eqs. (23) can be found in Sec. G.2.

In general, solving the min-max problem for deep networks is hard, because solving $\max_{\boldsymbol{\Phi}} S$ ($S$ is the action defined in Algorithm 1) means to find some values for the dual kernels $\{\hat{\boldsymbol{\Phi}}^{\ell}\}_{\ell=1}^{L}$ which tilt the non-Gaussian measure in Eq. (7) at each layer $\ell$. In Fig. 4 of the main text we solved the theory for Algorithm 1 by initializing the theoretical kernels $\{\boldsymbol{\Phi}, \hat{\boldsymbol{\Phi}}\}_{\ell=1}^{L}$ with a lazy guess as explained in Sec. G.2. However, an easier strategy for convergence is to initialize the solver Algorithm 1 with the empirical NN kernels $\{\boldsymbol{\Phi}^{\ell}\}_{\ell=1}^{L}$ at convergence obtained from Langevin simulations and perturbed with a multiplicative Gaussian noise. In this way, one only needs to solve for the dual variables. This warm start allows faster convergence. This is what we did to produce Fig. 7. Here, we plot the kernel alignment $\mathcal{A}(\boldsymbol{\Phi}^{\ell}, \boldsymbol{y}\boldsymbol{y}^{\top}) = \frac{\boldsymbol{y}^{\top}\boldsymbol{\Phi}^{\ell}\boldsymbol{y}}{\|\boldsymbol{y}\boldsymbol{y}^{\top}\|\|\boldsymbol{\Phi}^{\ell}\|}$, which is the cosine similarity between the kernel $\boldsymbol{\Phi}^{\ell}$ and the label covariance $\boldsymbol{y}\boldsymbol{y}^{\top}$ at each layer. We show that by increasing feature strength $\gamma_0$, the alignments increase. As expected, the last layer kernel aligns first with the labels as $\gamma_0$ increases, followed by the previous ones. Fig. 7(b) shows that the perturbed kernels of the theory retrieve the final NN kernels as locally stable fixed points.

In Fig. 6 we show that the non Gaussian preactivation distribution, depending on the activation function chosen, can develop a multimodal profile, and so a Mexican-hat profile for high values of feature learning strength $\gamma_0$.

## A.2. Generalization error

Knowing the posterior distribution makes it easy to compute the test error on a new (unseen) example $(\boldsymbol{x}_0, y_0)$, which is defined as

$$
\epsilon_g(\boldsymbol{x}_0, y_0) = \langle (y_0 - f_0(\boldsymbol{x}_0; \theta))^2\rangle_{\boldsymbol{\theta}\sim p(\boldsymbol{\theta}|\mathcal{D})}
\tag{32}
$$

where the sampling measure corresponds to Eq. (18). Here, we can include with an index $\mu = 0$ the test pattern contribution, and just compute

$$
\epsilon_g(\boldsymbol{x}_0, y_0) = \frac{1}{Z}\int \prod_{\mu\nu=0}^{P}\prod_{\ell=1}^{L} \frac{d\tilde{\Phi}_{\mu\nu}^{\ell} d\hat{\tilde{\Phi}}_{\mu\nu}^{\ell}}{2\pi N^{-1}} \int \prod_{\mu=0}^{P} \frac{ds_{\mu} d\hat{s}_{\mu}}{2\pi(\gamma_0 N)^{-1}} e^{\sum_{\mu\nu=0}^{P}\sum_{\ell=1}^{L} N_{\ell}\tilde{\Phi}_{\mu\nu}^{\ell}\hat{\tilde{\Phi}}_{\mu\nu}^{\ell} + \frac{N_L}{2\lambda_L}\sum_{\mu,\nu=1}^{P}\Phi_{\mu\nu}^{L}\hat{s}_{\mu}\hat{s}_{\nu} + \frac{N_L}{2\lambda_L}\Phi_{00}^{L}(\hat{s}_0)^2} \times
$$

$$
\times e^{\frac{N_L}{\lambda_L}\hat{s}_0\sum_{\mu}\Phi_{\mu}\hat{s}_{\mu} - N_L\sum_{\mu=0}^{P}s_{\mu}\hat{s}_{\mu} - \frac{\beta}{2}N\sum_{\mu=1}^{P}(y^{\mu}-s^{\mu})^2 + N\sum_{\ell=0}^{L}\ln \tilde{\mathcal{Z}}_{\ell}} \times (y_0 - s_0)^2
\tag{33}
$$

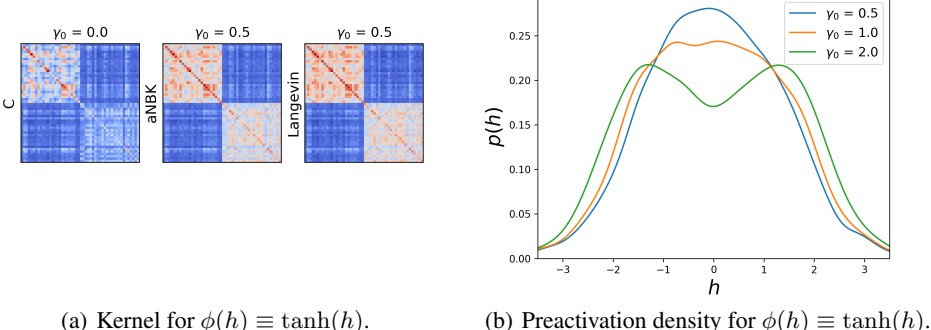

(a) Kernel for $\phi(h) \equiv \tanh(h)$.

(b) Preactivation density for $\phi(h) \equiv \tanh(h)$.

*Figure 6.* (a) Bayesian theory (aNBK) and empirical (Langevin) adaptive kernels with feature learning strength $\gamma_0$ and for $P = 50$ patterns of $0/1$ classes of MNIST. $C$ is the Gram matrix of data. (b) The preactivation distribution is in general non-Gaussian at each value of feature strength $\gamma_0$. For activation $\phi(h) \equiv \tanh(h)$ we show that $p(h)$ can develop a Mexican-hat profile.

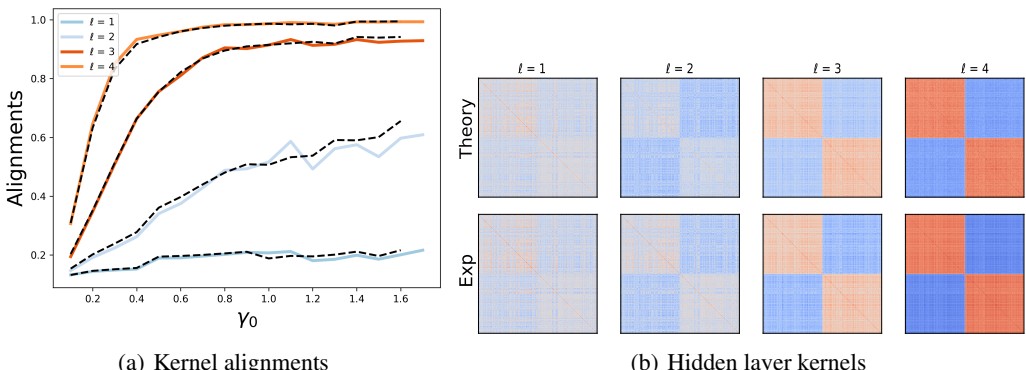

(a) Kernel alignments

(b) Hidden layer kernels

*Figure 7.* Bayesian $L = 5$ ReLU MLP trained on $P = 1000$ data of CIFAR10. Colored curve are experiments, dashed lines are the predictors calculated from Algorithm 1. (a) Kernel alignments $\mathcal{A}(\mathbf{\Phi}^\ell, \boldsymbol{yy}^\top)$ at each layer vs feature strength. (b) Theory vs empirical kernels at each layer. Here, theory refers to the intermediate layer kernels calculated via Algorithm 1.

where the single-site action contains now all the possible interactions with the test point at each layer $\ell$ in the test-test kernel $\Phi_{00}^\ell = \langle \phi(h_0)^2 \rangle$ and the test-train kernel $\Phi_\mu^\ell = \langle \phi(h_0)\phi(h_\mu) \rangle$

$$\tilde{\mathcal{Z}} = \int \prod_{\mu=0}^{P} \frac{dh_\mu d\hat{h}_\mu}{2\pi} e^{i\sum_{\mu=0}^{P} h_\mu \hat{h}_\mu - \frac{1}{2}\sum_{\mu\nu=1}^{P} \hat{h}_\mu \frac{\Phi_{\mu\nu}^{\ell-1}}{\lambda_{\ell-1}} \hat{h}_\nu - \sum_{\mu\nu=1}^{P} \phi(h_\mu)\hat{\Phi}_{\mu\nu}^\ell \phi(h_\nu) - \frac{1}{2}\frac{\Phi_{00}^{\ell-1}}{\lambda_{\ell-1}}\hat{h}_0^2 - \hat{\Phi}_{00}^\ell \phi(h_0)^2 - \sum_{\mu=1}^{P} \frac{\Phi_\mu^{\ell-1}}{\lambda_{\ell-1}}\hat{h}_0\hat{h}_\mu}$$

$$\times e^{-\sum_{\mu=1}^{P} \hat{\Phi}_\mu^\ell \phi(h_0)\phi(h_\mu)}. \tag{34}$$

Exploiting the saddle point equations, we realize that the dual kernels concerning the test point $\hat{\Phi}^\ell_\mu, \hat{\Phi}_{00} = 0$, and that

$$\Phi^\ell_{\mu\nu} = \langle \phi(h_\mu)\phi(h_\nu) \rangle \quad \forall \ell = 1, \dots, L \tag{35a}$$

$$\hat{\Phi}^\ell_{\mu\nu} = \frac{1}{2\lambda_\ell}\langle \hat{h}^{\ell+1}_\mu \hat{h}^{\ell+1}_\nu \rangle \quad \forall \ell = 1, \dots, L-1 \tag{35b}$$

$$\hat{\Phi}^L_{\mu\nu} + \frac{1}{2\lambda_L}\hat{s}_\mu \hat{s}_\nu = 0 \tag{35c}$$

$$\hat{\Phi}^\ell_\mu = \frac{1}{\lambda_\ell}\langle \hat{h}^{\ell+1}_0 \hat{h}^{\ell+1}_\mu \rangle \quad \forall \ell = 1, \dots, L-1 \tag{35d}$$

$$\hat{s}_0 = \hat{\Phi}^\ell_\mu = \hat{\Phi}_{00} = 0 \tag{35e}$$

$$s_0 = \frac{1}{\lambda_L}\sum_\mu \Phi^L_\mu \hat{s}_\mu \tag{35f}$$

$$\hat{s}_\mu - \beta\Big(y_\mu - s_\mu\Big) = 0 \tag{35g}$$

$$s_\mu = \frac{1}{\lambda_L}\sum_\nu \Phi^L_{\mu\nu}\hat{s}_\nu. \tag{35h}$$

This allows to rewrite the single site density in a much simpler form, where the non-Gaussian contribution includes just the train points, while the Gaussian part has a $(P+1) \times (P+1)$ covariance matrix $\tilde{\Phi} = \begin{pmatrix} \Phi_{00} & \Phi^\top_\mu \\ \Phi_\mu & \Phi_{\mu\nu} \end{pmatrix}$

$$\tilde{\mathcal{Z}} = \int \frac{\prod_\mu dh_\mu}{\sqrt{2\pi \det\left(\frac{\tilde{\Phi}^{\ell-1}}{\lambda_{\ell-1}}\right)}} e^{-\frac{\lambda_{\ell-1}}{2}\sum^P_{\mu\nu=0} h_\mu \left(\tilde{\Phi}^{\ell-1}\right)^{-1}_{\mu\nu} h_\nu - \frac{1}{2}\sum^P_{\mu\nu=1}\phi(h_\mu)\hat{\Phi}^\ell_{\mu\nu}\phi(h_\nu)} \tag{36}$$

This means that once we solved for Eq. (31) we can marginalize Eq. (36) to get $p(h^\ell_0|\boldsymbol{h}^\ell)$ and hence the test-train vector kernel $\boldsymbol{\Phi}_\mu$. The test error expression is

$$\epsilon_g(\boldsymbol{x}_0, y_0) = \left(y_0 - \frac{1}{\lambda_L}\boldsymbol{\Phi}^{L\top}\left[\frac{\boldsymbol{\Phi}^L}{\lambda_L} + \frac{\boldsymbol{I}}{\beta}\right]^{-1}\boldsymbol{y}\right)^2 \tag{37}$$

meaning that the predictor $f(x) = \frac{1}{\lambda_L}\boldsymbol{\Phi}^L(x)^\top\left[\frac{\boldsymbol{\Phi}^L}{\lambda_L} + \frac{\boldsymbol{I}}{\beta}\right]^{-1}\boldsymbol{y}$ is again a kernel predictor, with adaptive kernels from Eqs. (31).

An alternative way to obtain the predictor in the specific regression setting from Eq. (28) and when $\sigma(s) = s$ is by solving for $\Delta_\mu \equiv -\frac{\partial L}{\partial s_\mu}$. In this case

$$\Delta_\mu = y_\mu - f_\mu = y_\mu - \frac{\beta}{\lambda}\sum_\nu \Delta_\nu \Phi_{\mu\nu} \tag{38}$$

and by solving this equation for $\boldsymbol{\Delta} \in \mathbb{R}^P$ we obtain

$$\boldsymbol{\Delta} = \left[\boldsymbol{I} + \frac{\beta}{\lambda}\boldsymbol{\Phi}\right]^{-1}\boldsymbol{y}. \tag{39}$$

Lastly, combining this with the original Eq. (28), we get

$$f(x) = \boldsymbol{\Phi}(x)^\top\left[\boldsymbol{I} + \frac{\beta}{\lambda}\boldsymbol{\Phi}\right]^{-1}\boldsymbol{y} \tag{40}$$

as is it in the main text Eq. (5).

### A.3. Deep linear case

In the deep linear case $\phi(h^\ell) = h^\ell$, the action of Eq. (30) gets simplified because the single-site density is now Gaussian, and this leads to

$$S(\{\boldsymbol{\Phi}^\ell, \hat{\boldsymbol{\Phi}}^\ell\}) = -\frac{1}{2}\sum_\ell \mathrm{Tr}\left(\boldsymbol{\Phi}^\ell \hat{\boldsymbol{\Phi}}^\ell\right) + \frac{\gamma_0^2}{2}\boldsymbol{y}^\top\left(\frac{\boldsymbol{I}}{\beta} + \frac{\boldsymbol{\Phi}^L}{\lambda_L}\right)^{-1}\boldsymbol{y} + \frac{1}{2}\sum_{\ell=1}^L \ln\det\left(\boldsymbol{I} + \frac{\boldsymbol{\Phi}^{\ell-1}}{\lambda_{\ell-1}}\hat{\boldsymbol{\Phi}}^\ell\right) \tag{41}$$

with the following saddle point equations

$$\boldsymbol{\Phi}^\ell - \frac{\boldsymbol{\Phi}^{\ell-1}}{\lambda_{\ell-1}}\left(\boldsymbol{I} + \frac{\boldsymbol{\Phi}^{\ell-1}}{\lambda_{\ell-1}}\hat{\boldsymbol{\Phi}}^\ell\right)^{-1} = 0 \qquad \forall \ell = 1, \ldots, L \tag{42a}$$

$$\hat{\boldsymbol{\Phi}}^\ell - \frac{\hat{\boldsymbol{\Phi}}^{\ell+1}}{\lambda_\ell}\left(\boldsymbol{I} + \frac{\boldsymbol{\Phi}^\ell}{\lambda_\ell}\hat{\boldsymbol{\Phi}}^{\ell+1}\right)^{-1} = 0 \qquad \forall \ell = 1, \ldots, L-1 \tag{42b}$$

$$\hat{\boldsymbol{\Phi}}^L + \frac{\gamma_0^2}{\lambda_L}\left(\frac{\boldsymbol{I}}{\beta} + \frac{\boldsymbol{\Phi}^L}{\lambda_L}\right)^{-1}\boldsymbol{y}\boldsymbol{y}^\top\left(\frac{\boldsymbol{I}}{\beta} + \frac{\boldsymbol{\Phi}^L}{\lambda_L}\right)^{-1} = 0. \tag{42c}$$

In principle, this is a closed set of equations, which can be iteratively solved as mentioned in Sec. 4. If we choose input data that are whitened, with $\boldsymbol{\Phi}^0 = \boldsymbol{K}^x = \boldsymbol{I}$ and label norm $|\boldsymbol{y}| = 1$, the equations can be simplified even further, since feature kernels at each layer $\boldsymbol{\Phi}^\ell$ only evolve in the rank-one direction $\boldsymbol{y}\boldsymbol{y}^\top$. This allows to define the variables $\{c_\ell, \hat{c}_\ell\}$ which are the overlaps with the label direction $\boldsymbol{y}$

$$\boldsymbol{y}^\top \boldsymbol{\Phi}^\ell \boldsymbol{y} \equiv c^\ell , \ \boldsymbol{y}^\top \hat{\boldsymbol{\Phi}}^\ell \boldsymbol{y} \equiv \hat{c}^\ell. \tag{43}$$

In this setting, the reduced saddle point equations become

$$c_1 = \frac{1}{1 + \hat{c}_1} , \ c_{\ell+1} = \frac{c_\ell}{1 + c_\ell \, \hat{c}_{\ell+1}} \tag{44}$$

$$\hat{c}_L = -\frac{\gamma^2}{(\beta^{-1} + c_L)^2} , \ \hat{c}_\ell = \frac{\hat{c}_{\ell+1}}{1 + c_\ell \, \hat{c}_{\ell+1}} \tag{45}$$

which are $2L$ scalar equations for the overlaps $c_\ell$ at each layer. We note the following conservation

$$c_\ell \hat{c}_\ell = c_{\ell+1}\hat{c}_{\ell+1} = -\frac{\gamma^2 c_L}{(\beta^{-1} + c_L)^2} \equiv \chi(c_L) \tag{46}$$

which implies that

$$1 = \frac{c_\ell}{c_{\ell+1} + c_\ell \chi(c_L)} \implies c_{\ell+1} = c_\ell \left(1 - \chi(c_L)\right) = \left(1 - \chi(c_L)\right)^\ell c_1. \tag{47}$$

Since we have $c_1 = \frac{1}{1 + \chi/c_1} \implies c_1 = 1 - \chi(c_L)$, hence we find

$$c_\ell = \left(1 - \chi(c_L)\right)^\ell \tag{48}$$

which means that here is an exponential dependence of the overlap on layer index. In practice, we can solve Eq. (48) for the last layer overlap $c_L$ since $\chi(c_L)$ and then move backward in computing all the previous layer overlaps.

We can also extract the following small $\gamma$ or small $L$ asymptotics. Precisely, when $L$ is fixed and $\gamma_0 \to 0$ we recover a perturbative feature learning regime

$$c_L \sim 1 + \frac{L\gamma^2}{c_L} \implies c_L \sim 1 + L\gamma^2 , \ \gamma^2 L \to 0 \tag{49}$$

where correction are $\mathcal{O}(\gamma^2 L)$. Similarly, at large $\gamma$ with fixed $L$, which stands for a shallow but very rich regime, the overlaps scale as

$$c = \left[1 + \gamma^2 c^{-1}\right]^L \implies c \sim \gamma^{2L/(L+1)} \tag{50}$$

while, alternatively the large $L$ asymptotics for a very deep network have the form

$$c^{1+1/L} = c + \gamma^2 \implies c\ln c \sim L\gamma^2 \implies c \sim \frac{L\gamma^2}{\ln(L\gamma^2)} \tag{51}$$

which we show to be predictive in Fig.3 of the main text.

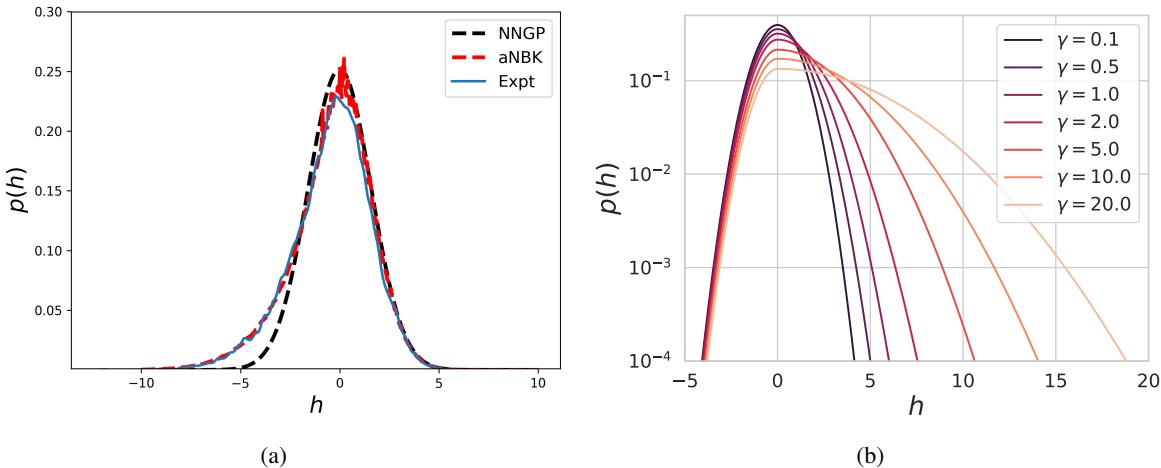

(a)             (b)

*Figure 8.* (a) Bayesian two-layer MLP trained on a whitened covariance matrix $\mathbf{\Phi}^0 = \boldsymbol{I}$ on $P = 4$ train points. Feature learning ($\gamma_0 > 0$) leads to a non-Gaussian pre-activation distributions. Black-dashed curve is the lazy NNGP when $\boldsymbol{h} \sim \mathcal{N}(0, \mathbf{\Phi}^0)$; *red* curve is the aNBK theory when $\boldsymbol{h}$ is sampled from Eq. 7; *blue* is the empirical pre-activation distribution of a $N = 5000$ network trained in the rich regime. (b) Non-gaussian pre-activation distribution as a function of feature learning strength $\gamma$ for a Bayesian 2-layer MLP trained with Squared Error (SE) on 0-1 classes of MNIST dataset. Here sample size $P = 100$.

### A.4. Non-gaussian pre-activation density

In the non-linear case, as Eq. (23) shows, when $\gamma_0 > 0$ the pre-activation density at each layer is non-Gaussian, with $\gamma_0$ entering in the saddle point equation for the dual kernel at the last layer $\hat{\mathbf{\Phi}}^L$ (see Eq. (31)). This means that, once we have $\{\mathbf{\Phi}^\ell, \hat{\mathbf{\Phi}}^\ell\}_{\ell=1}^L$ from the solver (Alg 1), we can evaluate Eq. (23) with importance sampling and compute $p(h_\mu)$ for a given pattern $\mu$. In Fig. A.4(a) we show that, while in the lazy regime where no feature learning enters, the hidden layer pre-activation of the NNGP predictors are Gaussian, this is not the case in our setting.

### A.5. Perturbative approximation

In the $\gamma_0 \to 0$ limit, we recover the static kernels of NNGP predictor (Neal, 1995; Lee et al., 2018). Corrections to this lazy limit can be extracted at small but finite $\gamma_0$. In order to do so, we can expand each macroscopic variable $q(\gamma_0)$ in power series of $\gamma_0$, such as $q = q^{(0)} + \gamma_0^2 q^{(1)} + \gamma_0^4 q^{(2)} + \ldots$, and compute the corrections up to $\mathcal{O}(\gamma_0^2)$. First of all, we notice that at leading order in $\gamma_0^2$

$$\hat{\mathbf{\Phi}}^L = -\gamma_0^2 \left(\mathbf{\Phi}_0^L + \beta^{-1}\right)^{-1} \boldsymbol{y}\boldsymbol{y}^\top \left(\mathbf{\Phi}_0^L + \beta^{-1}\right)^{-1} \tag{52}$$

where we set each $\lambda_\ell = 1$ for clarity of notation. For each dual kernel at previous layer $\ell = 1, \ldots, L-1$ we have instead a recursion

$$\frac{1}{2}\hat{\mathbf{\Phi}}^\ell = -\frac{\partial}{\partial \mathbf{\Phi}^\ell} \ln \mathcal{Z}(\mathbf{\Phi}^\ell, \hat{\mathbf{\Phi}}^{\ell+1}) \tag{53}$$

where non-perturbatively

$$\frac{1}{2}\hat{\mathbf{\Phi}}^\ell = -\frac{1}{\mathcal{Z}} \int dh \frac{\partial}{\partial \mathbf{\Phi}} \exp\left(-\frac{1}{2}h(\mathbf{\Phi}^\ell)^{-1}h - \frac{1}{2}\phi(h)\hat{\mathbf{\Phi}}^{\ell+1}\phi(h)\right) \tag{54}$$

$$= \frac{1}{2}\frac{1}{\left\langle \exp\left(-\frac{1}{2}\phi(h)\hat{\mathbf{\Phi}}^{\ell+1}\phi(h)\right)\right\rangle_0} \times \left\langle \frac{\partial^2}{\partial \boldsymbol{h}^2} \exp\left(-\frac{1}{2}\phi(h)\hat{\mathbf{\Phi}}^{\ell+1}\phi(h)\right)\right\rangle_0 \tag{55}$$

being $\langle\rangle_0$ the Gaussian average with covariance $\Phi^\ell$. With a little bit of algebra the numerator can be written as

$$\left\langle \frac{\partial^2}{\partial h_\mu \partial h_\nu} \exp\left(-\frac{1}{2}\phi(h)\hat{\Phi}\phi(h)\right)\right\rangle_0 = -\left\langle \frac{\partial}{\partial h_\nu}\left[\exp\left(-\frac{1}{2}\phi(h)\hat{\Phi}\phi(h)\right)\dot{\phi}(h_\mu)\hat{\Phi}_{\mu\alpha}\phi(h_\alpha)\right]\right\rangle_0$$

$$= \sum_{\alpha\beta}\hat{\Phi}_{\mu\alpha}\hat{\Phi}_{\nu\beta}\left\langle \dot{\phi}(h_\mu)\dot{\phi}(h_\nu)\phi(h_\alpha)\phi(h_\beta)\exp\left(-\frac{1}{2}\phi(h)\hat{\Phi}\phi(h)\right)\right\rangle_0$$

$$- \delta_{\mu\nu}\sum_\alpha \hat{\Phi}_{\mu\alpha}\left\langle \ddot{\phi}(h_\mu)\phi(h_\alpha)\exp\left(-\frac{1}{2}\phi(h)\hat{\Phi}\phi(h)\right)\right\rangle_0$$

$$- \left\langle \dot{\phi}(h_\mu)\dot{\phi}(h_\alpha)\exp\left(-\frac{1}{2}\phi(h)\hat{\Phi}\phi(h)\right)\right\rangle_0 \hat{\Phi}_{\mu\alpha}. \tag{56}$$

Under the leading order approximation, we find the following relationship between successive layers

$$\hat{\boldsymbol{\Phi}}^\ell \sim \frac{1}{2}\left\langle \frac{\partial^2}{\partial \boldsymbol{h}\partial \boldsymbol{h}^\top}\phi(\boldsymbol{h})^\top \hat{\boldsymbol{\Phi}}^{\ell+1}\phi(\boldsymbol{h})\right\rangle \tag{57}$$

The entries of this Hessian matrix can be computed in terms of derivatives of the activation function

$$\frac{\partial}{\partial h_\mu}\frac{\partial}{\partial h_\nu}\sum_{\alpha\beta}\phi(h_\alpha)\phi(h_\beta)\hat{\Phi} = \frac{\partial}{\partial h_\mu}\left[\dot{\phi}(h_\nu)\phi(h_\beta)\hat{\Phi}_{\nu\beta} + \dot{\phi}(h_\nu)\phi(h_\alpha)\hat{\Phi}_{\nu\alpha}\right] \tag{58}$$

$$= 2\left\langle \dot{\phi}(h_\mu)\dot{\phi}(h_\nu)\right\rangle \hat{\Phi}_{\mu\nu} + 2\delta_{\mu\nu}\left\langle \ddot{\phi}(h_\mu)\sum_\beta \phi(h_\beta)\right\rangle \hat{\Phi}_{\mu\beta} \tag{59}$$

all of these Gaussians can be evaluated at the unperturbed NNGP kernels Gaussian densities. Once these $\hat{\boldsymbol{\Phi}}^\ell$ matrices have been computed, the $\boldsymbol{\Phi}^\ell$ matrices can be asymptotically approximated as

$$\boldsymbol{\Phi}^\ell \sim \boldsymbol{\Phi}_0^\ell - \frac{1}{2}\left\langle \phi(\boldsymbol{h})\phi(\boldsymbol{h})^\top\left(\phi(\boldsymbol{h})^\top\hat{\boldsymbol{\Phi}}^\ell\phi(\boldsymbol{h})\right)\right\rangle_0 - \boldsymbol{\Phi}_0^\ell\left\langle \phi(\boldsymbol{h})^\top\hat{\boldsymbol{\Phi}}^\ell\phi(\boldsymbol{h})\right\rangle_0 \tag{60}$$

which agree with the perturbative treatment of (Zavatone-Veth et al., 2022a) obtained in the NTK parameterization which is similar to our small $\gamma_0$ expansion (neglecting finite width fluctuations).

## B. Expanded Related Works

In this section we provide additional details comparing our contributions with previous works. We break these up into (1) scale-renormalization theories (2) Adaptive Kernel Theories in the NTK scaling (3) and rescaling the likelihood.

### B.1. Scale Renormalization Theories

Many theories of proportional limits of Bayesian neural networks predict that the mean predictor has the form

$$\langle f(x)\rangle = Q(\alpha)\sum_{\mu\nu}\Phi^L(x,x_\mu)\left[Q(\alpha)\boldsymbol{\Phi}^L + \beta^{-1}\boldsymbol{I}\right]_{\mu\nu}^{-1}y_\nu = \sum_{\mu\nu}\Phi^L(x,x_\mu)\left[\boldsymbol{\Phi}^L + \beta^{-1}Q(\alpha)^{-1}\boldsymbol{I}\right]_{\mu\nu}^{-1}y_\nu, \tag{61}$$

where $Q(\alpha)\in\mathbb{R}$ is a scalar that is determined self consistently as a function of $\alpha = P/N$ and $\Phi^L(x,x')$ is the final layer's NNGP kernel under the prior (Li & Sompolinsky, 2021; Pacelli et al., 2023). We rewrote this expression in the last line to emphasize that this is equivalent to a data dependent ridge with the original NNGP kernel. In the limit of low temperature $\beta\to\infty$ (zero regularization), these effects have no impact on the mean predictor. This theory does not capture adaptation in the entries of the kernel.

Extensions of this theory to CNNs, gated linear networks, and transfer learning result in slightly more order parameters which compute gate $\times$ gate overlaps for gated linear networks (Li & Sompolinsky, 2022), or space $\times$ space overlaps for CNNs (Aiudi et al., 2023), or task $\times$ task overlaps for transfer/continual learning (Shan et al., 2024; Ingrosso et al., 2025). While these theories are slightly more flexible, they do not allow for adaptation for the kernels to adapt within each block (gate, spatial location, task) but just reweight the blocks.

## B.2. Adaptive Kernel Theories in NTK Scaling

Both Seroussi et al. (2023) and Fischer et al. (2024) derive mean field actions for the kernels and conjugate kernels for Bayesian networks under the posterior. Seroussi et al. (2023) use a variational Gaussian approximation to the preactivation density which finds the best Gaussian density which approximates the true posterior density for $h$. Fischer et al, explicitly describe how the joint distribution of kernels obeys a large deviation principle at large $N$ (or in the proportional limit $P, N \to \infty$ with $P/N = \alpha$)

$$\frac{1}{N} \ln p(\{\boldsymbol{\Phi}^{\ell}\}) \sim - \max_{\{\hat{\boldsymbol{\Phi}}^{\ell}\}} S(\{\boldsymbol{\Phi}^{\ell}, \hat{\boldsymbol{\Phi}}^{\ell}\}) \tag{62}$$

where $S$ is an action. However, in their theory the single-site distribution for hidden layer $\ell$ have the form

$$p_{\ell}(h) \propto \exp \left( -\frac{1}{2} \sum_{\mu\nu} h_{\mu} h_{\nu} [\boldsymbol{\Phi}^{\ell-1}]_{\mu\nu}^{-1} - \frac{1}{2N} \sum_{\mu\nu} \phi(h_{\mu})\phi(h_{\nu})\hat{\Phi}_{\mu\nu}^{\ell} \right), \tag{63}$$

which reveal that the non-Gaussianity is explicitly suppressed with respect to $N$. As they focus on large width networks $N \gg 1$, they expand the single-site densities at leading order

$$p_{\ell}(h) \approx \exp \left( -\frac{1}{2} \sum_{\mu\nu} h_{\mu} h_{\nu} [\boldsymbol{\Phi}^{\ell-1}]_{\mu\nu}^{-1} \right) \left[ 1 - \frac{1}{2N} \sum_{\mu\nu} \phi(h_{\mu})\phi(h_{\nu})\hat{\Phi}_{\mu\nu}^{\ell} \right] \tag{64}$$

which enables computation of *Gaussian* averages with a perturbed density.

Some key differences between our approach and this approach is that

- Our theory does not allow for linearization of the preactivation densities unless the richness hyperparameter $\gamma_0$ is explicitly made small. We therefore do not rely on linear response theory to solve our equations but rather confront the min-max problem directly.

- Our theory results in *deterministic kernels* rather than random matrices for the kernels $\Phi_{\mu\nu}^{\ell}$. The kernels arising from a proportional limit are random matrices. Reducing the noise from random initialization results in better performance as we discuss in Appendix C.

## B.3. Feature Learning At Infinite Width By Rescaling Likelihood

An alternative approach, in analogy to mean-field/$\mu$P scaling in the dynamics case (Mei et al., 2018; Chizat et al., 2020; Geiger et al., 2020; Yang & Hu, 2022; Bordelon & Pehlevan, 2022), one could instead reparameterize the network definition so that feature learning persists as $N \to \infty$ even if $P$ is kept constant. This approach was pursued for Bayesian networks by Yang et al. (2023). However, they do not reparameterize the definition of the network predictor so their saddle point equations for the kernels are not in agreement with ours. To see this concretely in the case of a one hidden layer linear networks. For direct comparison, the posterior kernels satisfy the following saddle point equations in our theory and theirs at zero temperature

$$\begin{cases} \Phi = \boldsymbol{C}^{-1} - \gamma_0^2 \boldsymbol{\Phi}^{-1} \boldsymbol{y}\boldsymbol{y}^{\top} \boldsymbol{\Phi}^{-1} & \text{Ours} \\ 0 = \boldsymbol{C}^{-1} - \boldsymbol{\Phi}^{-1} \boldsymbol{y}\boldsymbol{y}^{\top} \boldsymbol{\Phi}^{-1} & \text{Theirs} \end{cases} \tag{65}$$

where the second equation is provided in Equation 128 of Yang et al. (2023). We can obtain their result from our result by taking an ultra-rich limit $\gamma_0 \to \infty$ and rescaling the kernel as $\boldsymbol{\Phi} = \gamma_0 \bar{\boldsymbol{\Phi}}$ which gives

$$\boldsymbol{C}^{-1} = \bar{\boldsymbol{\Phi}}^{-1} \boldsymbol{y}\boldsymbol{y}\bar{\boldsymbol{\Phi}}^{-1}. \tag{66}$$

We thus see that our scaling allows for more intermediate behaviors between the ultra-rich $\gamma_0 \to \infty$ and the lazy learning $\gamma_0 \to 0$ limits. Further, Yang et al. (2023) utilize a different spline-based numerical method to approximate the preactivation density under the posterior, whereas we utilize importance sampling.

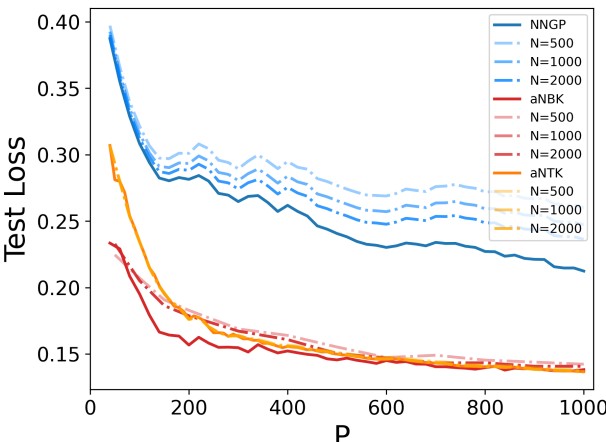

*Figure 9.* Finite-width effects in NTK and $\mu$P parameterizations. *Blue* curves are for different $N$ values, showing that NNGP predictor is not consistent across network widths. We compare predictions of finite width $N$ networks (as in (Pacelli et al., 2023)) to the NNGP infinite width limit. Finite width effects are instead negligible in $\mu$P parameterization, where finite $N$ networks trained with Langevin consistently lie on the theory full *orange* and *red* curves.

## C. Finite-width effects

In principle, in $\mu$P, the kernels $\Phi_{\mu\nu}^{\ell}$ and predictions $f_\mu$ at width $N$ exhibit $\mathcal{O}\left(\frac{1}{\sqrt{N}}\right)$ fluctuations around their limiting values (Bordelon & Pehlevan, 2024). Because these fluctuations generically add variance to the predictor (see (Li & Sompolinsky, 2021; Pacelli et al., 2023)), they increase the test loss compared to the large $N$ limit as we show in Figure 9. Here, we compare the finite width *light blue* effects in 1HL fully connected architecture (parameterized with NTK parameterization) at finite width and finite $\alpha = P/N = \Theta_N(1)$, with *orange* and *red* curves, which are the finite width effects in our mean-field parameterization. *Light blue* curves in this plot are the predictor performance as obtained in (Pacelli et al., 2023). *Light orange* and *light red* curves are the finite width effects of our aNTK and aNBK predictors respectively.

When $N$ is comparable to $P$ in either parameterization, the kernels $\mathbf{\Phi}^{\ell}$ should actually be thought of as *random matrices* with significant deformations to their spectra compared to the $N \to \infty$ limit.

## D. Deep Bayesian CNNs

In this section, we describe the Bayesian posterior of a Deep convolutional neural network (CNN) for infinitely many channels. Here, we need to add an index $a$ for each weight $W_{ija}^{\ell}$ in order to account for the filter value at spacial displacement $a$ from the filter center at each layer, while $\mathcal{S}^{\ell}$ is the spatial receptive field at layer $\ell$. The $L-1$ hidden layers of the CNN can be expressed as

$$h_{\mu ia}^{1} = \frac{1}{\sqrt{D}} \sum_{j=1}^{D} \sum_{b \in \mathcal{S}^0} W_{ijb}^{0} x_{\mu,j,a+b} \tag{67a}$$

$$h_{\mu ia}^{\ell+1} = \frac{1}{\sqrt{N}} \sum_{j=1}^{N} \sum_{b \in \mathcal{S}^\ell} W_{ijb}^{\ell} \phi(h_{\mu,j,a+b}^{\ell}) \tag{67b}$$

$$f_\mu = \frac{1}{\gamma_0 N} \sum_{i=1}^{N} \sum_{a} w_{ia}^{L} \phi(h_{\mu ia}^{L}) \tag{67c}$$

From these definitions, the partition function turns out to be

$$Z = \int \prod_{\ell=0}^{L-1} \prod_{ijb} dW_{ijb}^{\ell} \prod_{ia} dw_{ia}^{L} e^{-\frac{\beta}{2}\gamma_0 N^2 \sum_\mu (y_\mu - \frac{1}{\gamma_0 N} \sum_{i=1}^{N} \sum_a w_{ia}^L \phi(h_{\mu ia}^L))^2 + \frac{\lambda}{2} \sum_{\ell=0}^{L-1} \sum_{ijb} (W_{ijb}^\ell)^2 + \frac{\lambda}{2} \sum_{ia} (w_{ia}^L)^2} \tag{68}$$

By imposing the pre-activation definitions with the use of the integral representation of some Dirac delta functions as we did in Eq. (20), we can integrate out the weights contribution and just move in the space of representations, and get

$$Z = \int \prod_{\ell=0}^{L-1} \prod_{\mu i a} \frac{dh_{\mu i a}^{\ell+1} d\hat{h}_{\mu i a}^{\ell+1}}{2\pi} \int \prod_{\mu} \frac{ds_\mu d\hat{s}_\mu}{2\pi} e^{i\sum_\ell \sum_{\mu i a} h_{\mu i a}^{\ell+1}\hat{h}_{\mu i a}^{\ell+1} + i\sum_\mu s_\mu \hat{s}_\mu} \int \prod_\ell \prod_{ijb} dW_{ijb}^\ell \prod_{ia} dw_{ia} e^{-\frac{\beta}{2}\gamma_0 N^2 \sum_\mu (y_\mu - s_\mu)^2}$$

$$\times e^{\frac{\lambda}{2}\sum_\ell \sum_{ijb}(W_{ijb}^\ell)^2 + \frac{\lambda}{2}\sum_{ia}(w_{ia}^L)^2 - i\sum_\ell \sum_{\mu i a} \hat{h}_{\mu i a}^{\ell+1}\left(\frac{1}{\sqrt{N}}\sum_{j=1}^N \sum_{b \in \mathcal{S}^\ell} W_{ijb}^\ell \phi(h_{\mu,j,a+b}^\ell)\right) - i\sum_\mu \hat{s}_\mu \left(\frac{1}{\gamma_0 N}\sum_{i=1}^N \sum_a w_{ia}^L \phi(h_{\mu i a}^L)\right)} \tag{69}$$

$$Z = \int \prod_{\ell=1}^{L-1} \prod_{\mu\nu} \prod_{aa'} \frac{d\Phi_{\mu\nu,aa'}^\ell d\hat{\Phi}_{\mu\nu,aa'}^\ell}{2\pi} \int \prod_{\mu\nu} \frac{d\Phi_{\mu\nu}^L d\hat{\Phi}_{\mu\nu}^L}{2\pi} e^{N\sum_{\ell=1}^{L-1}\sum_{\mu\nu,aa'} \Phi_{\mu\nu,aa'}^\ell \hat{\Phi}_{\mu\nu,aa'}^\ell + N\sum_{\mu\nu} \Phi_{\mu\nu}^L \hat{\Phi}_{\mu\nu}^L}$$

$$\times \int \prod_\mu \frac{ds_\mu d\hat{s}_\mu}{2\pi} e^{+i\gamma_0 N \sum_\mu s_\mu \hat{s}_\mu - \frac{\beta}{2}\gamma_0 N^2 \sum_\mu (y_\mu - s_\mu)^2 - \frac{1}{2}\sum_{\mu\nu} \hat{s}_\mu \hat{s}_\nu \frac{1}{\lambda}\Phi_{\mu\nu}^L}$$

$$\times \left[ \int \prod_{\ell=1}^L \prod_{\mu a} \frac{dh_{\mu a}^\ell d\hat{h}_{\mu a}^\ell}{2\pi} e^{\sum_{\ell=1}^L \sum_{\mu a} h_{\mu a}^\ell \hat{h}_{\mu a}^\ell - \frac{1}{2}\sum_{\ell=1}^L \sum_{\mu\nu}\sum_{aa'} \hat{h}_{\mu a}^\ell \hat{h}_{\nu a'}^\ell \frac{\Phi_{\mu\nu,aa'}^{\ell-1}}{\lambda}} e^{-\sum_{\ell=1}^{L-1}\sum_{\mu\nu,aa'} \hat{\Phi}_{\mu\nu,aa'}^\ell \left(\sum_b \phi(h_{\mu,a+b}^\ell)\phi(h_{\nu,a'+b}^\ell)\right)} \right.$$

$$\left. e^{-\sum_{\mu\nu} \hat{\Phi}_{\mu\nu}^L \left(\sum_a \phi(h_{\mu a}^L)\phi(h_{\nu a}^L)\right)} \right]^N$$

$$= \int \prod_{\ell=1}^{L-1} \prod_{\mu\nu,aa'} \frac{d\Phi_{\mu\nu,aa'}^\ell d\hat{\Phi}_{\mu\nu,aa'}^\ell}{2\pi} \int \prod_{\mu\nu} \frac{d\Phi_{\mu\nu}^L d\hat{\Phi}_{\mu\nu}^L}{2\pi} e^{N\sum_{\ell=1}^{L-1}\sum_{\mu\nu,aa'} \Phi_{\mu\nu,aa'}^\ell \hat{\Phi}_{\mu\nu,aa'}^\ell + N\sum_{\mu\nu} \Phi_{\mu\nu}^L \hat{\Phi}_{\mu\nu}^L}$$

$$\times e^{-\gamma_0^2 \frac{N}{2} \sum_{\mu\nu} y^\mu \left(\frac{\mathbb{I}}{\beta} + \frac{\Phi^L}{\lambda_L}\right)^{-1}_{\mu\nu} y^\nu + N \ln \mathcal{Z}} \tag{70}$$

With saddle point equations

$$\Phi_{\mu\nu,aa'}^\ell = \langle \sum_b \phi(h_{\mu,a+b}^\ell)\phi(h_{\nu,a'+b}^\ell)\rangle \quad \forall \ell = 1,\ldots,L-1 \tag{71a}$$

$$\hat{\Phi}_{\mu\nu,aa'}^\ell = \frac{1}{2\lambda}\langle \hat{h}_{\mu a}^\ell \hat{h}_{\nu a'}^\ell \rangle \quad \forall \ell = 1,\ldots,L-1 \tag{71b}$$

$$\Phi_{\mu\nu}^L = \langle \sum_a \phi(h_{\mu a}^L)\phi(h_{\nu a}^L)\rangle \tag{71c}$$

$$\hat{\Phi}_{\mu\nu}^L = -\frac{\gamma_0^2}{2\lambda}\sum_{\alpha\beta} \left[\frac{\mathbb{I}}{\beta} + \frac{\Phi}{\lambda}\right]^{-1}_{\mu\alpha} y_\alpha y_\beta \left[\frac{\mathbb{I}}{\beta} + \frac{\Phi}{\lambda}\right]^{-1}_{\beta\nu} \tag{71d}$$

## E. DMFT review

In this section, we briefly recall the dynamical mean field theory derivation of a gradient flow dynamics for a multi-layer fully connected neural network. As clarified in the main text, we are interested in a fully connected feedforward network with $L$ layers, each of width $N$, defined as

$$f_\mu = \frac{1}{\gamma} h_\mu^{L+1} = \frac{1}{\gamma\sqrt{N}} w^{(L)} \cdot \phi(h_\mu^L), \qquad h_\mu^{\ell+1} = \frac{1}{\sqrt{N}} W^\ell \phi(h_\mu^\ell), \qquad h_\mu^1 = \frac{1}{\sqrt{D}} W^{(0)} x_\mu \tag{72}$$

where each trainable parameter $W^\ell$ is initialized as a Gaussian random variable $W_{ij}^\ell \sim \mathcal{N}(0,1)$ with unit variance. Here, the gradient updates for the weights $W^\ell(t)$ and the output function $f_\mu$ are given by

$$\frac{dW^\ell(t)}{dt} = \frac{\gamma}{N}\sum_\mu \Delta_\mu(t) g_\mu^{\ell+1}(t)\phi(h_\mu^\ell(t))^\top - \lambda W^\ell(t), \quad \frac{df_\mu}{dt} = \sum_{\alpha=1}^P K_{\mu\alpha}^{\text{NTK}}(t,t)\Delta_\alpha(t) - \lambda\kappa f_\mu \tag{73}$$

where $\Delta_\mu(t) = -\frac{\partial \mathcal{L}}{\partial f_\mu(t)}$ represents the pattern error signal for a pattern $\mu$, and $\boldsymbol{g}_\mu^\ell(t) = \sqrt{N}\frac{\partial h_\mu^{L+1}(t)}{\partial \boldsymbol{h}_\mu^\ell(t)}$ captures the backpropagated gradients flowing from the downstream layers. Instead, the term $\phi(\boldsymbol{h}_\mu^\ell(t))$ involves the activations of the $\ell$-th layer and reflects the forward pass contribution to the weight update.

As specified in the main text, for the representer theorem to be valid in this case, we restrict to $\kappa$ degree homogeneous network, whose output scales as $f(a\boldsymbol{\theta}) = a^\kappa f(\boldsymbol{\theta})$. Here, the predictor dynamics is governed by the driving force of the error signal propagated through the *adaptive Neural Tangent Kernel* $K_{\mu\alpha}^{\text{aNTK}}(t,t') = \frac{\partial h_\mu^{L+1}(t)}{\partial \boldsymbol{\theta}} \cdot \frac{\partial h_\alpha^{L+1}(t')}{\partial \boldsymbol{\theta}}$. This quantifies the interaction between parameter gradients for outputs of pattern pairs $(\mu, \nu)$ at times $(t, t')$. The homogeneity factor $\kappa$ appears in the second term from the weight decay contribution. Some quantities of interest to define are the forward and gradient kernels at each layer

$$\Phi_{\mu\nu}^\ell(t,t') = \frac{1}{N}\phi(\boldsymbol{h}_\mu^\ell(t)) \cdot \phi(\boldsymbol{h}_\nu^\ell(t')), \quad G_{\mu\nu}^\ell(t,t') = \frac{1}{N}\boldsymbol{g}_\mu^\ell(t) \cdot \boldsymbol{g}_\nu^\ell(t') \tag{74}$$

which allows to rewrite $K_{\mu\nu}^{\text{aNTK}}(t,t') = \sum_{\ell=0}^L G_{\mu\nu}^{\ell+1}(t,t')\Phi_{\mu\nu}^\ell(t,t')$. If we take care of the initial conditions over the weights, in the DMFT limit where $N, \gamma \to \infty$ with $\gamma = \gamma_0\sqrt{N}$, we can determine the final kernels by solving the field dynamics

$$h_\mu^\ell(t) = e^{-\lambda t}\xi_\mu^\ell(t) + \gamma_0 \int_0^t dt' \, e^{-\lambda(t-t')} \sum_\nu \Delta_\nu(t') \, g_\nu^\ell(t') \, \Phi_{\mu\nu}^{\ell-1}(t,t')$$

$$z_\mu^\ell(t) = e^{-\lambda t}\psi_\mu^\ell(t) + \gamma_0 \int_0^t dt' \, e^{-\lambda(t-t')} \sum_\nu \Delta_\nu(t')\phi(h_\nu^\ell(t'))G_{\mu\nu}^{\ell+1}(t,t'). \tag{75}$$

In Eq. (75), both pre-activations $h_\mu^\ell(t)$ and pre-gradient signals $z_\mu^\ell = \frac{1}{\sqrt{N}}W^\ell g_\mu^{\ell+1}$ decouple over the neuron index and factorize over the layer index, and the contribution from initial conditions $\xi_\mu^\ell(t) = \frac{1}{\sqrt{N}}W^\ell(0)\phi(h_\mu^{\ell-1})(t)$ and $\psi_\mu^\ell(t) = \frac{1}{\sqrt{N}}W^\ell(0)g_\mu^{\ell+1}(t)$ is exponentially suppressed at large time $t$. Simulating a stochastic process like Eq. (75) requires keeping track of the entire history trajectory, and computing at each step produces of kernel matrices that have dimension $PT \times PT$. This scales cubically in both sample and time dimensions $\mathcal{O}_N(P^3T^3)$, allowing in principle to solve for the field dynamics when $P, T = \mathcal{O}_N(1)$. A sketch of an iterative algorithm procedure can be found in Algorithm 1, where given an initial guess on $\{\Phi^\ell, G^\ell\}_{\ell=1}^L$ one can compute $K^{\text{aNTK}}$ and solve for the predictor dynamics of Eq. (72) once drawn a number $\mathcal{S}$ of samples $\{\xi_{\mu,n}^\ell(t)\}_{n=1}^{\mathcal{S}} \sim \mathcal{N}(0, \Phi^{\ell-1})$, $\{\psi_{\mu,n}^\ell(t)\}_{n=1}^{\mathcal{S}} \sim \mathcal{N}(0, G^{\ell+1})$ and solved Eq. (75) for each $\{h_{\mu,n}^\ell(t), z_{\mu,n}^\ell(t)\}_{n=1}^{\mathcal{S}}$. A sketch of the solver can also be found in the main text Algorithm 2. As can be noticed in Figures 10, solving for the field dynamics gives a good agreement with simulations. Precisely, once knowing the predictor, one can study how both train and test losses updates during the training dynamics and for different values of $\gamma$. Fig. 10(b) (top left panel) shows that the *lazy* learning regime when $\gamma = 0$ does not allow the network to interpolate on the training data and leads to a higher test loss compared to the rich cases with $\gamma > 0$. Increasing $\gamma$ has also the effect of speed up learning, while for that given sample size value ($P = 200$ here) there exists an optimal degree of feature learning (i.e. $\gamma$) concerning the test loss. The specifics of the learning task can be found in the figure caption. Fig. 10 also shows the non-Gaussianity of the pre-activation and pre-gradient distributions once the system has thermalized, at the end of training. As mentioned above, because of the weight initialization as $W_{ij}^\ell \sim \mathcal{N}(0, 1)$, both $\{h, z\}$ are Gaussian distributed when the training starts. Then feature learning has the effects of accumulating non-Gaussian contributions as the training proceeds.

### E.1. DMFT for Convolutional Networks

In Fig. 11 we show how our DMFT theory can be extended to a two-layer CNN which is trained on CIFAR10 images on a subset of $P = 100$ (*left*) or $P = 1000$ (*right*) points. When data sample is small, for any value of $\gamma_0$ there is an optimal early stopping time for the best Test Loss performance.

The theory to get the predictor is an easy extension to the multi-layer fully-connected setting and can be found in (Bordelon & Pehlevan, 2022). However, we recall here for the sake of clarity how the field dynamic equations get modified in this setting. Again, in our notation $a$ is the spatial displacement from the center of the filter at each layer where $b \in \mathcal{S}^\ell$ is the spatial relative field at layer $\ell$ as in Eq. (D). The pre-activation definitions still remain the same as it is in Appendix D. In the same way, the gradient signal are now defined as

$$\boldsymbol{g}_{\mu,a}^\ell = \gamma_0 N \sum_b \frac{\partial f}{\partial \boldsymbol{h}_{\mu,b}^{\ell+1}} \cdot \frac{\partial \boldsymbol{h}_{\mu,b}^{\ell+1}}{\partial \boldsymbol{h}_{\mu,a}^\ell}. \tag{76}$$

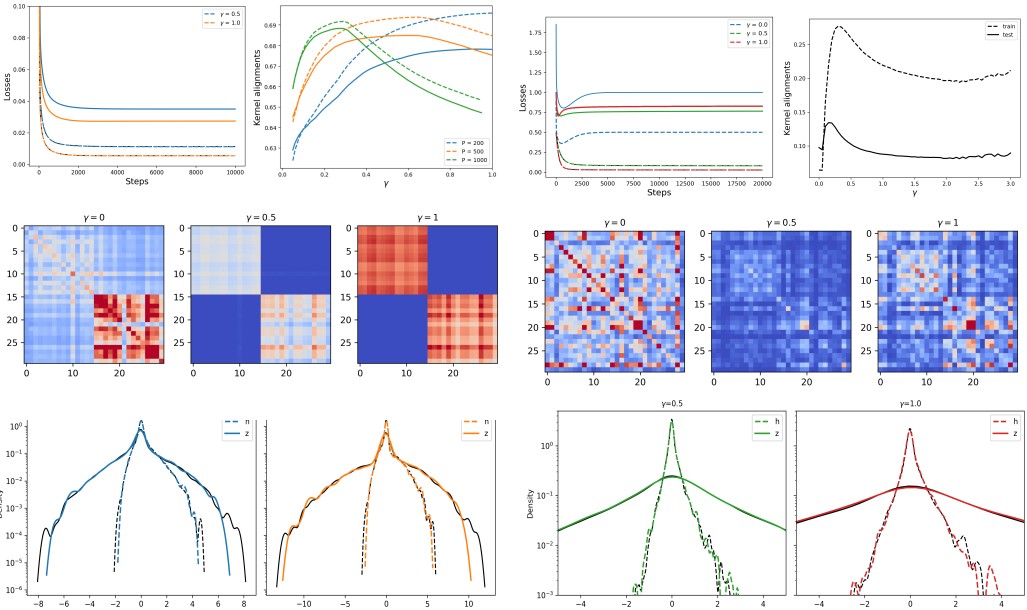

*Figure 10.* The training dynamics of two layer ReLU MLPs trained with weight decay. The richly trained networks achieve lower training and test errors at equal levels of regularization. Our theory can reproduce the final preactivation and pregradient densities in each setting.

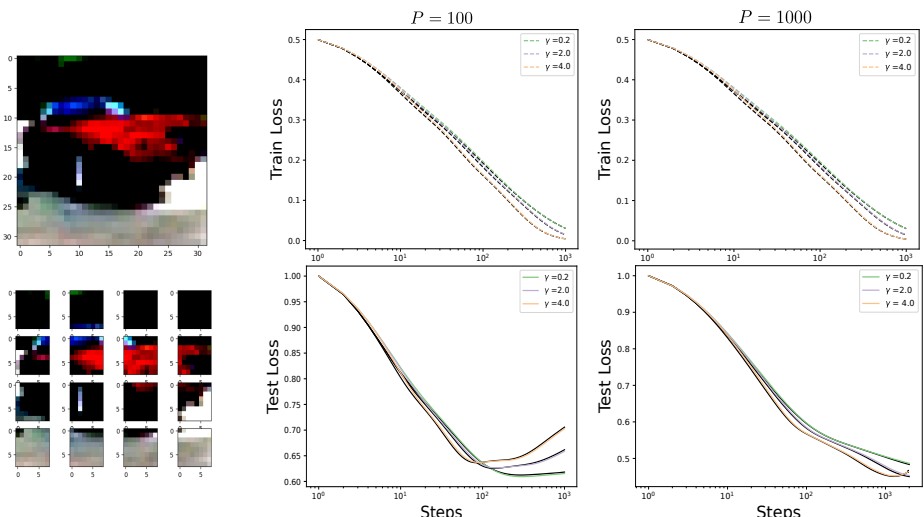

*Figure 11.* Training dynamics for a two layer CNN for varying richness $\gamma_0$ on CIFAR-10 images. The images are turned into patches before computing cross-spatial correlations in the data. The training dynamics for infinite width networks (black) is compared to training finite width networks. For small training set sizes, richer training can result in faster overfitting, while this effect is less severe when there is more data.

while the weight dynamics per filter is

$$\frac{d}{dt}\boldsymbol{W}_b^\ell(t) = \frac{\gamma_0}{\sqrt{N}}\sum_{\mu,a}\Delta_\mu \boldsymbol{g}_{\mu,a}^{\ell+1}\phi(\boldsymbol{h}_{\mu,a+b})^\top - \lambda\boldsymbol{W}_b^\ell(t). \tag{77}$$

Given that, the stochastic dynamics for the pre-activation and pre-gradient signals (similar to Eq. (75)) becomes

$$\boldsymbol{h}_{\mu,a}^{\ell+1}(t) = e^{-\lambda t}\boldsymbol{\xi}_{\mu,a}^{\ell+1}(t) + \gamma_0\int_0^t dt' e^{-\lambda(t-t')}\sum_{\nu,b,c}\Delta_\nu(t')\Phi_{\mu\nu,a+b,a+c}^\ell(t,t')\boldsymbol{g}_{\nu,c}^{\ell+1}(t')$$

$$\boldsymbol{z}_{\mu,a}^\ell(t) = e^{-\lambda t}\boldsymbol{\psi}_{\mu a}^\ell(t) + \gamma_0\int_0^t dt' e^{-\lambda(t-t')}\sum_{\nu,b,c}\Delta_\nu(t')G_{\mu\nu,a-b,c-b}^{\ell+1}(t,t')\phi(\boldsymbol{h}_{nu,c^\ell}) \tag{78}$$

where again $\boldsymbol{\xi}_{\mu,a}^{\ell+1}(t) = \frac{1}{\sqrt{N}}\boldsymbol{W}^\ell(0)\phi(\boldsymbol{h}_{\mu a}^\ell(t))$ and $\boldsymbol{\psi}_{\mu a}^\ell(t) = \frac{1}{\sqrt{N}}\boldsymbol{W}^\ell(0)\boldsymbol{g}_{\mu,a}^{\ell+1}(t)$ from the initial conditions. At large time $t$, as it is for the fully connected dynamics, the contribution form initial condition get suppressed and the fixed point predictor is a kernel predictor, being the feature and gradient kernels now

$$\Phi_{\mu a,\nu b}^\ell(t,t') = \frac{1}{N}\phi(\boldsymbol{h}_{\mu a}^\ell(t))\cdot\phi(\boldsymbol{h}_{\nu b}^\ell(t')), \quad G_{\mu a,\nu b}^\ell(t,t') = \frac{1}{N}\boldsymbol{g}_{\mu a}^\ell(t)\cdot\boldsymbol{g}_{\nu b}^\ell(t'). \tag{79}$$

The Neural Tangent Kernel is instead $K_{\mu\nu}^{\text{aNTK}}(t,t) = \sum_\ell\sum_{ab}\Phi_{\mu a,\nu b}^\ell(t,t)G_{\mu a,\nu b}^{\ell+1}(t,t)$ and the predictor again at convergence $f(\boldsymbol{x}) = \frac{1}{\kappa\lambda_L}\sum_\nu\Delta_\nu K^{\text{aNTK}}(\boldsymbol{x},\boldsymbol{x}_\nu)$.

## F. Fixed point structure of GD with Weight Decay

In what follows, we will be interested in the infinite time limit of a dynamics such that in Eq. (75). Prior work on the dynamics of L2 regularization in the kernel regime revealed that training a wide network for infinite time leads to collapse of the features and network predictor to zero (Lewkowycz & Gur-Ari, 2020). However, if one instead adopts a $\mu$P scaling, then it is *possible* to have a non-trivial fixed point at infinite width as the feature learning updates and regularization updates are of the same order (Bordelon & Pehlevan, 2022). This is because from Eq. , we realize that in the setting where $\lambda > 0$, not only the initial contribution of the fields dynamics are suppressed at large time $t$, but also the second terms contribute the most when the system has equilibrated, leading to a predictor which is a kernel predictor

$$f(\boldsymbol{x}_\star) = \boldsymbol{k}(\boldsymbol{x}_\star)^\top[\boldsymbol{K} + \lambda\kappa\boldsymbol{I}]^{-1}\boldsymbol{y} \tag{80}$$

Because of the simple interpretation of DNNs in this regime, we wish to say something about the fixed point structure of the field dynamics, which are

$$h_\mu^\ell = \frac{\gamma_0}{\lambda}\sum_\nu\Delta_\nu\Phi_{\mu\nu}^{\ell-1}\dot\phi(h_\nu)z_\nu \ , \ z_\mu^\ell = \frac{\gamma_0}{\lambda}\sum_\nu\Delta_\nu\phi(h_\nu^\ell)G_{\mu\nu}^{\ell+1}. \tag{81}$$

In what follows, we specialize to simple solvable cases to gain intuition for these fixed point constraints. In general, these constraints imply a set of possible joint densities over $h, z$ as well as determine the final feature and gradient kernels and the predictor.

### F.1. Two Layer MLP with whitened data

Let's consider a single data point with a white covariance matrix $K^x = 1$, label $y = 1$ and a transfer function $\phi(x) = \text{ReLU}(x)$. In this setting, the dynamics of training for the pre-activation and pre-gradient signals are

$$\frac{d}{dt}h(t) = \gamma_0\Delta(t)g(t) - \lambda h(t) \ , \ \frac{d}{dt}z(t) = \gamma_0\Delta(t)\phi(h(t)) - \lambda z(t). \tag{82}$$

At the fixed point, the following conditions are satisfied

$$h = \frac{\gamma_0}{\lambda}\Delta\dot\phi(h)z \ , \ z = \frac{\gamma_0}{\lambda}\Delta\phi(h). \tag{83}$$

In principle, there are infinitely many solutions to these equations, and combining them gives the following constraint on $h$

$$h = \frac{\gamma_0^2}{\lambda^2} \Delta^2 \dot{\phi}(h)\phi(h) = \frac{\gamma_0^2}{\lambda^2}\Delta^2 \phi(h). \tag{84}$$

Since we know that here $\phi(h) = \max(0, h)$, this means that the following two constraints on the pre-activation density must be satisfied

$$\forall h < 0 \quad p(h) = 0 \;,\; \Delta = \frac{\lambda}{\gamma}. \tag{85}$$

Lastly, we have the equation that fixes the value of the pattern error signal $\Delta$ through the predictor definition, which is

$$\Delta = 1 - \frac{1}{\gamma_0} \langle z\phi(h)\rangle = 1 - \gamma_0^{-1}\langle h^2 \rangle \tag{86}$$

implying that the second moment of $h$ must give

$$\langle h^2 \rangle = \gamma_0 - \lambda. \tag{87}$$

The solution here is correct if $\gamma_0 > \lambda$. Otherwise $\langle h^2 \rangle = 0$ and $p(h) = \delta(h)$. We can verify that this is true by comparing the pre-activation density of a two-layer MLP trained until interpolation with the theoretical predictions of the fixed points. As shown in Fig. 12, there is a sharp phase transition in the limiting density right when $\gamma_0 = \lambda$. Precisely, when $\gamma_0 < \lambda$, the effect of feature learning here is to kill the left-side of the distribution and adjusting the variance of $h > 0$ such that the constraint of Eq. (87) is satisfied. Another way of saying this is that in the infinite time limit $t \to \infty$, the $\{\gamma_0, \lambda_0\} \to 0$ limits do not commute. In the first case of Fig.12, when $\lim_{\gamma_0 \to 0, \lambda \to 0}$ we get a stable non-Gaussian behavior for the $p(h)$. In the second case of Fig. 12 we see a collapse when $\lim_{\lambda, \gamma \to 0}$ and nothing is learned by the network. In the same way, in the limit where we fix $t = \mathcal{O}_N(1)$ and study the ridge-less limit of a lazy network ($\gamma_0 = 0$), we recover the Neural Tangent Kernel predictor, and consequently the Gaussian pre-activation density at initialization.

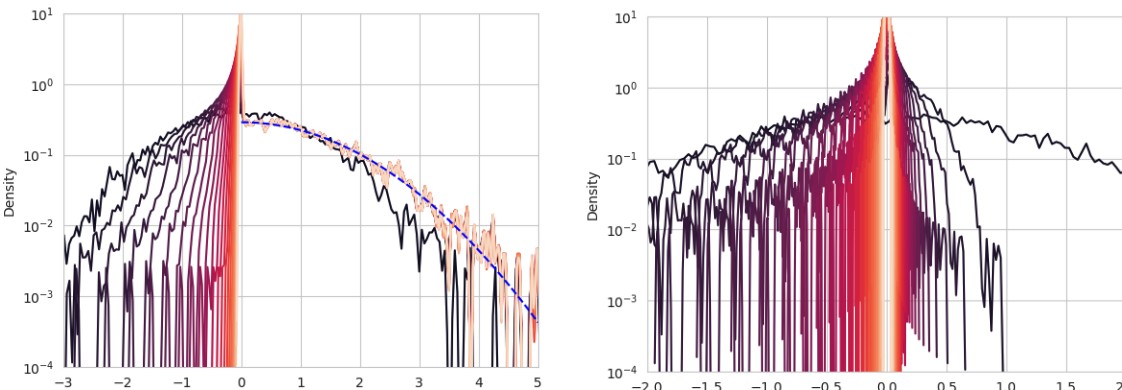

*Figure 12.* Pre-activation densities of a two-layer MLP trained with GD and weight decay at different times. Ligther colors represent the end of training. Dashed blue line is the theoretical prediction from the fixed point in the infinite time limit.

### F.2. Linear case

In principle, the constraints one gets by looking at fixed points of Eq. (81) fix the first two moments of the pre-activation density distribution, but are not enough to determine the full marginal $p(h)$. Indeed, this remains history dependent and one in principle has to track the entire update dynamics in order to get the full description. One possible way of understanding this is by looking at the simplest, linear case. Here, we initialize the weights of a two-layer MLP to be Laplace distributed. Here, we expect the distribution $p(h)$ to be Gaussian distributed if we train with weight decay until interpolation. Fig. 13 shows that training with weight decay to the fixed point does not recover a Gaussian single site density. But it does have the properties demanded by the saddle point equations, which are again $\Delta = \lambda/\gamma_0$; $\langle h^2 \rangle = \gamma_0 - \lambda$.

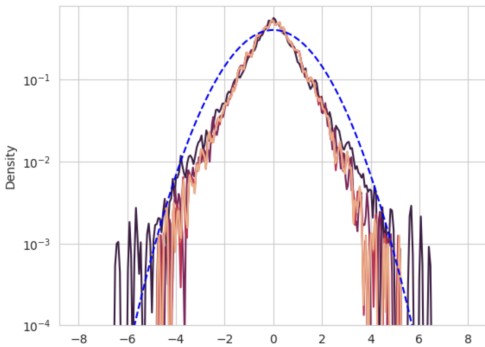

*Figure 13.* Pre-activation density of a two-layer MLP trained with $P = 1$ with a white covariance matrix. Dark colors correspond to early time training, being the weights initialized as Laplace distributed at $t = 0$. Light colors coincide with the end of training, when the system has thermalized. Blue dashed line is the theory prediction from the fixed point equations.

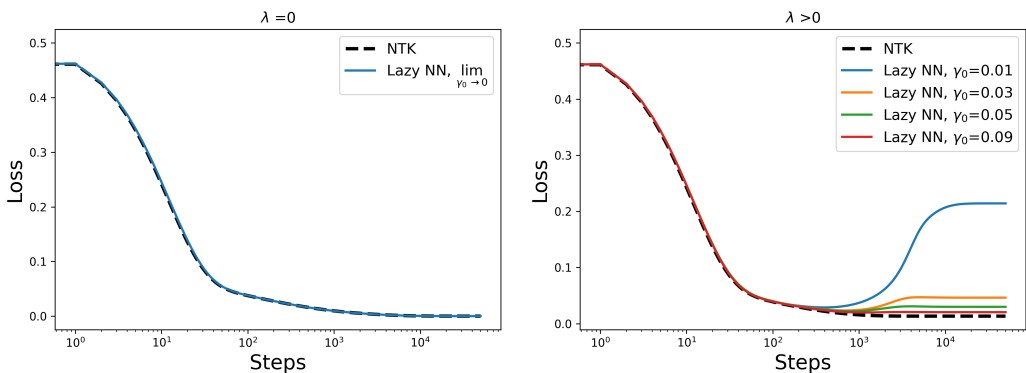

*Figure 14.* Weight decay in the lazy training regime can cause a model to "unlearn" and reduce its output after the features start to decay. Provided $\gamma_0$ is sufficiently large compared to $\lambda$, however, the final predictor will still be nontrivial, unlike the zero predictor obtained in NTK parameterization (Lewkowycz & Gur-Ari, 2020).

## G. Algorithmic implementation and numerical details

In this section, we provide more details regarding the numerical methods used for solving both the adaptive kernel theories and for training the finite width $N$ neural networks.

### G.1. Data pre-processing

As clarified in the main text, we focus on regression problems on two classes (respectively labeled as $-1/1$) for both MNIST and CIFAR10 datasets. For MLPs architectures, the input patterns have $28 \times 28$ pixels except when comparing the MLP test prediction with CNN as we did in Figure 5 . Indeed, for all the experiments except the one in Figure 5, we resize CIFAR10 images to $28 \times 28$ pixels by reducing their spatial resolution and then converting them to grayscale. In each case, we normalize the pixel values to have a consistent scale, and then we flatten the images as a $D = 784$ input vectors. For CNNs on CIFAR10, we preserve the spatial structure of the $32 \times 32$ pixel images rather than flattening them. Specifically, after normalization, each image is partitioned into non-overlapping patches of size $8 \times 8$, resulting in a multi-dimensional array that maintains the local neighborhood relationships. This representation enables the convolutional layer to apply a $K = 8 \times 8$ spatial filter directly, thereby effectively capturing localized features relevant to the regression tasks.

### G.2. Theory solver: Min-Max optimization for Bayesian DNNs

In this section, we provide more detail on the solver for the aNBK algorithm. We use automatic differentiation (via JAX) to compute gradients of the action $S(\{\boldsymbol{\Phi}^\ell, \hat{\boldsymbol{\Phi}}^\ell\})$ (Equation 6) with respect to the order parameters $\{\boldsymbol{\Phi}^\ell, \hat{\boldsymbol{\Phi}}^\ell\}$ (Bradbury et al.,

2018). A key challenge for this is to estimate the single site moment generating functions $\mathcal{Z}_\ell$ as a differentiable function of both variables $\boldsymbol{\Phi}^\ell$ and $\hat{\boldsymbol{\Phi}}^\ell$. To do so, we use importance sampling, by recognizing that, conditional on $\boldsymbol{\Phi}^{\ell-1}$, the $\mathcal{Z}_\ell$ can be expressed as an average of a nonlinear function with respect to a Gaussian with covariance $\boldsymbol{\Phi}^{\ell-1}$

$$
\begin{aligned}
\mathcal{Z}_\ell &= \left\langle \exp\left(-\frac{1}{2}\phi(\boldsymbol{h})^\top \hat{\boldsymbol{\Phi}}^\ell \phi(\boldsymbol{h})\right)\right\rangle_{\boldsymbol{h}\sim\mathcal{N}(0,\boldsymbol{\Phi}^{\ell-1})} \\
&\approx \frac{1}{B}\sum_{k=1}^{B}\exp\left(-\frac{1}{2}\phi(\boldsymbol{h}_k)^\top \hat{\boldsymbol{\Phi}}^\ell \phi(\boldsymbol{h}_k)\right).
\end{aligned}
\tag{88}
$$

In our code, in practice, each of the vectors $\boldsymbol{h}_k$ are i.i.d. draws from $\mathcal{N}(0,\boldsymbol{\Phi}^\ell)$, by setting $\lambda_\ell = 1\,\forall\ell\in\{L\}$.

At each step of the iteration scheme for the min-max solver, we resample a new batch of $B$ vectors $\boldsymbol{h}_k$ and use these to estimate $\mathcal{Z}_\ell$, which provides fresh samples at each iteration of the algorithm.

Our solver proceeds by alternately optimizing over the conjugate kernels $\{\hat{\boldsymbol{\Phi}}^\ell\}_{\ell=1}^{L}$ (via gradient ascent in the inner loop) and the kernels $\{\boldsymbol{\Phi}^\ell\}_{\ell=1}^{L}$ (via gradient descent in the outer loop). In practice, the inner loop runs for $t_{\text{inner}}\sim 200$ steps gradient steps, after which the outer loop updates $\{\boldsymbol{\Phi}^\ell\}_{\ell=1}^{L}$ given $\{\hat{\boldsymbol{\Phi}}^\ell\}_{\ell=1}^{L}$ for a maximum number $t_{\text{outer}}\sim 20000$ of iterations. The code implement this scheme as follows:

- **Kernel Initialization:** A function `init_kernels` computes the initial *lazy* guesses for kernels $\{\boldsymbol{\Phi}^\ell,\hat{\boldsymbol{\Phi}}^\ell\}_{\ell=1}^{L}$, given a data covariance matrix $\boldsymbol{\Phi}^0 = \frac{1}{D}\boldsymbol{x}\boldsymbol{x}^\top \in \mathbb{R}^{P\times P}$. At layer $\ell=1$, this involves generating a samples $= 20000$ number of Gaussian vectors $\boldsymbol{h}^1\in\mathbb{R}^{P\times\text{samples}}$ (by drawing from $\mathcal{N}(0,\boldsymbol{\Phi}^0)$ after computing the square-root of $\boldsymbol{\Phi}^0$) and then computing an empirical kernel via $\boldsymbol{\Phi}^1 = \frac{1}{\text{samples}}\phi(\boldsymbol{h}^1)\phi(\boldsymbol{h}^1)^\top \in \mathbb{R}^{P\times P}$. The same iterative procedure is applied at each layer, given $\boldsymbol{\Phi}^\ell = \frac{1}{\text{samples}}\phi(\boldsymbol{h}^{\ell-1})\phi(\boldsymbol{h}^{\ell-1})^\top$. For the conjugated kernels, $\hat{\boldsymbol{\Phi}}^\ell = \boldsymbol{0}^{P\times P}\,\forall\ell\in\{L\}$.

- **Single-Site Estimation:** A function `single_site` computes the log-normalized moment generating function by taking an average over the batch of $B$ samples as described above: $\log\mathcal{Z}_\ell = \log\left(\frac{1}{B}\sum_{k=1}^{B}\exp\left(-\frac{1}{2}\phi(\mathbf{h}_k)^\top\hat{\boldsymbol{\Phi}}^\ell\phi(\mathbf{h}_k)\right)\right).$

- **Action Function:** The overall action $S(\{\boldsymbol{\Phi}^\ell,\hat{\boldsymbol{\Phi}}^\ell\}_{\ell=1}^{L})$ is computed through a function `action` as in Equation (6).

- **Gradient Updates:** The gradients of $S(\{\boldsymbol{\Phi}^\ell,\hat{\boldsymbol{\Phi}}^\ell\}_{\ell=1}^{L})$ with respect to $\boldsymbol{\Phi}^\ell$ and $\hat{\boldsymbol{\Phi}}^\ell$ are computed using JAX's automatic differentiation and are used to update the corresponding variables via gradient ascent/descent (GD) as detailed above. Two separate learning rates (or update steps), `up_step_Phi` $\sim 1e-5$ and `up_step_hatPhi` $\sim 1e-4$, are used to control the outer and inner updates, respectively.

### G.3. Theory solver: DMFT dynamics

In this section, we detail the implementation of our DMFT dynamics solver, which is used to simulate the gradient flow dynamics with weight decay (given a regularization parameter $\lambda$ at each layer $\ell\in\{L\}$). Minor variations to take into account spatial positions are implemented for the CNN case. This solver is based on the DMFT derivation described in Section E and captures the evolution of the pre-activation fields $h_\mu$ and gradient signals $z_\mu$ via Monte Carlo sampling. For simplicity, we restrict to the case $L=1$ (i.e., two layers architecture).

The implementation proceeds as follows:

- **Fields initialization:** an initial set of samples$= 20000$ Gaussian pre-activation fields $\boldsymbol{h}(0)\in\mathbb{R}^{P\times\text{samples}}$ is generated by drawing Monte Carlo samples from $\mathcal{N}(0,\boldsymbol{K}^{\boldsymbol{x}})$, where $\boldsymbol{K}^{\boldsymbol{x}} = \boldsymbol{\Phi}^{\ell=0} = \frac{1}{D}\boldsymbol{x}\boldsymbol{x}^\top \in \mathbb{R}^{P\times P}$ is the Gram matrix of the $P$ data. In the same way, an auxiliary array of length samples$= 20000$, $\boldsymbol{z}(0)\in\mathbb{R}^{\text{samples}}$ is generated by sampling Gaussian random variables $z_s(0)\sim\mathcal{N}(0,1)$. From that, the gradient field $\boldsymbol{g}(0)\in\mathbb{R}^{P\times\text{samples}}$ is computed as $\boldsymbol{g}(0) = \dot{\phi}(\boldsymbol{h}(0))\odot\boldsymbol{z}(0)$ (Equation (13)).

- **Initial kernels and error signal:** Having $\boldsymbol{h}(0)\in\mathbb{R}^{P\times\text{samples}}$, the feature kernel is evaluated as $\boldsymbol{\Phi}^{\ell=1}(0) = \frac{1}{\text{samples}}\phi(\boldsymbol{h}(0))\phi(\boldsymbol{h}(0))^\top \in \mathbb{R}^{P\times P}$, being $\phi(\cdot) = \max(0,\cdot)$ the ReLU activation function. The initial gradient kernels are instead: $\boldsymbol{G}^{\ell=1}(0) = \frac{1}{\text{samples}}\boldsymbol{g}(0)\boldsymbol{g}(0)^\top \in \mathbb{R}^{P\times P}$, $\boldsymbol{G}^{\ell=2} = \boldsymbol{I}\in\mathbb{R}^{P\times P}\,\forall t\in\{T\}$.

According to Section E, the initial error signal for $P$ patterns $\boldsymbol{\Delta}(0) \in \mathbb{R}^P$ is computed as $\boldsymbol{\Delta}(0) = \boldsymbol{y} - \frac{1}{\gamma_0 \text{samples}} \phi(\boldsymbol{h}(0)) \boldsymbol{z}(0)$ because we focus on MSE loss $\mathcal{L} = \frac{1}{2} \sum_{\mu=1}^{P} (\Delta_\mu)^2$.

- **Field dynamics:** at each step for $T = 20000$ steps, the fields and the error signals are updated according to the dynamics

$$\boldsymbol{h}(t+1) = \boldsymbol{h}(t) + \eta \gamma_0 \left(\boldsymbol{g}(t) \odot \boldsymbol{\Delta}(t)\right) \boldsymbol{\Phi}^0(t) - \lambda \boldsymbol{h}(t) \tag{89a}$$

$$\boldsymbol{z}(t+1) = z(t) + \eta \gamma_0 \left(\phi(\boldsymbol{h}(t)) \odot \boldsymbol{\Delta}(t)\right) - \lambda \boldsymbol{z}(t) \tag{89b}$$

$$\boldsymbol{\Delta}(t) = \boldsymbol{y} - \frac{1}{\gamma_0 \text{samples}} \phi(\boldsymbol{h}(t)) \, \boldsymbol{z}(t). \tag{89c}$$

- **Loss:** Training loss $\mathcal{L}(t)$ are computed as the mean squared error over the corresponding $\boldsymbol{\Delta}(t)$, since $\mathcal{L}(t) = \frac{1}{2} \sum_{\mu=1}^{P} \Delta_\mu(t)^2$.

- **Return:** The function `solve_dynamics_reg` returns: list of losses over iterations $\mathcal{L}(t)$, the target vector $\boldsymbol{y}$, final states $\{\boldsymbol{h}(T), \boldsymbol{z}(T)\}$ for the fields and the final kernels $\{\boldsymbol{\Phi}^{\ell=1}(T), \boldsymbol{G}^{\ell=1}(T)\}$. From that, the aNTK kernel is computed as $\boldsymbol{K}^{\text{aNTK}}(T) = \text{Tr}(\boldsymbol{\Phi}^{\ell=0}(T) \boldsymbol{G}^{\ell=1}(T)) + \boldsymbol{\Phi}^{\ell=1}(T)$.

In our theory solver, learning rate $\eta = 1 \times 10^{-3}$, samples $= 20000$, and we span over $P, \gamma_0$ values.

### G.4. MLP Experiments

For designing the model, we construct a deep multilayer perceptron (MLP) in JAX. The architecture is built as a sequence of weight matrices with input dimension $D = 784$ (except for Figure 5 where the input dimension is $D = 1024$) and hidden layer width $N = 1024$ for a number $L + 1$ of hidden layers, each randomly initialized from a normal distribution such that $W_{ij}^\ell \sim \mathcal{N}(0, 1)$. Each hidden layer is followed by a ReLU non-linearity after being normalized according to the mean-field scaling as in Equation (2). The last layer returns a scalar output that serves as the regression prediction.

For training, we either employ a variant of deep Langevin dynamics, or a gradient descent with weight decay dynamics according to Equation (3). The MLP is trained to minimize a regularized mean squared error (MSE) loss (rescaled by the factor $\frac{1}{2} \gamma_0^2 N$) in both cases. For Langevin training, the otimizer we use - `optax.noisy_sgd` - is designed to inject Gaussian white noise into the gradient updates at each iteration. This noise is drawn from a zero-mean Gaussian distribution whose variance is controlled by both the learning rate $\eta$ and the inverse temperature parameter $\beta^{-1}$ as in Equation (3). This white noise plays a critical role in approximating the posterior distribution (Equation (9)) over the network parameters.

For both Langevin and gradient descent dynamics, we use a weight decay contribution proportional to $\lambda$ as it is for (3) (in all our experiments we use $\lambda = 1$ for each layer, except when we compare with CNN test loss, where $\lambda = 1 \times 10^{-2}$). For Langevin, we average the $T = 20000$ steps fluctuations every 1000 steps after $t > 5000$ epochs. We use a learning rate $\eta = 5 \times 10^{-4}$ and an inverse temperature $\beta = 50$. For gradient descent, we train until convergence for $T = 20000$ epochs and we use a learning rate $\eta = 1 \times 10^{-3}$. Both the experiments are performed by varying the sample size $P$ and the feature learning strength $\gamma_0$.

### G.5. CNN Experiments

We implement our CNN using Flax's Linen module in JAX. The model architecture and training strategy are designed to capture localized features from the animate/inanimate regression tasks on CIFAR10 images. The CNN consists of a single convolutional layer with a kernel size of $8 \times 8$ and stride equal to the kernel size. This choice effectively splits each $32 \times 32$ input image (with 3 color channels) into non-overlapping patches. We set the number of filters to $N = 1024$, and the convolution weights are initialized using a normal distribution with unit variance. To ensure that the variance is appropriately scaled, the convolutional output is divided by $\sqrt{3 \times 8^2}$. A ReLU activation is applied afterwards. This activated output is then flattened and fed to a dense layer that outputs a single scalar prediction; the dense layer's weights are further scaled by a factor of $1/(\gamma_0 N)$ to maintain the mean field scaling as in Equation (2).

The CNN is trained using a gradient-based optimization strategy over $T = 20000$ epochs, over a MSE loss. Before each parameter update, the gradients are scaled by a factor of $\frac{1}{2}\gamma_0^2 N$ and regularized with a weight decay term proportional to $\lambda$ as it is in the dynamics of Equation (3). The key hyperparameters for training the CNN are chosen as follows: learning rate $\eta = 1 \times 10^{-3}$, regularization $\lambda = 1 \times 10^{-2}$, and the experiments are performed by varying the number of training sample $P$ and the feature learning strength $\gamma_0$.

### G.6. Computational overhead

**Time for Kernels**   Algorithm (1): the most computationally expensive part is the Monte Carlo estimation of the non-Gaussian single-site density. At each inner step, we sample a batch of $B$ Gaussian vectors (as clarified in G.2) and estimate the single site desity function, an operation costing roughly $O(BP^2)$ per inner update (since it involves $P \times P$ matrix operations). Repeating this for $t_{\text{inner}}$ inner steps gives an overhead of approximately $O(t_{\text{inner}}BP^2)$ per outer update.

In contrast, when training a finite-width network via full Langevin dynamics, the end-to-end cost scales as $O(TN^2P)$, where $N = 1024$ is the network width. For comparable $T$ (we use $T = t_{\text{outer}} = 20000$ for both Langevin and theory solver), this is typically much lower than the overhead of the full theory solver, $O(t_{\text{outer}}t_{\text{inner}}BP^2)$, especially when the sample size $P$ is are large.

For deep linear networks—where the single-site density is Gaussian—the overhead of Algorithm (1) is significantly reduced because the conjugate kernels can be solved explicitly (see Eq. (15)), leading to a cost of $O(P^3)$ per step.

When comparing to DMFT-based analysis (Algorithm (2)), the full dynamics require handling $PT \times PT$ kernels, leading to a time complexity of $O(T^3P^3)$ for the complete dynamics. In our work, Algorithm (1) already represents an improvement over Algorithm (2). In Appendix E.1, we propose an $O(TP^3)$ version of Algorithm (2) by leveraging weight decay to directly solve the fixed-point equations.

To get an order of magnitude, in most of our simulations $t_{\text{outer}} = 10^4$, $t_{\text{inner}} = 2 \times 10^2$, and $T = 2 \times 10^4$, $B = 2 \times 10^4$ and sample size at most $P = 10^3$.

**Time for output predictions**   Generating the final predictions using Algorithm (1) requires approximately $O(t_{\text{outer}}t_{\text{inner}}BP^2)$ operations, as the procedure involves repeated sampling and kernel evaluations over the training dynamics. By comparison, Algorithm (2) incurs a heavier cost of $O(T^3P^3)$, while a simple kernel regression with a known kernel only requires $\mathcal{O}(P^3)$ time. End-to-end training of a finite-width network scales as $\mathcal{O}(TN^2P)$ (for both Langevin and gradient descent with weight decay dynamics), which is significantly more efficient in practice when $P$ is large.

**Memory for kernels**   For Algorithm (1), the memory footprint is primarily dictated by the storage of the kernels $\{\boldsymbol{\Phi}^\ell, \hat{\boldsymbol{\Phi}}^\ell\}_{\ell=1}^L$, each requiring roughly $\mathcal{O}(P^2)$ space. In contrast, Algorithm (2) have a memory cost $\mathcal{O}(P^2T^2)$. Meanwhile, finite-width network training typically demands a memory proportional to $\mathcal{O}(N^2)$.

## H. Glossary

Here we provide more explanation of our choice of terminology for the various kernel predictors. We use the letter K at the end of a name to indicate that this predictor is a kernel method with *no variance from random weight prior or initialization*. We use the prefix letter "a" to indicate an *adaptive kernel method* where the feature kernels adapt to the structure of the learning task.

- NNGP: the Gaussian process with $\Theta_N(1)$ variance for the network outputs under both the prior and the posterior. The mean of this process is a kernel method with the matrix $\boldsymbol{\Phi}^\ell$ in the lazy limit.

- NNGPK: the mean kernel predictor for lazy training with the initial final kernel neglecting. This corresponds to $N \to \infty$ first followed by $\gamma_0 \to 0$ in our parameterization. In this limit, there is no variance of the predictor under either prior or posterior.

- NTK: a kernel method for the initial neural tangent kernel at infinite width without any randomness or variability in the predictor from initialization. This is the predictor obtained in the NTK parameterization if the initial output of the model is subtracted off (ie if a centering operation is performed where $f(\boldsymbol{\theta}, \boldsymbol{x}) \to f(\boldsymbol{\theta}, \boldsymbol{x}) - f(\boldsymbol{\theta}_0, \boldsymbol{x})$).

- aNBK: the adapted Bayesian kernel method in our scaling limit. This corresponds to regression with the adapted $\boldsymbol{\Phi}^L$ kernel.

- aNTK: the adapted NTK kernel in our feature learning scaling limit. This corresponds to regression with the adapted $\boldsymbol{K}$ kernel.

