# OpenReview forum: "Adaptive kernel predictors from feature-learning infinite limits of neural networks"
_ICML.cc/2025/Conference — ICML 2025 poster_

### Official Review · Reviewer_Za7T · 2025-03-12

**Overall Recommendation:** 4

**Summary:**

The manuscript develops an approach for predicting deep non-linear neural networks outputs in mean-field and muP scaling. Central to their approach is the notion that kernels within each layer (Gram matrices) which, strictly speaking, are stochastic objects, concentrate leading to an effective description of the trained neural networks as a series of layer-dependent kernels (Adaptive Kernels) which obey deterministic non-linear matrix equations. They provide explicit expressions for these equations for linear settings (where they also solve the equations) and weakly-mean-field/perturbative settings (for the Bayesian case). In general, this leads to an effective GP like predictions though with those "Adaptive kernels" rather than the original data agnostic ones. The authors establish this for Bayesian inference and an NTK-like training away from the lazy regime (i.e. gradient flow). They also provide impressive numerical demonstrations which compare the solutions of their equations to the numerics and discuss statistics of activations.

**Claims And Evidence:**

The manuscript is generally well written and, while short of being a proof, shows solid arguments combined with numerical experiments.

**Essential References Not Discussed:**

See above.

**Ethical Review Concerns:**

None.

**Experimental Designs Or Analyses:**

I did not find any issues with the actual experiments.

**Methods And Evaluation Criteria:**

The methods are appropriate.

**Other Comments Or Suggestions:**

I'd be willing to raise my score if the above presentation issues are fixed.

**Other Strengths And Weaknesses:**

Stregths:

1. The generalization of kernel adaptation to the ``equilibrium" of gradient flow is a potentially interesting addition to the existing literature, however, only if this can be well separated from Ref. https://arxiv.org/abs/2205.09653.

2. The numerics are quite impressive compared to the existing ones on kernel adaptation.

3. The paper is also well written and, apart from some minor points, the appendices are quite detailed and accessible for those with previous knowledge in statistical mechanics.

In terms of weakness,

1. The positioning of this paper within the existing literature in the Bayesian setting should be improved. Similarly, since their NTK result involves running the full time dynamics, the novelty of their findings when compared to https://arxiv.org/abs/2205.09653 is not fleshed out.

2. The need for numerics in writing down the kernel equation (i.e. App. F.) as well as the need for numerics to solve these equations is an obvious shortcoming, though admittedly, it is hard to see how one can avoid the latter for realistic datasets. More assurance on the convergence/needed-sample-size for the sampling procedure in App. F.  can be helpful. For instance, how does the number of samples scale with the number of data-points and input-dimension? How does this behave on the strength of feature learning?

**Questions For Authors:**

See above

**Relation To Broader Scientific Literature:**

The main contribution of this work, in my mind, is numerical, especially in analyzing much larger datasets than previous works and deeper networks.

The authors, however, place too much focus on Bayesian results which already appeared, in minor variations, in various previous settings, often without noticing or informing the reader. Specifically, their equations for the kernels in the Bayesian appeared in a previous publication (https://proceedings.mlr.press/v235/fischer24a.html) as well as the perturbative analysis and the solution for a linear network. Furthermore, in that work it was established that these equations, upon taking the variational Gaussian approximation within each layer, also coincide with those of (https://www.nature.com/articles/s41467-023-36361-y). In addition, focusing on linear networks, the same equations were obtained by https://proceedings.mlr.press/v202/yang23k/yang23k.pdf. Earlier works by the latter authors also discussed the interpretation of trained networks in the rich regime via kernel flexibility/adaptation.

Similarly, their results regarding the NTK appear to largely overlap with Ref. https://arxiv.org/abs/2205.09653 and the *qualitative* addition made in this paper compared to that previous work are not clearly fleshed out. This seems to involve the introduction of weight decay which facilitates their numerical solutions, however, if so, this is appears as a somewhat technical increment (especially given that solvers are not posed in the center of the work).

**Theoretical Claims:**

I have checked their claims and agreed with their findings.

---

> ### Author Rebuttal · Authors · 2025-03-31
>
> ### Summary Clarification
>
> Regarding the theoretical results, to avoid possible confusion, we decided to change the name *adaptive Neural Network Gaussian Process Kernel* (aNNGPK) into *adaptive Neural Bayesian Kernel* (aNBK) in the newer draft of the paper. Our feature learning theory for deep non linear Bayesian networks does not resort to any perturbative approach nor any Gaussian approximation of hidden layer pre-activation distributions, which are non-Gaussian in the regime where $\gamma_0 = \Theta_N (1)$ (Fig. 3(b) in the main text).
>
> To the best of our knowledge, no prior work has attempted to solve the Bayesian feature learning saddle point equations non-perturbatively (Alg 1). We gave some intuition on how the fixed point equations for the kernels simplify in the $\gamma_0 \to 0$ limit using a perturbative approximation in Appendix A.5, but the theory and all the numerical results in the main text are not derived under this assumption.
>
> ### Relations to Other Works
>
> We thank the reviewer for pointing out these references, which are **all worth to acknowledge** in our "Related works" section. We had no intention of leaving them out on purpose, and we provided to expand and clarify how these works stand in comparison to our in the newer draft of the paper ("Related works" section) as well as next to the equations where they are presented.
>
> Regarding Ref. https://proceedings.mlr.press/v235/fischer24a.html, we give a detailed explanation on how and why our work differs from the one of Fischer et al. in the "Relationship to Fischer et al" of Reviewer MHbU (which we invite the Reviewer to read).
>
>
> The same can be said for Ref. https://www.nature.com/articles/s41467-023-36361-y, where the authors derive feature learning as a finite 1/width effect, being the predictor a *Gaussian Process Regression* (GPR) predictor. In the paper conclusions they say *Central to our analysis was a series of mean-ﬁeld approximations, revealing that pre-activations are weakly correlated between layers and follow a Gaussian distribution within each layer with a pre-kernel $K(l)$*. In the paper analysis they add *Remarkably, despite strong changes to the kernels and various non-linearities in the action, the pre-activation is almost perfectly Gaussian.*
>
> Again, our theoretical results are substantially different:
> 1. feature learning is not a 1/width effect in the parameterization we chose, and the kernels can exhibit arbitrarily large changes in their structure at infinite width (see Fig. (7) in the Appendix) when $\gamma_0 = \Theta_N(1)$.
> 2. The predictor has not the form of a GPR, but instead is a kernel predictor with task-adaptive kernel where **the pre-activation distributions is highly non-Gaussian** in deep non-linear networks ( as in Equation (7)). Feature learning in our setting has the $\Theta_N(1)$ effect of accumulating non-Gaussian contributions in single site density measure.
>
> Regarding Ref.  https://proceedings.mlr.press/v202/yang23k/yang23k.pdf, what the authors define as a "The Bayesian representation learning limit" in Sec. 4.3 consists in sending the number of output features (what they call $N_{L+1}$) to infinity, which has the effect of rescaling the likelihood term in the action by $N$, but **they don't rescale the readout of the last layer weights as we did which leads to some differences in the resulting eqns**.
>
> As an example, consider linear network with $L=1$ hidden layer. Our equation is
> \begin{align}
>     \Phi^{-1} = C^{-1} - \gamma_0^2 \Phi^{-1} yy^\top \Phi^{-1}
> \end{align}
> This differs from the equation 128 of Yang et al which states
> \begin{align}
>     C^{-1} = \Phi^{-1} yy^\top \Phi^{-1}
> \end{align}
> which shows that our theoretical predictions are **different in general**, especially for small $\gamma_0$.
>
> We can obtain their equation from our result by taking a very rich limit $\gamma_0 \to \infty$ and rescaling the definition of $\Phi = \gamma_0 \bar{\Phi}$ leading to the equation
> \begin{align}
>     C^{-1} = \bar{\Phi}^{-1} yy^\top \bar{\Phi}^{-1} + \mathcal{O}(\gamma_0^{-1})
> \end{align}
>
> However our equation holds for a variety of richness levels, interpolating between lazy and rich regimes.
>
>
> ### What is new for GF/GD?
>
> Please see response to reviewer MHbU on this topic.
>
>
> ### Positioning of Paper in Bayesian NN Literature
>
> We thank the reviewer for pointing out these issues. In the newer paper draft, we expanded the "Related work" section and provided a more detailed comparison to these relevant prior works.
>
> ### Numerics Details
>
> Thanks for this important question. We expanded Appendix F with details concerning the theory solvers and the numerical experiments on finite width NNs. We have an entire new subsection to discuss the computational costs (time for kernels, time for final outputs and memory) to solve the theory compared to end-to-end training. We discuss this in detail in the answer of "Methods And Evaluation Criteria" for Reviewer eXoJ, which we invite the Reviewer to read.

---

> > ### Comment · Reviewer_Za7T · 2025-04-06
> >
> > Same as prev "Official Comment":
> >
> > I appreciate the authors' effort in addressing my comments related to other Kernel Adaptation works, which also appeared in one other review. However, there are still some lingering issues and statements which seem at odds with past literature. I think authors should find a way of referencing that clarifies these doubts. Below, I delve into various points which remain unclear, then give another technical comparison between the works which the author can use to verify the claims below.
> >
> > 1. The equations are the same as those obtained by Fischer et. al., apart from an inconsequential log(Det) term, which goes away in mean-field scaling/muP scaling.  Furthermore, those of Fischer et. al. coincide with those of Seroussi et. al. (prior to taking Gaussian pre-activation approximation) and thus also with the current manuscript. The latter work considered both standard scaling muP scaling, however,  mainly studied Gaussian pre-activations. The former work handled pre-activations with similar care as the authors, however, did not explicitly consider muP scaling, only standard scaling. The value I see in the current work is in clearly establishing kernel adaptation in Bayesian setting for muP scaling without VGA. Previously, that required delving into the appendices of Fischer/Seroussi et. al. to make sure that the computation still works.
> >
> > 2. Since the equations are the same at muP scaling, the level of non-perturbativeness is the same. In fact, one of the stated main points of Seroussi et. al. is the nonperturbative nature of their results. Specifically, the results neglect some 1/width corrections but keep various dataset-size/width corrections. All of these results are non-perturbative in the standard sense of the word, which is that one is keeping arbitrary orders of expansion parameters (but not necessarily all diagrams).
> >
> > 3. Similarly, due to the near-perfect identity between the equations, the mechanisms driving feature learning (kernel alignment to reduce the discrepancy of the average predictor) remain the same. The kernel fluctuation picture of Fischer et al. simply provides a different intuition to the same phenomenon, in particular one valid at finite N and P, following the notion that fluctuations and susceptibility are identical in equilibrium systems. Moreover, this is not a key part of the results of Seroussi et. al. which, as mentioned, coincides with the author's work prior to applying the Gaussianity approximation for the preactivations.
> >
> > On more technical grounds
> >
> > The similarities between Fischer et. al. and the current work are:
> >
> > - what the authors call the min-max principle in case of Fischer et al. 2024
> >    corresponds to the supremum condition on \tilde{C} (max, Eqs. 6,7 in Fischer et al.)
> >    and the stationary point in the kernel C (min, Eqs. (9, 11) in Fischer et al.)
> >
> > - on the level of equations, the only difference is that they neglect
> >    the determinant term (\propto P) from their integral over \hat{s} they get in
> >    their Eq. (21); this term goes away by their argument of keeping P
> >    finite, which allows them to disregard it compared to the terms \propto N;
> >    Another view is that muP scaling diminishes fluctuations, which likewise
> >    suppresses the term from the determinant, which equals the NNGP variance
> >
> > - Fischer et al. 2024 study the linear case in Appendix C
> >    to obtain the same equations as the authors in their section 4,
> >    Eqs. (15) (again, except that they lack the determinant term \propto P)
> >    In particular this consideration does not require any additional
> >    approximations on the side of Fischer et al., except the large
> > deviation principle, which the authors agree is identical in the two approaches
> >
> > - For non-linear activation functions, the authors do not present an
> > analytical solution of the min-max problem. Instead, they resort to sampling in
> > their algorithm to estimate the partition function Eq. (7), which has the same measure as
> >    Eq. (8) in Fischer et al.
> >
> > - Fischer et al. thus also present the same set of non-perturbative
> > equations, as this set of equations is formulated by a variational principle;
> > This set of equations does not contain any perturbative treatment in 1/N
> >
> > - In addition to the variational equations, Fischer et al. present a
> > perturbative result for small \tilde{C} in Eqs. (15, 16), which is an expansion close to the NNGP (where \tilde{C}=0); the authors likewise present a perturbative treatment in their Eqs. (47-49) around the NNGP.
> >
> > The only true differences which do not amount to a simple change of formalisms are:
> >
> > Fischer et al. study N \propto P -> infty (in the main text) and
> > N, P finite (in Appendix), but both are done with standard scaling for all weights. Generalization to muP is straightforward but has not been done there.
> >
> > The authors study muP scaling (mean-field scaling of readout weights,
> > but standard scaling of inner weights and P finite but N->infty.
> >
> > I'd be happy to understand if the authors agree.

---

> > > ### Author Response · Authors · 2025-04-06
> > >
> > > We thank the reviewer for their commments. We will address the detailed points below but first we want to start by saying we disagree with the high level conclusion that our work is only a simple change in formalism and makes identical predictions as the Fischer et al paper or the Seroussi et al papers. While there are mathematical similarities in our theories, we want to point out that the **particular scaling limit matters** and gives fundamentally different conclusions about how networks behave as $N$ is varied.
> > >
> > > 1. Our theory predicts **nonlinear** changes to the kernels at any fixed value of $P$ as $N \to \infty$. In this type of limit, the Fischer et al model predicts that the hidden layers recover the NNGP (lazy learning) kernels.
> > > 2. Because our predictor $f$ and kernels $\Phi$ concentrate in our scaling limit $N \to \infty$ in our scaling our theory can accommodate **other loss functions** (such as cross-entropy, hinge loss etc) easily whereas this would be extremely challenging in the scaling limits of prior works (the log-det term comes from a Gaussian integral over the *non-concentrating* $f_\mu$ random variables which arises from MSE). Our Appendix A is written for a **generic loss function**.
> > > 3. Fischer et al try to relate fluctuations in kernels due to random initialization to the scale of feature updates, but this equivalence **explicitly relies on linear response theory for small perturbations from NNGP**. Our theory has no fluctuations in the kernels, yet significant and non-linear relationships between kernel changes.
> > > 4. Empirical evidence from many works, including experiments in our work (like Figure 7) actually indicate that $P/N$ effects are **detrimental** to performance in either NTK or $\mu$P scaling. Rather, the best performing model for a given dataset of size $P$ is the **infinite limit** one converges to in $\mu$P.
> > > 5. We show using our solver and in experiments, that in the general case, neither the Gaussian preactivation approximation nor the linearization of saddle point eqns around NNGP give good agreement with experiment when networks are tuned to the rich regime.
> > >
> > >
> > > Detailed comments
> > > 1. The equations of Fischer et al have similar structural form from ours once the log-det term is dropped (specifically our eqn 8 and their eqn 6), but the dependence of the equations on $N$ differs from ours and gives different results in our limit (see above). Specifically, we have to upscale the likelihood by $N$ and downscale the readout to be $1/N$ instead of $1/\sqrt{N}$. The combination of these rescalings causes the single site MGF to be $\mathcal Z = \left< \exp\left( - \frac{1}{2} \phi(h)^\top \hat{\Phi}^{\ell} \phi(h) \right) \right>_{h \sim \mathcal N(0, \Phi^{\ell-1})}$, which does not contain any factors of $1/N$ (and rescaling $\tilde{C} \to N \tilde{C}$ in their action will not recover the same saddle point as ours).
> > >
> > > 2, 3. On the level of non-perturbativeness, we really want to distinguish two types of non-perturbative theory that one can have for BNNs. First, one could capture all diagrams of order $P/N$ due to random matrix fluctuations of the $P\times P$ kernels. Second, one could capture non-perturbative (nonlinear coupled equations) for the **changes to the kernels**. We are primarily focused on the latter type of non-perturbativeness. Fischer et al's relationship between fluctuations and kernel changes due to feature learning **rely on linearization due to small kernel updates**.
> > >
> > > Technical points
> > >
> > > 1. We agree that fact that the large deviation equations take the form of a min-max problem is shared in both equations.
> > > 2. Unlike prior works, we **try solving the full min-max problem without linearization** for nonlinear networks which requires this sampling procedure.
> > > 3. Fischer et al's equation 6 is non-perturbative but then at large $N$ they linearize the saddle point equations around the NNGP solution.
> > >
> > > We hope that this response clears up some of the remaining confusion and that the reviewer will consider a reevaluation of our results.

---

### Official Review · Reviewer_eXoJ · 2025-03-13

**Overall Recommendation:** 5

**Summary:**

This paper expands kernel theory to study networks in the feature learning limit. Coupled systems of equations are found for two different "adaptive" kernels, the aNNGP and aNTK using physics-based methods and based on different scaling assumptions on weight variances and learning dynamics. Helpful interpretations for the equations are given as well as complete solutions in the linear network case. The kernels are derived at the fixed-point of training dynamics. Thus the theory doesn't describe the evolution of training, just the endpoint. The theoretical results are well-supported with numerical experiments, and a major result of the paper are the numerical methods, since in general these kernels are hard to solve in the nonlinear case.

**Claims And Evidence:**

The main claims that are made are theoretical and concern the match between theory and experiment. I will treat these in the theory section below.

**Essential References Not Discussed:**

N/A

**Experimental Designs Or Analyses:**

The experiments are standard image classification tasks and seem standard. I don't find there to be enough detail to fully evaluate the design of the numerical methods.

**Methods And Evaluation Criteria:**

The main "methods" in this paper, beyond theory, concern the numerical solution to the theory equations. My understanding of the past results in this area is that solving the coupled mean-field theory equations is very costly. In fact, I have heard it said that solving the theory equations is more costly than just performing end-to-end training of a deep network. For this reason, I wish the authors had offered more:
1. Explanation of the numerical methods, in sufficient detail to understand the implementation level.
2. An analysis of the cost of these numerical methods or at least some description of the cost relative to network training.
Overall, I found the numerical results to offer good support for the theory.

**Other Comments Or Suggestions:**

* Line 72 right col: "intermadiate" typo
* Figure 1: The y-axis "Test Prediction $\mu$P-AK" labels in this plot don't seem consistent with the rest of the paper. If the results in (b) and (c) are comparing NN output with aNTK and aNNGPK, I don't know what you're using for comparison in (a).
* In section 3.1, the authors move from having a single weight decay parameter $\lambda$ to layer-dependent ones $\lambda_\ell$. These need to be defined.
* Line 234 right col: the function $s(a\theta) = a^\kappa(\theta)$, is this supposed to be the pre-readout of the network, in which case you'll have dependence of $\kappa$ on the number of layers, or the homogeneity of a single layer nonlinearity $\phi$? This jumped out at me.
* Line 308, "Implement the non-Markovian dynamics" this is a big mystery to me. The appendix also doesn't really describe this, at least it doesn't describe a practical way to solve equation 62.
* Figure 3: "Langevin" isn't described, is this the same mystery method compared in Fig 1(a)?
* Supp, line 695, eqns 24, 25: The factor of 1/2 in front of the first sum and the $\gamma_0^2$ in front of the second sum in (26) are not consistent with the form of the exponent in (25), if $NS$ is supposed to be the exponent. This appears to be a typo.
* Supp, line 1323, distribution $\mathcal{N}(0,\Phi^{\ell-1})$ seems like it should be $\mathcal{N}(0,\Phi^{\ell-1}/\lambda_{\ell-1})$ to be consistent with the main text.
* Supp, line 1333, I found this sentence confusing in terms of which functions are being optimized for which variables. To find the dual variables $\hat{\Phi}^\ell$ you have to optimize $Z_\ell$; to find the non-dual ones it seems you have to optimize $Z_{\ell+1}$. This should be spelled out in a bit more detail, in terms of what order you do the optimization and for which variables.

**Other Strengths And Weaknesses:**

I find the paper clearly written, for the most part, with good intuition provided for the forms of the equations. I like that the theory is able to predict overfitting and the generalization error for realistic networks (CNNs not just MLPs) on real data. The only real weaknesses I see are with respect to clarity.

**Questions For Authors:**

Can you please expand the detail of the numerical methods explanation and/or release the code that solves the theory? This would be helpful for reproducibility.

Does the numerical approach offer significant advantages over previous methods, and how does it compare to training a network end-end? What kind of scaling does it have relative to, say, a kernel method with known kernel? This is relevant to the introduction's argument that "directly training infinitely-wid feature learning networks" could be a promising future approach.

**Relation To Broader Scientific Literature:**

I believe that the authors are well-aware of related work in this area and appear to have addressed most of the important related work.

**Theoretical Claims:**

I'll admit I'm not an expert in the specific methods used by the authors, but I did read the derivations. I didn't notice any real errors but potentially caught some typos that I'll enumerate below. The methods presented are non-rigorous, so no proofs need checking. I did find some of the derivations difficult to follow, in particular the steps where the saddle point equations are derived which were left out. I'd encourage the authors to include some of those for readers who might want a hand filling in the details.

---

> ### Author Rebuttal · Authors · 2025-03-31
>
> ### Analysis on the Cost of Solving Mean Field Eqns
>
> We thank the reviewer for this important question, and in the new draft (Appendix F) we gladly added an extensive discussion on numerical methods and computational costs.
>
> Regarding the cost of solving for Algorithm (1) compared to the network training, we can elaborate about the computational requirements, which is the time for kernels $\{\mathbf \Phi^{\ell}, \hat{\mathbf \Phi}^{\ell}\}_{\ell=1}^L$.
>
>  If our neural network has an input dimension $D$ fixed and a number $P$ of training points, our infinite width limit theory predicts that in order to solve for the kernels to get the network predictor, one has to solve the $\text{min-max}$ optimization problem as in Equation (8). So, in principle, we have an inner loop that solves for $\hat{\mathbf \Phi}^{\ell} (\mathbf \Phi^{\ell})= \arg \max_{\{\hat{\mathbf\Phi}^\ell\}} S(\{ \mathbf\Phi^{\ell}, \hat{\mathbf\Phi}^\ell \})$ with a $t_{\text{inner}}$ number of gradient steps, and an outer loop which, given $\hat{\mathbf \Phi}^{\ell}$ it solves for $\mathbf \Phi^{\ell}(\hat{\mathbf \Phi}^{\ell})= \arg \min_{\{\mathbf\Phi^\ell\}} S(\{ \mathbf\Phi^{\ell}, \hat{\mathbf\Phi}^\ell \})$ for a $t_{\text{outer}}$ number of outer gradient steps.
>
> An estimate of the computational overhead per self-consistent (outer) update in the theory solver is approximately $\mathcal{O}(t_{\text{inner}} B P^2)$. Indeed, the most expensive operation in computing the action $S(\{ \mathbf\Phi^{\ell}, \hat{\mathbf\Phi}^\ell \})$ from Equation (6) is the Monte Carlo estimate of the non-Gaussian single site density $\mathcal Z_\ell$. At each inner step $t=1,\ldots t_{\text{inner}}$, we sample a batch of $B$ Gaussian vectors $\mathbf h_k$ and use these to estimate
>
> $$\mathcal{Z}_\ell  = \left<  \exp\left( - \frac{1}{2} \phi(\mathbf{h})^\top \hat{\mathbf{\Phi}}^\ell \phi(\mathbf{h}) \right) \right>$$
>
> $$\approx \frac{1}{B} \sum_{k=1}^B \exp\left( - \frac{1}{2} \phi(\mathbf{h}_k)^\top \hat{\mathbf{\Phi}}^\ell \phi(\mathbf{h}_k) \right) $$
>
> This operation has a cost of $\mathcal{O}(BP^2)$. Because we have to repeat the same sampling for $t_{\text{inner}}$ steps, we get $\mathcal{O}(t_{\text{inner}} B P^2)$. Notice that, in the case of deep linear network, because the single site is Gaussian, the computational overhead is much smaller, and since one can explicitly solve for the conjugated kernels $\hat{\mathbf\Phi}^{\ell}$ as in Equation (15), the cost of each step is instead $\mathcal{O}(P^3)$.
>
> For the full Langevin dynamics for $T$ gradient steps, the cost of training is instead $\mathcal{O}(T N^2 P)$, being $N$ the network width. This is much less than the $\mathcal{O}(t_{\text{outer}} t_{\text{inner}} B P^2)$ for the full theory if we suppose the number of gradient steps for convergence between theory and simulations ($T$) to be of the same order.
>
> Despite this being the case in general, for which working on optimizing the computational overhead of the theory can be a future work focus, our Algorithm (1) still represents a step forward compared to Algorithm (2) as introduced in Bordelon et al 2022. In the case of Algorithm (2), solving for the DMFT analysis has a time requirement that scales as $\mathcal{O}( T^3 P^3)$ for the full dynamics, because in this case the kernels are $PT \times PT$ matrices. For optimizing this Algorithm (2), in Appendix E.1 we show a very simple and concrete example on how with weight decay the fixed point equations of DMFT at convergence can be solved without integrating the full dynamics, which we see as a promising future direction.
>
> To get an order or magnitude, in our simulations: $N=1024$, $t_{\text{outer}}= 10^4$, $t_{\text{inner}}=2\times 10^2$, and $T=2 \times 10^4$, $B=2\times 10^4$ and sample size at most $P=10^3$.
>
> ### Derivations could be clearer
>
> We are working on a cleaner and exaustive Appendix A, especially concerning the saddle point for which we provided the derivation in the new Appendix A.
>
> ### Experimental Details
>
> We apologize for missing out the details concerning our numerical methods. We added a new section in Appendix F to help clarifying the settings.
>
> ### More Numerical Details
>
> We are currently working on releasing both the codes that solve for the theories, which we have all intention of doing. In the meantime, regarding the numerical methods explanation, we added two new sections (Appendix F.2/F.3) as mentioned.
>
> #### Time complexity compared to known kernel
>
> The time for final outputs are instead: $\mathcal{O}(t_{\text{inner}} t_{\text{outer}} B P^2)$ for Algorithm (1), $\mathcal{O}(T^3 P^3)$ for Algorithm (2), $\mathcal{O}(P^3)$ for a kernel regression with a known kernel, and $\mathcal{O}(TN^2P)$ for the full network dinamics.

---

> > ### Comment · Reviewer_eXoJ · 2025-04-02
> >
> > Thanks, I think this added detail (and public code) will greatly improve the relevance to others who are trying to work with these kernels.
> >
> > I see that I am the outlier in terms of positive reviews. I can't increase my score, so I wish you good luck.

---

### Official Review · Reviewer_MHbU · 2025-03-13

**Overall Recommendation:** 4

**Summary:**

The paper shows that a GP describes neural networks trained in the rich infinite-width regime with a data-dependent kernel. Two different settings for the training dynamics are considered: one with a noise term and the other with no noise, where in both settings, there is a weight decay term. These correspond, respectively, to adaptive versions of NNGP and NT kernels. Explicit expressions for the kernel predictors and prescriptions to calculate them numerically are provided, using saddle point equations and dynamical mean field theory. It is shown that the adaptive kernels achieve lower test loss on benchmark datasets compared with kernels derived from the lazy regime.

**Claims And Evidence:**

As far as I can see, all claims are clearly stated and well-supported by theoretical or empirical evidence.

**Essential References Not Discussed:**

Here are some missing refs:

1. The paper by Fischer et al. 2024 is missing, see “Relation To Broader Scientific Literature”.

2. Lines 79-81: “interpreting the dynamics of gradient flow with added white noise as sampling the weights configuration from a Bayesian posterior”, the following papers are worth citing: Welling and Teh 2011, Naveh et al. 2021, Mingard 2021.

3. Under: Related works - Neural networks as kernel machines, it's worth mentioning and differentiating your work from Avidan et al. Your work is well separated from that paper but might seem superficially similar for some readers.

4. Lines 131-132: “Some works pursue perturbative approximations to the posterior in powers of 1/width”, the following papers are worth citing: Antognini 2019, Naveh et al. 2021.

• Fischer et al. 2024 - Critical feature learning in deep neural networks

• Avidan et al. - Connecting ntk and nngp: A unified theoretical framework for neural network learning dynamics in the kernel regime

• Welling and Teh 2011 - Bayesian learning via stochastic gradient Langevin dynamics

• Naveh et al. 2021 - Predicting the outputs of finite deep neural networks trained with noisy gradients

• Mingard 2021 - Is SGD a Bayesian sampler? Well, almost

• Antognini 2019 - Finite size corrections for neural network Gaussian processes

**Experimental Designs Or Analyses:**

I found the experiments to be generally sound and valid. A few comments on the figures:

1. Figure 3a - I don't understand the difference between NNGP and NNGPK here.

2. Figure 2c - are the cols going from $\ell=1$ to $\ell=L$ ?

3. Figure 3c, 2nd line - what is GF(wd)? is this the ground truth?

4. Figure 5 - I did not find a mention of what data was used (CIFAR10 ?)

**Methods And Evaluation Criteria:**

The proposed methods and evaluation criteria make sense for the problem or application at hand. For the numerics, providing more details could be useful.

**Other Comments Or Suggestions:**

Typos and other small issues:

1. Figure 1a panel caption (right under the figure) reads $\lambda>0$ and I believe should be $\lambda=0$.

2. Figure 1 main caption -

(a) reads: “When $\lambda=0$ the NN predictor of gradient flow *with* weight decay is not a kernel predictor” - should read “*without* weight decay”.

(b) “... is not a kernel predictor” - why not?

3. Line 181: know -> known.

4. Line 221: the inline eq for $K^{\mathrm{aNTK}}$ seems to be missing $\mathrm{lim}_{t\to\infty}$ and $\mu,\nu$.

5. Line 277: the inline eq for $s_{\mu}$ seems to be missing $\mu,\nu$.

6. Eq 17 lines 2,3 - $c\to c_{L}$ ?

**Other Strengths And Weaknesses:**

Weaknesses:

1. The main weakness I see is the lack of novelty that I elaborate on under “Relation To Broader Scientific Literature”.
*Edit*: I have raised my score to 4, provided that the authors made a substantial revision to acknowledge these prior works and differentiate their contributions.

2. More details on the experiments could help.


Strengths:

1. The paper is well structured, clearly written, and relatively easy to follow, given the heavy math involved.

2. It gives some nice intuitions.

**Questions For Authors:**

1. This paper studies feature learning and adaptive kernels in the $N\to\infty$ limit arising due to $\gamma_{0}>0$ but with finite $P$. The paper mentions other studies that consider the same problem but in the lazy regime with $\gamma_{0}=0$ and large $P$ (e.g. Li and Sompolinsky 2021, Fischer et al. 2024, Seroussi et al. 2023, Rubin et al. 2024 etc), but it would be interesting to comment on how the present theory behaves in this regime and if there is a simple relation between the scaling of $\gamma$ and $P$ that would yield similar results. For example, in the deep linear case, perhaps one could directly compare to the aforementioned papers. I'm curious to get your reaction to this.

2. What is new in the NTK part other than taking $t\to\infty$ and adding weight decay? Do you still need to numerically compute the time integrals?

3. Why is it obvious that gradient flow with $\lambda=0$ is not a kernel predictor?

4. AFAIK, the lazy scaling is still the one used by practitioners - would you advocate for the adoption of the rich scaling in real world settings? or is it still mainly a theoretical tool?

5. Eq. 11 - for the NTK theory $\kappa$ plays the role of $\beta^{-1}$ in the NNGP theory:

(a) in theory, what happens for non-homogenous DNNs, e.g. with biases and/or non-homogenous activation functions (e.g. sigmoid, tanh)?

(b) in the experiments, did you try activation functions other than ReLU?

6. What is the dimension $D$ of the inputs in your experiments? did you do some dim reduction for the CIFAR10 data?

7. The non-Gaussianity in your experiments always seems to be unimodal, can you say if there are other settings where we would expect a multi-modal distribution?

8. From figure 3a it seems that aNNGPK is slightly better than aNTK - do you think that is the typical case? can you speculate why?

**Relation To Broader Scientific Literature:**

Here are my main concerns with this paper:

There are two main parts to the contributions of this paper: the NNGP setting and the NTK setting. As far as I can see, there is limited novelty in both settings and insufficient acknowledgement of earlier related papers.

In the NNGP setting the theory leads to almost the same equations as in the missing reference Fischer et al. 2024 (see below), where the differences are only due to slightly different conventions and the choice of focusing on the regime where $P\sim O_{N}\left(1\right)$ [an explicit resetting of the noise $\kappa\to\kappa/\gamma_{0}^{N}$ to account for the $\mu$P scaling and ignoring terms of order $P$ such as the determinant term]. Can you point to any new result in your paper that does not already appear in this previous paper? Furthermore, I think that it would be fairer to give more emphasis in acknowledging other earlier works that are very much related e.g. the cited papers Seroussi et al. 2023 and Rubin et al. 2024a.

As for the NTK part, I am also not sure what is novel here relative to the earlier DMFT paper, other than adding the weight decay term and taking $t\to\infty$. Can you please clarify?

• Fischer et al. 2024 - Critical feature learning in deep neural networks

**Theoretical Claims:**

I checked the theoretical claims and found no major issues with their correctness.

---

> ### Author Rebuttal · Authors · 2025-03-31
>
> ### Relationship to Fischer et al
>
> We thank the reviewer for pointing us toward Fischer et al 2024 which we have added a detailed comparison to in the main text, as well as expanded the discussion of Seroussi et al and Rubin et al.
>
> Our work differs from Fischer et al in a number of important ways. These differences lead to different predictors and interpretations on what drives feature learning.
>
> 1. We study a different feature-learning scaling limit than Fischer et al., therefore the features learned by our networks differ than Fischer et al. We keep $P= \Theta_N(1)$, and take width $N\to \infty$ in $\mu P$ parameterization (where $f= \frac{1}{N} w \cdot \phi$ and likelihood scaled up by $N$). They study a proportional limit in NTK parameterization. These scalings behave differently (**see Figure 7**).
> 2. We arrive at kernel equations in different ways. In our setting, the deterministic saddle point equations for the learned kernels in this feature learning limit are **exact** at infinite width.
>
> In our theory the saddle point equations for the kernels at infinite width are provided in Eqn (31), which involve **non-Gaussian single-site densities** and no linearization around the lazy solution.
>
> In Fischer et al., the saddle points equations for kernels at fixed $P$ and infinite width are instead (with 1/$N$ corrections included) provided in Eqns 15-16, which are clearly **different** from ours and involve **Gaussian single-site averages**. In their equations feature learning contribution vanishes when $N\to \infty$ at fixed $P$; they are effectively computing weak corrections to the lazy limit and fluctuations of the kernels.
>
> 3. We develop a solver for the min-max problem (Alg 1) for the exact eqns, which we verify this with experiments. To our knowledge no prior work has attempted to solve the exact saddle point equations.
>
> In our work, it is not the finite width fluctuations that drive useful feature learning as in Fisher et al, but feature learning can already be accessed in the $N \to \infty$ limit where kernels concentrate. This is at odds with one of the primary takeaways of Fischer et al who argue that "larger kernel fluctuations enable stronger feature learning."
>
> That said, both Fischer et al and our work identify kernels and conjugate kernels as the key order parameters ($\{ \Phi, \hat\Phi\}$ in our work $\{ C, \tilde{C} \}$ in their work) and both papers express a large deviation principle for them (equation 8 in our work and equation 6 in their work).
>
> We will cover this in greater detail in App B.
>
> ### What is new in GD setting?
>
> The key contribution in this part is recognizing that knowledge of the **final NTK** is sufficient to characterize the learned solution, but only if weight decay is included. This suggests that alternative solvers focused on weight decay fixed points could circumvent the need to simulate through the dynamics.
>
> ### More details about Expts
>
> Additional experimental details are provided in the new version of Appendix F.
>
> ### Questions
> 1. If $\gamma_0 = \frac{1}{\sqrt N}$, then we recover the NTK scaling and corrections to this lazy limit can be exacted at small but finite $\gamma_0$ by considering a perturbation series for the NNGP (or NTK) kernel in powers of $1/N$. In Appendix B we discussed finite width effects in NTK parameterization. We stress that at fixed $\gamma_0$ and finite width, our theories (Fig 7) are different from these theories.
> 2. In principle, one no longer needs to compute the time integrals if weight decay is used (see App E.1). In practice, however, we have been mainly integrating through the dynamics to solve for the final weight decay solution, but fixed point solvers for GD+weight decay could potentially be accelerated compared to DMFT.
> 3. In general, for gradient flow, one needs to know the full history of the NTK over training.
> 4. $\mu$P scaling is becoming the industry standard in many SOTA models, including for transformers https://arxiv.org/abs/2203.03466, https://arxiv.org/abs/2407.05872, https://x.com/andrew_n_carr/status/1821027437515567510. Part of the attraction for practitioners is the consistency of dynamics across widths https://arxiv.org/abs/2305.18411.
> 5. Our Bayesian theory has no assumptions on the homogeneity of the activations function. We did experiments in this setting with $\tanh$, that we will gladly add in App A.
>
> Instead, GF + weight decay for non-homogeneous activation functions no longer satisfies a representer theorem for the $\text{aNTK}$ kernel as in Equation (10).
>
> 6. For CIFAR we resized the images to $28\times 28$ pixels and then we convert them to grayscale.
>
> 7. The simplest case is $P=1$ and $L=1$ with $\phi(h) = \tanh (h)$,
> $$p(h)  \propto \exp \Big(-\frac{1}{2}C^{-1} h^2 - \frac{1}{2}\hat{\Phi}\tanh(h)^2 \Big)$$
> Since $\hat{\Phi} < 0$, $p(h)$ can develop a Mexican-hat shape.
>
> 8. We suspect that Langevin noise acts as an entropic regularizer that reduces overfitting in small data regimes.

---

> > ### Comment · Reviewer_MHbU · 2025-04-05
> >
> > Dear authors,
> >
> > Thank you for your thoughtful and detailed response.
> >
> > I acknowledge the merit and clarity of your work, particularly in terms of exposition and experimental validation. That said, I still believe that some important clarifications are needed before I would feel confident raising my score. In particular, it would help greatly if your revised version includes the expanded discussion on related work that you outlined in your response.
> >
> > Here are my main points:
> >
> > Relation to Prior Work: Fischer et al. and Seroussi et al. use the same action as your work before applying a variational Gaussian approximation. The only difference lies in the log-determinant term, which vanishes in the μP scaling when $P=O_N(1)$. Thus, the mechanism by which feature learning arises is not fundamentally different across these works. The distinction in scaling limits, while important, does not in itself establish a new conceptual framework.
> >
> > Feature Learning and Kernel Fluctuations: In the earlier works, feature learning is not attributed to kernel fluctuations per se. Rather, while the kernels themselves do concentrate, they adapt in a data-dependent way, leading to changes in the fluctuations of the pre-activations. This is a subtle but important point that challenges the stated contrast between your theory and that of Fischer et al.
> >
> > Non-Perturbative Nature of Prior Work: Although the previous approaches involve certain approximations (e.g., neglecting specific classes of terms), they are non-perturbative in the technical sense. That is, they cannot be expressed as a finite-order expansion in $1/N$. This is more than semantics — it speaks directly to how the theory captures feature learning effects beyond the lazy regime. For this reason, I feel that the novelty claims in your paper could benefit from a more careful positioning relative to this body of work.
> >
> > NTK Setting and Potential Solvers: As noted in my original review, the paper is clearly written and includes compelling empirical results. With respect to novelty, I would say the main contributions are (a) a cleaner derivation of the feature-learning kernel in μP scaling, and (b) the inclusion of weight decay in the NTK setting. The latter is a potentially interesting direction, especially if it opens the door to practical solvers that avoid simulating the full training dynamics. I would encourage the authors to elaborate — or at least speculate — more concretely on what such solvers might look like. This would significantly strengthen the NTK part of the paper.
> >
> > Again, I appreciate your efforts and the interesting directions raised in the rebuttal. With additional clarity on the points above, especially the precise differences from prior work and the implications for practical algorithms, I would be open to revisiting my assessment.
> >
> > Best regards

---

> > > ### Author Response · Authors · 2025-04-05
> > >
> > > We thank the reviewer for their additional feedback and comments. We really want to properly include and represent these prior works and are trying to make sure we understand them correctly compared to our results.
> > >
> > > 1. Relation to prior work: we agree that since both papers are studying Bayesian MLPs that the structure of the posterior when written in terms of $\{ \Phi, \hat\Phi \}$ is the same (up to log-det forms that disappear in our $1/N$ readout scaling with $P=O(1)$). This paper exists within a set of prior works that operate in this conceptual framework (as did the Fischer et al work). We plan to give Fischer et al and Seroussi et al etc much more credit in the related works for identifying the large deviation principle.
> > >
> > >
> > > One novel point that we wanted to make in our study (which of course shares similarity to prior BNN papers) is that **parameterization matters** in Bayesian networks (ie the $1/\sqrt{N}$ vs $1/N$), even at the level of model performance, which is why we referenced Fig 7. In this Figure, the blue lines represent NTK parameterization where decreasing width **causes worse performance**, while in the rich $\mu$P network, models **quickly converge** to their limiting large width behavior (note that $\alpha = P/N$ is even greater than unity by the end of this plot!). This is consistent with empirical studies on non-Bayesian networks (see for instance [this paper](https://arxiv.org/pdf/2305.18411) Figure 14).  We will try making this point more explicit in the next version of the draft.
> > >
> > > 2. Feature learning and kernel fluctuations. We apologize for confusion and are trying to understand your response.
> > >
> > > There are many places in Fischer et al where fluctuations in the $C$ variables due to finite size are explicity cited as important for feature learning.
> > >     a) Figure 1 caption "Larger kernel fluctuations enable stronger feature learning"
> > >     b) In the introduction: "A wide distribution $p(C)$ leads to a rich prior and thereby enables strong adaptation to the training data"
> > >     c) On page 6, "finite-width effects lead to kernel corrections in the direction of the target kernel"
> > >     d) Section 4.1 titled "Fluctuations lead to feature learning" states that "larger fluctuations lead to stronger feature learning"
> > >
> > > The reviewer also claims that the kernels concentrate in the Fischer et al setting, but at large but finite $N$ or in a proportional limit, the kernels would be random variables with a distribution $p(C)$ that is not Dirac (while our theory gives $p(C) = \delta(C-C^*)$ ).
> > >
> > > Is the proper way to interpret these claims that both the perturbative feature learning expansion and an analysis of fluctuations around the saddle point **both invoke linear response theory** and explicitly solve a set of linear equations for $\delta C, \delta \tilde{C}$, where $\delta C$ measures deviations from NNGP? If so, is this statment more about the structure of the $\chi$ response variables and the $V$ functions controlling adaptations to kernels at leading order?
> > >
> > > In our scaling limit and theory (with the $1/N$ scaling) no such linearization can be performed in general, even at infinite width.
> > >
> > >
> > > 3. Non-Perturbative Nature of Prior Work: Thank you for pointing this out. We will be careful to mention that Fischer et al is a **nonperturbative description of a proprtional limit** with $\alpha = P/N$. We just wanted to say that our theory does not expand the result to leading order in $\gamma_0$ in nonlinear networks. We will be more clear about this and apologize for the mistake.
> > >
> > > 4. Novelty of our work: we agree that (a) BNN's in $\mu$P scaling and (b) NTK + weight decay are the two main novel results. We thank the reviewer for acknowledging them.
> > >
> > > 5. On the NTK + WD setting, we have started experimenting with a solver that explicitly tries to iterate fixed point equations for the **final $h, z$** variables (equation 68). To do this, we start from an initial guess for the kernels (such as the lazy solutions). Then we run a fixed point iteration scheme for 68 from many initial conditions to sample the space of solutions. Then the kernels $\Phi, G$ would updated as averages over these samples. This iteration procedure works for the DMFT equations for gradient descent at finite time, but is very costly to track for the full dynamics. For a monte-carlo batch of size $B$ and $P$ data points this kind of scheme would costs $\mathcal O(P^2 B)$ per iteration for the NTK + weight decay whereas tracking the dynamics with DMFT for $T$ training steps in a deep network would cost $\mathcal{O}( P^2 T^2 B )$ due to the memory terms that arise from the limiting dynamics. We plan to add more discussion of a potential algorithm to solve these equations in Appendix E.1.
> > >
> > > We sincerely hope that this resolves some issues and the reviewer would consider a reevaluation of our work, as it will be properly related to the prior literature.

---

### Official Review · Reviewer_upAJ · 2025-03-14

**Overall Recommendation:** 2

**Summary:**

This paper proposes two ways of taking the infinite-width limit of fully connected and convolutional neural networks trained by noisy gradient flow to arrive at kernel solutions in what the authors call the "rich regime", in which feature learning takes place, which contrasts with the "lazy regime", in which the weights stay in a neighbourhood of the values at initialisation. The first is the adaptive Neural Network Gaussian Process Kernel (aNNGPK), and the second is the adaptive Neural Tangent Kernel (aNTK). Derivations are given for these two solutions, as well as algorithms to compute them. No statistical results are given.

**Claims And Evidence:**

In general, I think studying the NTK in the feature-learning regime is a really interesting problem, and I thank the authors for investigating this problem. However, based on my knowledge and evaluation, I think this current submission is not ready for acceptance.

The authors claim in the introduction that the behaviour of neural networks in the infinite-width limit is "analysed", from which I expected some kind of statistical generalisation error, but what the authors give is simply a derivation of the kernel regression solution that they converge to. Moreover, these derivations are heavily dependent on previous results in both cases, and the references to those results are very weak. For the aNNGPK, the derivation of (9) from (3) is not given in the paper, and it is unclear which result from the previous works the authors used to arrive at this derivation. For the aNTK, the authors cite [Bordelon and Pehlevan, 2022] heavily, to the extent that they use notations and quantities from that paper without saying what they are. For example, $\xi^l_\mu(t),\psi^l_\mu(t)$ are said to be "stochastic processes inherited from the initial conditions", but nowhere are these initial conditions given, nor are the expressions of $\xi^l_\mu(t),\psi^l_\mu(t)$ given explicitly.

In general, the mathematical presentation is weak, with many inconsistencies, abuses of notations being used without being introduced, and results being used without derivation or precise citation.

Moreover, the "derivation sketches" are given without a precise statement of the results that they are supposed to prove. Why did the authors decide not to state the results as Theorems or Propositions, before giving the proofs?

**Essential References Not Discussed:**

There are of course many papers missing, but the literature in which this submission resides is vast and I wouldn't expect any paper in this field to be able to cite every paper in this field, so I do not count this against the submission.

**Experimental Designs Or Analyses:**

I spent less time on the experiments section. I couldn't immediately spot any problems.

**Methods And Evaluation Criteria:**

I spent less time on the experiments section. I couldn't immediately spot any problems.

**Other Comments Or Suggestions:**

15R: "analyzing expressivity" -> "analyzing the expressivity"

49R: For consistency with (1), $a^\mu$ -> $a_\mu$.

74L: "Further, complexity" -> "Further, the complexity"

72R: "intermadiate" -> "intermediate"

126: "at converge" -> "at convergence"

162R: (59) and (2) seem to be the same, but I think it would be much better here to refer to (2) than (59). I suspect this is due to a glitch in the labelling of the equations?

189L: "already know lazy" -> "already known lazy"

208L: Footnote 2 doesn't seem to exist?

138R: In (2), from what's written on 150R and 170R, I think there is a square-root missing in $N_L$ in the denominator right?

149R: If all the widths $N_l$, $l=1,...,L$ are assumed to be the same at $N$, then I suggest you just use $N$ everywhere. It is somewhat confusing, for example, to see both $N_L$ and $N$ being used on 170R and 171R.

174R: Is the superscript $L$ missing from the $\Phi$'s?

192R: I think it would be much better to denote the dependence of $S$ on $\boldsymbol{\Phi}$'s explicitly, as done in (8) for example.

209R: In (8), this is bad notation. The same symbols that are minned and maxed out on the right-hand side are used again on the left-hand side.

224L: "ie" -> "i.e."

232L: "means to solving" -> "means solving" or "means to solve"

222R: I think it would be more accurate to say that $K_{\mu\alpha}^{\text{aNTK}}$ is the adaptive Neural Tangent Kernel (Gram) matrix, rather than saying it is the kernel itself. Same goes for the gradient kernel $G^\ell_{\mu\nu}$. Also, for $\Phi^\ell(t,t)$, of course I can guess what this means but again this is mathematically bad notation, since, before, you used $\boldsymbol{x}$ as arguments in $\Phi$. Abuse of notations should be clearly stated if you're going to do it.

613: A new sentence should start here?

**Other Strengths And Weaknesses:**

In general, I think the presentation and structure of the paper could be improved a lot. For example, in Sections 3.1 and 3.2, there are "derivation sketches" but it is not immediately clear what these derivation sketches are supposed to prove. Also, see "Other Comments Or Suggestions" section for what is probably a non-exhaustive list of issues.

**Questions For Authors:**

175L: Here, $\delta$ is used without being introduced. What is this? Should it be there?

621: In (19), why is it that we have $\lambda_0,...,\lambda_L$, when in (18) we have just one $\lambda$ without subscript? Also, for consistency of notation, e.g. with 613, all the $W$'s should be boldface.

It may be that I'm mistaken, but I tried to derive (5) from (4) using $\sigma(s)=s$ and $\mathcal{L}(s_\mu)=(s_\mu-y_\mu)^2$, but I couldn't. Are you sure that the coefficients $\Delta_\mu$ are correct? Can you please spell out the derivations?

$\gamma$ is defined as $\gamma_0N$, and then for the NTK regime you say you take the $N\to\infty$ limit first then $\gamma_0\to0$ in Table 1. But this will result in $\gamma$ always being $\infty$ surely? Do you mean that you let $\gamma_0$ be some function of $N$ and take $N\to\infty$ such that $\gamma$ stays at a constant value or something like this? As far as I'm aware, usually in NTKs, you can simply take $\frac{d\boldsymbol{\theta}}{dt}=\nabla_{\boldsymbol{\theta}}\mathcal{L}$, possibly with a constant in front of the gradient. There shouldn't be a factor of $N$ that will take this to $\infty$.

I'm sorry if this is due to my ignorance, but is it obvious that (9) can be derived from (3) by taking fixed $\beta$ and taking $t\to\infty$? I looked at the two references you gave for this, but one of them is a book without reference to a specific result, and the other is a paper in which I still couldn't immediately find a result that implies (9).

**Relation To Broader Scientific Literature:**

I think the lazy regime is pretty saturated by now, and also deemed less interesting by the community. I think contributions in the feature-learning regime are extremely valuable, but also rigorous results in this regime are very difficult to obtain. Even though this submission does not aim at rigorous theoretical results, the goals of this submission are still welcome, but I unfortunately cannot give a very high evaluation for its execution.

**Theoretical Claims:**

It is hard for me to verify the correctness of the theoretical claims, given the mathematical presentation.

---

> ### Author Rebuttal · Authors · 2025-03-31
>
> We thank their reviewer for their detailed feedback and suggestions, which we try addressing below.
>
> ### How is the infinite limit analyzed?
>
> Here we provided exact expressions for the adaptive kernel predictors that one obtains in various feature learning limits through rigorous analyses of the underlying limits at fixed dataset. We believe our usage of the term "analysis" is appropriate, however we are happy to consider an alternative suggestion. We disagree that this is a "simple" derivation. Similar analyses, for example, providing kernel limits in the lazy regime without a study of the statistical generalization error have been very valuable in the past.
>
> ### Derivations Dependent on Prior Work / What is new?
>
>  We thank the reviewer for pointing out some of these deficiencies in our presentation. We didn't use any previous theoretical results for deriving the aNBK (adaptive Neural Bayesian Kernel, the old aNNGPK), because our theory is new (see "Relationship to Fischer et al" discussion in Rev. MHbU or "Relation to other work" in Rev. Za7T). The derivation of Equation (9) from (3) is a central result of Langevin dynamics and covered in textbooks such as (https://www.cambridge.org/core/books/statistical-physics-of-fields/06F49D11030FB3108683F413269DE945). We feel inclusion of this basic result in the paper is not necessary, however we will gladly provide a reference below.
> Regarding dependence on [Bordelon and Pehlevan, 2022], we have already included an Appendix Section D reviewing the necessary facts from that paper and pointed to it from the main text. We are working on extending this section and make it more clear.
>
>
> ### Add more precise statement of results (Theorem/Proof style)
>
> We thank the reviewer for this suggestion. We added statement results as Theorems in the newer draft version and improved the clarity of the assumptions required for our results. In particular, we improved the explanation and statement of equation 8 and provided a more detailed proof of this result (what was the derivation sketch).
>
> ### Appendix A confusing... How do we get generalization?
> Thanks for this point. We simply evaluate the predictions of our adaptive kernel predictors on a set of test points to measure/estimate the generalization error for that point. This is not meant to be an analysis of the data-averaged or worst-case generalization error. However, these predictions on test points will agree with neural network predictions with high probability over the random initialization of the network weights and the random Langevin noise in the case of Bayesian networks.
>
>
> ### Execution Issues with Result Statement beyond Lazy Limit
> We apologize if the results were not as clear as possible. We hope our responses above clarified our contributions. We are making changes in "Related works sections" to address these clarity issues and clarify our contributions.
>
>
>
> ### Questions
>
> 1. *$\delta$ is used without being introduced. What is this? Should it be there?*
>
> We will add that $\delta$ here is the **Dirac delta function** (a function with the property $\int dt' \delta(t-t') g(t') = g(t)$ for any $g$), which specifies that the noise in the Brownian motion is uncorrelated across time. To see the discrete time version of an uncorrelated process, we could define a collection of random variables $\{ \epsilon_t \}$ with covariance structure $\left< \epsilon_t^2 \right> = \sigma^2$ and $\left<  \epsilon_t \epsilon_{t'} \right> = 0$ for $t \neq t'$.
>
> The continuous time version of this is one where the covariance of the process $\{ \epsilon(t) \}$ is a Dirac delta function.
>
>
> 2. *What is up with Eqn 18?*
>
> We thank the reviewer for pointing out the typo in Eq.(18). We solved the inconsistency and fixed the notation for the $\mathbf W$s.
>
>
> 3. *How do we get eqn 5?*
>
> We apologize that this was unclear. We added a more detailed derivation of this in Appendix A.2.
>
> In the case of $\sigma(s)=s$ we have
>
> $$f(x) = \frac{\beta}{\lambda} \sum_{\mu} \Delta_\mu \Phi(x,x_\mu).$$
>
> We can solve for $\Delta_\mu \equiv -\frac{\partial L}{\partial s_\mu}$
>
> $$\mathbf \Delta = \mathbf y - \mathbf f= \mathbf y - \frac{\beta}{\lambda}\mathbf\Phi \mathbf \Delta=  \left[ \mathbf I + \frac{\beta}{\lambda} \mathbf\Phi \right]^{-1} \mathbf y$$
> as desired.
>
> 4. *Why does $\gamma$ show up in the dynamics?*
>
> The raw learning rate has to vary with $\gamma$ to maintain constant scale updates to the output to the network (see https://pehlevan.seas.harvard.edu/sites/g/files/omnuum6471/files/pehlevan/files/princeton_lecture_notes.pdf).
>
> 5. *Why do Dynamics Converge to the Posterior?*
>
> When $\beta = \Theta_N(1)$, the dynamics in Eq.(3) are known as *Langevin*, which are known to converge to a stationary density that has the form of Eq. (9). A useful and complete reference for this derivation can be found in https://www.thphys.uni-heidelberg.de/~wolschin/statsem23_6.pdf, Eq.(54), or in https://www.stats.ox.ac.uk/~teh/research/compstats/WelTeh2011a.pdf.

---

> > ### Comment · Reviewer_upAJ · 2025-04-03
> >
> > Dear Authors,
> >
> > Thank you for your rebuttal, and I am sorry that a few of the points were due to my ignorance. However, I think the manuscript would benefit from a thorough revision rather than just fixing the (probably non-exhaustive) points raised by myself and other reviewers, and also in light of the other reviews, I will unfortunately stick with my original evaluation.
> >
> > Best,
> > reviewer

---

> > > ### Author Response · Authors · 2025-04-05
> > >
> > > We thank and respect the reviewer’s decision. We are, however, sorry that the additional clarifications we provided - both to this reviewer and to others - were not sufficient to reconsider a revision of the score. In particular, we would like to highlight new discussions and points raised in the rebuttal (e.g., our Reply Rebuttal Comment to Reviewer MHbU), which we believe offer further insight into our contributions.
> > >
> > > We sincerely hope that the reviewer might consider reassessing the manuscript based on their own evaluation, independently of the opinions expressed in the other reviews.
> > >
> > > Bests,
> > > the authors

---

### Decision · Program_Chairs · 2025-05-01

**Decision:**

Accept (poster)

**Comment:**

This paper shows that infinite-width neural networks in the feature-learning regimes (e.g., via the mu-parametrization) also admit a kernel-regression/Gaussian-process view. This is useful since it enables further analysis of infinite-width neural networks. The authors show this theoretically, and provide numerical recipes to compute the derived data-dependent kernels.

While the reviewers pointed out that there are some crucial missing references, they agree that this work is novel and should be accepted. Feedback about the presentation of the paper also came up prominently during the review/rebuttal period. However, the majority of the reviewers think that this can be fixed easily.

In any case, I implore the authors to incorporate all reviewers' suggestions, especially re. references and presentation.